# Mitochondrial redox adaptations enable alternative aspartate synthesis in SDH-deficient cells

**Madeleine L Hart[1,2], Evan Quon[1], Anna-Lena BG Vigil[1], Ian A Engstrom[1], Oliver J Newsom[1], Kristian Davidsen[1], Pia Hoellerbauer[1], Samantha M Carlisle[3], Lucas B Sullivan[1]***

[1]Human Biology Division, Fred Hutchinson Cancer Center, Seattle, United States; [2]Molecular Medicine & Mechanisms of Disease Program, University of Washington, Seattle, United States; [3]Department of Chemistry and Biochemistry, New Mexico State University, Las Cruces, United States

**Abstract** The oxidative tricarboxylic acid (TCA) cycle is a central mitochondrial pathway integrating catabolic conversions of NAD$^+$ to NADH and anabolic production of aspartate, a key amino acid for cell proliferation. Several TCA cycle components are implicated in tumorigenesis, including loss-of-function mutations in subunits of succinate dehydrogenase (SDH), also known as complex II of the electron transport chain (ETC), but mechanistic understanding of how proliferating cells tolerate the metabolic defects of SDH loss is still lacking. Here, we identify that SDH supports human cell proliferation through aspartate synthesis but, unlike other ETC impairments, the effects of SDH inhibition are not ameliorated by electron acceptor supplementation. Interestingly, we find aspartate production and cell proliferation are restored to SDH-impaired cells by concomitant inhibition of ETC complex I (CI). We determine that the benefits of CI inhibition in this context depend on decreasing mitochondrial NAD+/NADH, which drives SDH-independent aspartate production through pyruvate carboxylation and reductive carboxylation of glutamine. We also find that genetic loss or restoration of SDH selects for cells with concordant CI activity, establishing distinct modalities of mitochondrial metabolism for maintaining aspartate synthesis. These data therefore identify a metabolically beneficial mechanism for CI loss in proliferating cells and reveal how compartmentalized redox changes can impact cellular fitness.

**\*For correspondence:** lucas@fredhutch.org

**Competing interest:** The authors declare that no competing interests exist.

## Editor's evaluation

Mutations in subunits of succinate dehydrogenase (SDH) are implicated in tumorigenesis, but it is not fully understood how cell proliferation occurs in cells that lack SDH. Hart et al. elegantly demonstrate that lack of mitochondrial complex I increases proliferation in cells lacking SDH. The authors have expertly dissected the mechanism behind the rescue and show that inhibition of complex I restores aspartate production by modulating NAD+/NADH. This study thus demonstrates that compartmentalized redox changes control distinct aspartate biosynthetic pathways that promote the proliferation of tumor cells.

## Introduction

Cell proliferation requires metabolic alterations to coordinate catabolism, the maintenance of cellular bioenergetics, with anabolism, the synthesis of macromolecules for cellular replication. Several metabolic pathways serve amphibolic roles, generating both bioenergetic molecules to support metabolic

homeostasis, like ATP, NADH, or NADPH, while also producing metabolic precursors used for biosynthesis of molecules such as nucleotides, amino acids, or lipids. Since these metabolic outputs are interconnected, coordinating metabolic pathway activities is critical for efficient cell proliferation. However, specific mechanisms for how metabolic alterations support cell proliferation remain poorly understood, especially in disease contexts where cell lineage, genotype, and environmental factors can all influence metabolic requirements (*Vander Heiden and DeBerardinis, 2017*).

Central to cell metabolism is the mitochondrion, a double membrane-bound organelle that plays critical roles in cell function. While mitochondria are often viewed through the catabolic lens of efficient ATP synthesis, many proliferating cells can meet their ATP demands from aerobic glycolysis alone yet require intact mitochondrial metabolism for cell proliferation (*Weinhouse, 1956*; *Tan et al., 2015*; *King and Attardi, 1989*). Two fundamental processes of mitochondrial metabolism are the tricarboxylic acid cycle (TCA) and the electron transport chain (ETC). The TCA cycle is an amphibolic pathway, serving as both the major catabolic source of mitochondrial NADH and as a primary anabolic route supporting aspartate biosynthesis. Metabolite progression through the TCA cycle is linked to ETC activity through two mechanisms: (1) the shared succinate dehydrogenase (SDH, also known as complex II) reaction, and (2) redox coupling, where complex I (CI) accepts electrons from mitochondrial NADH to yield NAD+, in turn driving oxidative TCA cycle reactions that convert NAD +to NADH. Disruptions to ETC complexes I, III, and IV diminish mitochondrial NAD +regeneration, reducing the cellular NAD+/NADH ratio, impairing the TCA cycle, and slowing cell proliferation. Supplementation with molecules that regenerate NAD+, such as pyruvate, alpha-ketobutyrate, or duroquinone, can restore the NAD+/NADH ratio and the proliferation of ETC impaired cells (*Birsoy et al., 2015*; *Luengo et al., 2021*; *Sullivan et al., 2015*). Similarly, heterologous expression of the bacterial NADH oxidase *LbNOX* can also reestablish NADH oxidation independent of the ETC and restore the proliferation of ETC impaired cells (*Titov et al., 2016*). These findings highlight that redox regulation is an essential function of ETC activity in proliferating cells.

A major metabolic consequence of decreased NAD+/NADH upon ETC impairment is depletion of the amino acid aspartate (*Birsoy et al., 2015*; *Sullivan et al., 2015*). Most cells are dependent on de novo aspartate synthesis since aspartate is poorly permeable at physiological concentrations (*Garcia-Bermudez et al., 2018*; *Sullivan et al., 2018*). Aspartate production occurs through transamination of the TCA cycle metabolite oxaloacetate, which is typically produced by NAD+-dependent reactions in the TCA cycle, linking aspartate production to NAD +regeneration by the ETC. Aspartate is a central metabolic node in proliferative metabolism, serving as a direct substrate for protein synthesis and as a precursor for the synthesis of other essential metabolites, including asparagine, arginine, and both purine and pyrimidine nucleobases. Thus, aspartate depletion likely impairs core metabolic processes necessary for cell proliferation. Indeed, providing cells with exogenous sources of aspartate circumvents the proliferative defects of impairments to complexes I, III, and IV without correcting the NAD+/NADH imbalance, indicating that ETC impairments block cell proliferation primarily by suppressing aspartate synthesis (*Birsoy et al., 2015*; *Sullivan et al., 2015*). Nonetheless, it remains unclear how alterations to other components of mitochondrial metabolism affect redox homeostasis and aspartate production, as well as how cells with mitochondrial dysfunction may adapt to the disruption of canonical aspartate production pathways.

The dual ETC/TCA cycle enzyme SDH serves important functions in mitochondrial metabolism yet is also a target of biological disruption. SDH is a heterotetrameric nuclear-encoded protein complex located in the inner mitochondrial membrane (IMM) that catalyzes the oxidation of succinate to fumarate and shuttles electrons to ubiquinone in the ETC. Biallelic loss-of-function mutations to the subunits of SDH (A-D) or SDH assembly factors (SDHAF1-2), can lead to neoplasms in neuroendocrine tissues and the kidney (*Astuti et al., 2001*; *Bardella et al., 2011*; *Baysal et al., 2000*; *Burnichon et al., 2010*; *Hao et al., 2009*; *Niemann and Müller, 2000*). Cancer cells arising from mutations in SDH components typically display a complete loss of oxidative TCA cycle metabolism at the SDH step and have a substantial accumulation of the upstream metabolite succinate, which can impair enzymes involved in oxygen sensing, epigenetic regulation, and metabolism (*Letouzé et al., 2013*; *Sullivan et al., 2016*). SDH loss also blocks the canonical route for aspartate biosynthesis, making it unclear how SDH-deficient cells adapt to fulfill their anabolic demands for cell proliferation. Understanding how SDH-deficient cancer cells reconfigure their metabolism to support cell proliferation will help

reveal the normal metabolic roles of SDH and could support the development of novel therapies targeting the unique metabolic liabilities of SDH-mutant cancers.

Here, we investigate the consequences of acute and chronic SDH deficiency and observe that SDH impairment causes aspartate-dependent proliferation defects. However, unlike impairments to other ETC complexes, exogenous electron acceptors are insufficient to restore aspartate levels and cell proliferation to SDH-deficient cells. Surprisingly, we find that additional impairment to CI is required to enable aspartate synthesis and cell proliferation in SDH-deficient cells. Specifically, we find that CI inhibition decreases mitochondrial NAD+/NADH, which drives pyruvate carboxylation and reductive carboxylation of glutamine-derived alpha-ketoglutarate (AKG) to generate precursors for aspartate synthesis in the cytosol, thereby supporting proliferation in SDH-impaired cells. Finally, we observe that disrupting or restoring SDH prompts selective pressure for progressive changes in CI activity to directly correspond to SDH activity, identifying distinct metabolic states that enable aspartate production. Altogether, our data reveal a metabolic mechanism where compartmentalized redox changes are required to enact alternative aspartate biosynthetic pathways that support cell proliferation during TCA cycle dysfunction.

## Results

### SDH inhibition impairs proliferation, which is partially restored by electron acceptors but robustly restored by aspartate

To understand the metabolic contributions of SDH/complex II (hereafter, SDH) to cell proliferation, we compared the proliferative consequences of the inhibitors atpenin A5 (AA5), a specific inhibitor of SDH, and the classic CI inhibitor rotenone in the respiration-intact osteosarcoma cell line 143B. Both inhibitors effectively blocked cell proliferation when cultured in media without electron acceptors (DMEM without pyruvate), consistent with an essential role of mitochondrial metabolism in support of cell proliferation (*Figure 1A*). Previous work from us and others identified that supplementation with the exogenous electron acceptors pyruvate (PYR) or alpha-ketobutyrate (AKB) can restore proliferation to cells with CI impairments by regenerating NAD +to support aspartate biosynthesis (*Birsoy et al., 2015*; *Sullivan et al., 2015*). While we confirm that PYR and AKB can robustly restore proliferation upon rotenone treatment, proliferation of AA5 treated cells was only modestly improved by electron acceptor supplementation (*Figure 1A*). In contrast, aspartate supplementation equivalently, albeit incompletely, restored cell proliferation to both rotenone and AA5-treated cells (*Figure 1A*). Aspartate is poorly permeable to most cells even at supraphysiological concentrations, so we generated 143B cells expressing the glial-specific aspartate transporter SLC1A3, which we confirm supports aspartate uptake at micromolar extracellular concentrations (*Figure 1—figure supplement 1A–C*). Notably, improved aspartate uptake with SLC1A3 expression allowed aspartate to fully restore proliferation upon AA5 treatment, regardless of PYR co-treatment (*Figure 1—figure supplement 1D*). Similarly, orthogonal aspartate acquisition by intracellular expression of the guinea pig asparaginase (gpASNase1) combined with asparagine treatment was also able to restore proliferation to AA5-treated cells (*Figure 1—figure supplement 1E–G*; *Sullivan et al., 2018*). These data indicate that aspartate is a primary metabolic limitation of SDH-impaired cells, consistent with previous findings (*Cardaci et al., 2015*; *Lussey-Lepoutre et al., 2015*). However, these data also indicate that the metabolic determinants of aspartate limitation upon SDH inhibition are distinct from those of CI loss.

To evaluate the metabolic differences between impairments to CI or SDH, we conducted liquid chromatography-mass spectrometry (LCMS) metabolomics on cells treated with either inhibitor, with or without pyruvate supplementation. As expected for SDH impairment, AA5 treatment caused succinate accumulation and depleted fumarate levels (*Figure 1—figure supplement 1H and I*). Consistent with aspartate being the mediator of proliferation defects from either inhibitor, aspartate levels were depleted by both drugs, fully restored by pyruvate in rotenone treated cells, and only partially increased in cells treated with AA5 (*Figure 1B*). Thus, although either inhibitor is associated with proliferation defects corresponding with aspartate levels, CI impairments have distinct metabolic phenotypes from SDH inhibition.

While intracellular aspartate levels correlate with proliferation rate in our datasets of mitochondrial impaired cells, the quantitative relationship between the two is non-linear making it unclear how to predict the degree to which changes in aspartate levels will constitute a functionally meaningful

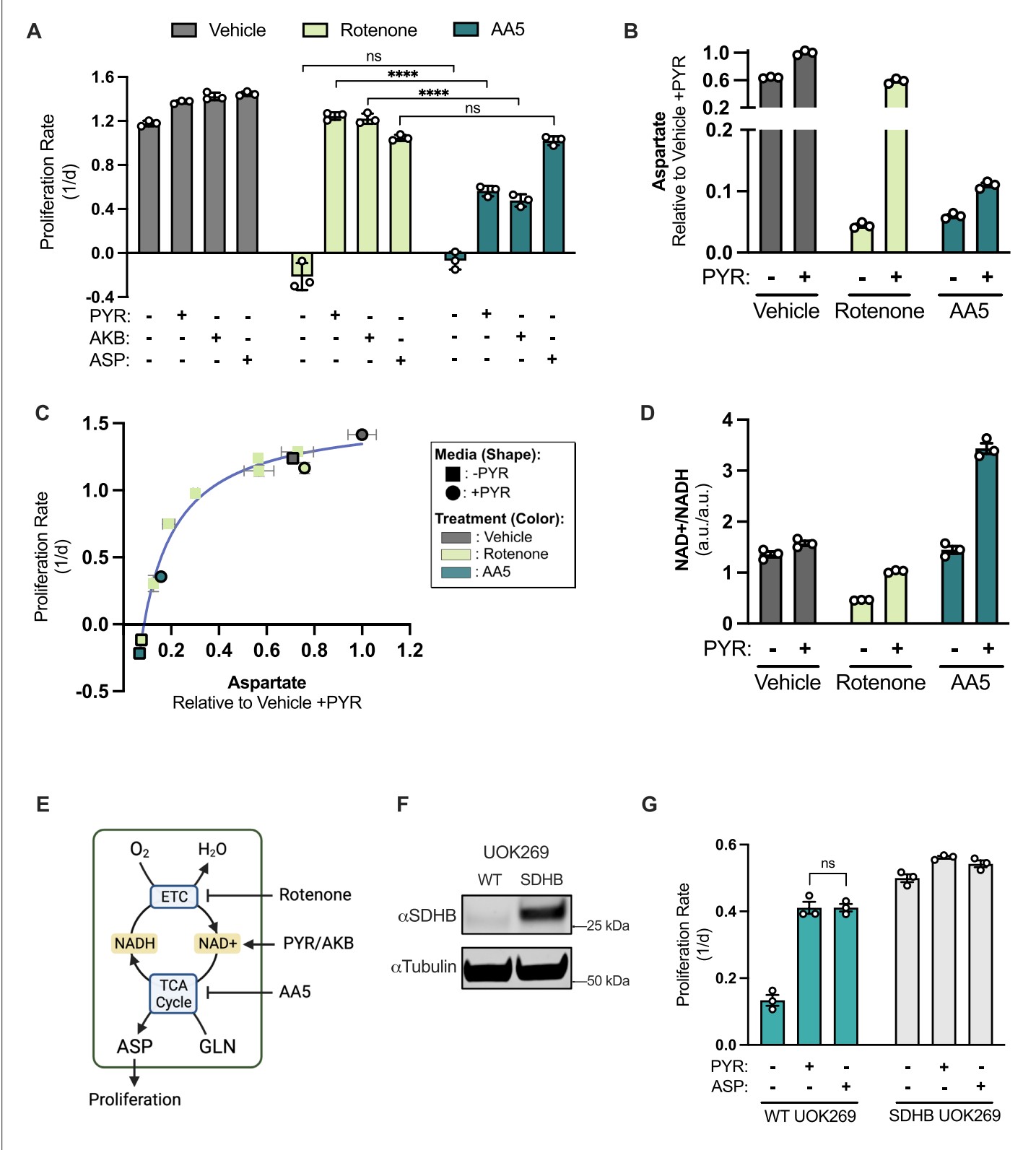

**Figure 1.** SDH inhibition blocks proliferation, which is incompletely rescued by electron acceptors but robustly restored by aspartate. (**A**) Proliferation rates of 143B cells treated with vehicle (DMSO), 50 nM rotenone, or 5 µM atpenin A5 (AA5) cultured in pyruvate free DMEM with no addition, 1 mM pyruvate (PYR), 1 mM alpha-ketobutyrate (AKB), or 20 mM aspartate (ASP). (**B**) Aspartate levels measured by liquid chromatography-mass spectrometry (LCMS) metabolomics in 143B cells, comparing normalized aspartate ion counts (a.u.; arbitrary units) from cells treated with vehicle (DMSO), 50 nM

*Figure 1 continued on next page*

*Figure 1 continued*

rotenone, or 5 µM AA5 in pyruvate free DMEM with no addition or 1 mM PYR for 6 hr (n=3). (**C**) Proliferation rate (y-axis) versus normalized aspartate levels (x-axis) of 143B cells after 6 hours of the indicated treatments (n=3). Data points without borders correspond to a dose titration of rotenone (25 12.5, 6.25, 3.125, 1.565 and 0 nM) in pyruvate free DMEM. Data points with a black border correspond to cells treated with standard dosing of 50 nM rotenone and 5 µM AA5 in DMEM, with and without pyruvate as indicated. See *Figure 1—figure supplement 1K-L* for additional information. (**D**) NAD+/NADH measured by LCMS metabolomics of 143B cells treated with vehicle (DMSO), 50 nM rotenone, or 5 µM AA5 in pyruvate free DMEM with no addition or 1 mM PYR for 6 hours (n=3). (**E**) Schematic showing that rotenone inhibits the ETC, blocking NADH oxidation causing an indirect TCA cycle impairment that can be overcome by exogenous electron acceptors (PYR/AKB), while AA5 directly blocks the TCA cycle and aspartate synthesis. (**F**) Western blot for SDHB and α-tubulin (tubulin) in WT UOK269 cells or UOK269 cells with restoration of SDHB (SDHB UOK269) as indicated. Tubulin is used as a loading control. (**G**) Proliferation of WT UOK269 and SDHB UOK269 cells cultured in pyruvate-free DMEM supplemented with vehicle (H$_2$O), 1 mM PYR, or 20 mM ASP (n=3). Data are plotted as means ± standard deviation (SD) and compared with an unpaired two-tailed student's t-test. p<0.05*, p<0.01**, p<0.001***, p<0.0001****.

The online version of this article includes the following source data and figure supplement(s) for figure 1:

**Source data 1.** Proliferation rates and relative metabolite levels by LCMS in *Figure 1*.

**Source data 2.** Uncropped western blot for *Figure 1F*.

**Figure supplement 1.** Alternative methods of aspartate acquisition and characterization of metabolic phenotypes in SDH-impaired cells.

**Figure supplement 1—source data 1.** Proliferation rates and metabolite levels by LCMS in *Figure 1—figure supplement 1*.

**Figure supplement 1—source data 2.** Uncropped western blot for *Figure 1—figure supplement 1B and F*.

change for proliferation rate (*Figure 1A and B*). To contextualize these findings, we generated a dose response curve measuring the effects of a range of rotenone concentrations on proliferation rate and intracellular aspartate levels, comparing them to SDH inhibition by AA5, with or without PYR treatment (*Figure 1—figure supplement 1K and L*). Notably, combining these datasets revealed that the correlation between proliferation rate and aspartate levels was consistent across inhibitors and environmental conditions, manifesting as a hyperbolic relationship (*Figure 1C*). These data are consistent with a model where the phenotypic consequences for changes in aspartate levels depend on the degree to which aspartate limits proliferative cell metabolism. Basal conditions are not aspartate-limited and so decreases in aspartate will only have a small effect on cell proliferation; however, as aspartate becomes metabolically limited (around 0.4, compared to vehicle treated cells in media containing PYR), changes in aspartate have more severe consequences on cell proliferation, until aspartate levels are insufficient to support proliferation (around 0.05). These data therefore provide critical context to understand how changes in aspartate levels correspond to proliferation upon mitochondrial inhibition.

We next investigated how each inhibitor affects intracellular NAD+/NADH, a critical mediator of the effects of CI inhibition on aspartate levels and cell proliferation. Unlike with rotenone treatment, AA5 treatment was not associated with suppression of NAD+/NADH, breaking the direct correlation between NAD+/NADH, aspartate levels, and proliferation rate seen with rotenone and other CI inhibitors (*Figure 1D*; *Gui et al., 2016*). Pyruvate supplementation increased NAD+/NADH in all cases; restoring NAD+/NADH in rotenone treated cells to near vehicle-treated levels and increasing NAD+/NADH beyond vehicle treated cells upon AA5 treatment (*Figure 1D*). These data are consistent with a model where CI impairments suppress aspartate production by lowering NAD+/NADH and thereby slowing the oxidative TCA cycle, whereas SDH inhibition directly blocks the oxidative TCA cycle, impairing aspartate synthesis and NADH production (*Figure 1E*). Indeed, we measured M+4 aspartate in cells cultured in U-$^{13}$C glutamine, reflective of aspartate produced via canonical oxidative TCA cycle activity and found that pyruvate treatment restores metabolic progression through the TCA cycle in rotenone-treated cells but not in cells treated with AA5 (*Figure 1—figure supplement 1J*).

We also evaluated the metabolic effects of alterations to SDH activity in UOK269 cells, a patient derived renal cell carcinoma cell line that arose from a heterozygous point mutation (R46Q) in SDHB and deletion of the other allele (*Saxena et al., 2016*). Re-expression of wild type SDHB in UOK269 cells had the expected metabolic effects by alleviating the accumulation of succinate and increasing aspartate levels (*Figure 1F*, *Figure 1—figure supplement 1M and N*). Similar to 143B cells, an intact TCA cycle ameliorated the dependence of UOK269 cells on pyruvate or aspartate for cell proliferation (*Figure 1G*). However, in contrast to AA5 treated 143B cells, pyruvate and aspartate treatment were equally effective at restoring cell proliferation in UOK269 cells (*Figure 1G*). These data suggest that

long-term SDH loss may be associated with metabolic adaptations that allow cells to fully take advantage of exogenous electron acceptors to restore aspartate and cell proliferation.

## CI inhibition promotes aspartate synthesis and cell proliferation in SDH-deficient cancer cells

We next questioned what metabolic adaptations may occur in SDH-deficient cells to promote cell proliferation. Intriguingly, several cell lines generated with genetic defects in SDH components have diminished respiration rates (*Cardaci et al., 2015*; *Lorendeau et al., 2017*; *Saxena et al., 2016*). Since CI and SDH are functionally independent contributors to the ETC, and CI-derived electrons account for the majority of respiration, it is not inherently obvious that SDH-deficient cells would be unable to maintain robust respiration activity. In support of this, we found that treatment with AA5 only partially impairs respiration after several hours of treatment, whereas rotenone severely abrogates respiration within minutes (*Figure 2—figure supplement 1A*). One potential explanation for the respiration loss in SDH-mutant cells is the observation that they manifest with decreased CI expression and activity (*Cardaci et al., 2015*; *Lorendeau et al., 2017*; *Saxena et al., 2016*). Indeed, it was reported that dual inhibition of SDH and CI is required to mimic the metabolic phenotype of SDH-mutant cancer cells (*Lorendeau et al., 2017*). Nonetheless, while these data suggest that CI loss may serve a metabolic role in SDH-deficient cells, no functional benefit for CI loss in this context has yet been described.

We tested if, in the presence of electron acceptors, rotenone could alter the proliferation of cells treated with AA5 and, surprisingly, found it caused a robust restoration of cell proliferation (*Figure 2A*). Since aspartate levels define the proliferation rate of SDH-impaired cells, we then measured aspartate levels and found that CI co-inhibition partially restored aspartate to SDH-impaired cells to a level consistent with their improved proliferation rate (*Figure 1C*, *Figure 2B*). CI co-inhibition also restored proliferation and increased aspartate levels across a panel of diverse human cell lines treated with AA5, including both transformed (A549, HCT116) and non-transformed (293T, TF-1) cells, indicating that this metabolic relationship was generalizable (*Figure 2—figure supplement 1B and C*). To test these effects genetically, we used CRISPR/Cas9 to generate gene knockouts (KO) of SDHB, the most frequently mutated subunit in SDH-null human cancers (*Amar et al., 2007*; *Badenhop et al., 2004*; *Klein et al., 2008*; *Linehan and Ricketts, 2013*). While we were able to achieve efficient disruption of SDHB, with ~96% loss in 143B cells, we were initially unable to generate single-cell SDHB KO clones in standard media (*Figure 2C*). Based on our previous observations, we repeated colony selection with or without rotenone and found that CI inhibition was sufficient for successful generation of clones from Cas9/sgSDHB nucleofected 143B cells (*Figure 2D*). Notably, every tested clone generated in this manner was devoid of SDHB expression (*Figure 2E* and *Figure 2—figure supplement 1D*). We also found that AA5 treatment of an SDHB KO clone had no significant effects on proliferation, aspartate levels or the NAD+/NADH ratio, indicating on-target effects of AA5 (*Figure 2—figure supplement 1G*). In each SDHB KO cell line generated, proliferation was slowed relative to their parental cells but was improved by treatment with the CI inhibitors rotenone or metformin, or by supplementation with aspartate (*Figure 2F*, *Figure 2—figure supplement 1H*). We also restored SDHB expression in a KO clone and found that it rescued proliferation to parental rates and reinstated wild type relationships with rotenone and AA5 treatment (*Figure 2—figure supplement 1I and J*). We then measured aspartate levels in WT and SDHB KO cells with or without rotenone treatment and found that rotenone decreases aspartate levels in WT cells but increases aspartate in SDHB KO cells, consistent with proliferation restoration (*Figure 2G*). Finally, to test if this metabolic relationship was an artifact of our cell culture media choice, which contains high levels of pyruvate, we also tested the effects of SDH inhibition in 143B cells cultured in Human Plasma-Like Media (HPLM), which closely matches the nutrient composition of human plasma (*Cantor et al., 2017*). We found that AA5 still impaired cell proliferation and that rotenone or aspartate co-treatment restored cell proliferation in HPLM (*Figure 2—figure supplement 1K*). Collectively, our data demonstrate that CI inhibition is sufficient to induce aspartate levels and cell proliferation in diverse cell lines and media conditions upon SDH disruption.

## Mitochondrial redox alterations are required for CI inhibition to rescue SDH-deficient cells

We next investigated the metabolic mechanism for how CI impairment could be providing a benefit to SDH-deficient cells. CI activity contributes to several metabolic processes, including NADH oxidation,

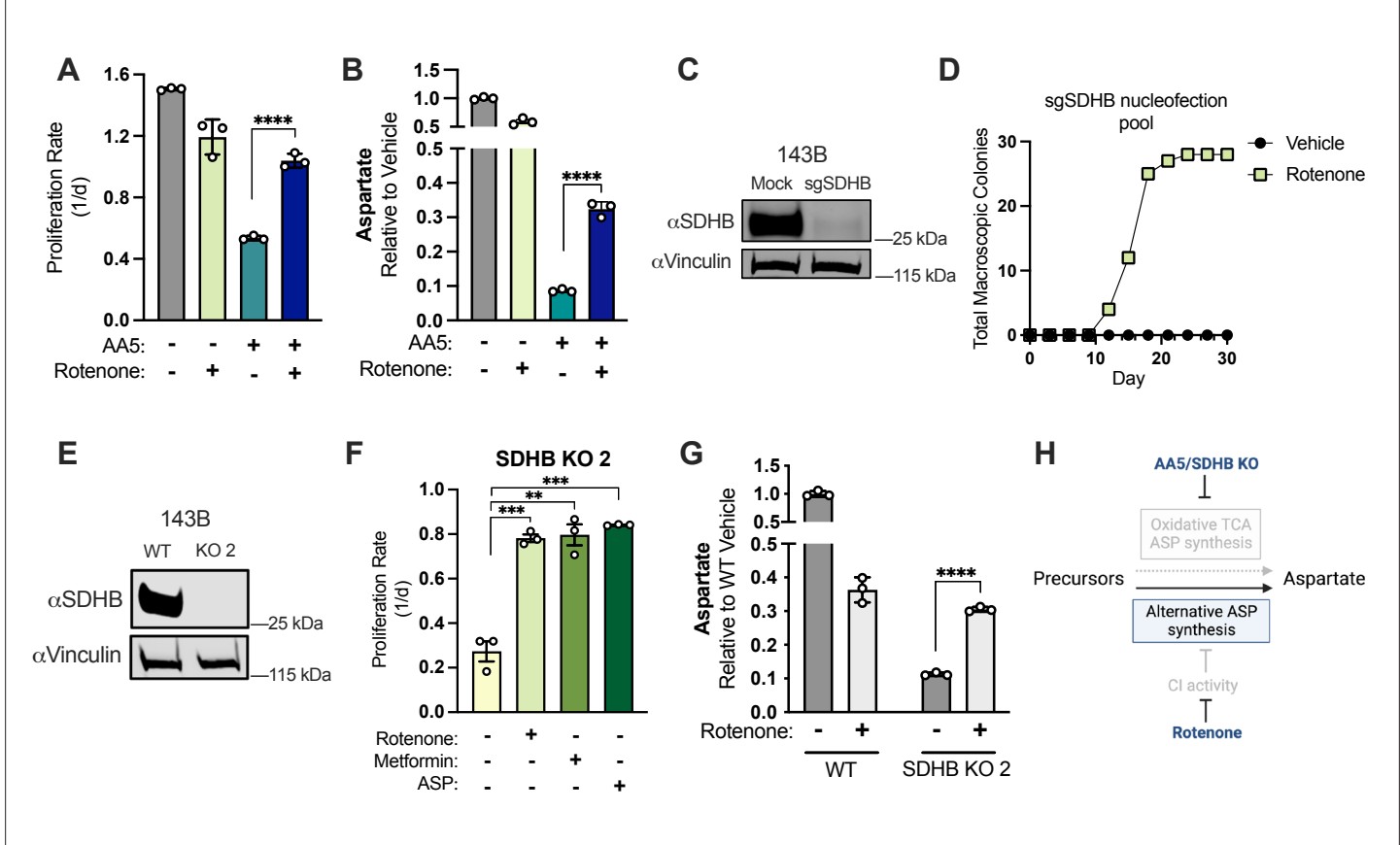

**Figure 2.** CI inhibition is sufficient to induce aspartate synthesis and cell proliferation in SDH-deficient cancer cells. (**A**) Proliferation rates of 143B cells cultured in DMEM with 1 mM PYR treated with vehicle (DMSO), 50 nM rotenone, 5 μM AA5, or 5 μM AA5 and 50 nM rotenone (n=3). (**B**) Aspartate levels of 143B cells cultured in DMEM with 1 mM PYR treated with vehicle (DMSO), 50 nM rotenone, 5 μM AA5, or 5 μM AA5 and 50 nM rotenone for 6 hours (n=3). (**C**) Western blot for SDHB and vinculin from 143B cells 3 days after nucleofection with plasmid GFP (Mock) or sgRNAs and sNLS-SpCas9 for SDHB as indicated. Vinculin is used as a loading control. (**D**) Number of single cell clones that formed colonies from sgSDHB pool in 143B cells treated with vehicle (DMSO) or 30 nM rotenone during a 30 day period. (**E**) Western blot for SDHB and vinculin from WT 143B cells and SDHB KO 143B clone 2. Vinculin is used as a loading control. (**F**) Proliferation rates of SDHB KO 143B cells (clone 2) cultured in DMEM with 1 mM PYR and treated with vehicle (DMSO), 50 nM rotenone, 1 mM metformin, or 20 mM ASP (n=3). (**G**) Aspartate levels measured by LCMS metabolomics of WT 143B cells and SDHB KO 2 cells cultured in DMEM with 1 mM PYR and treated with vehicle (DMSO) or 50 nM rotenone for 6 hr (n=3). (**H**) Schematic showing that CI inhibition promotes alternative aspartate synthesis pathways upon SDH disruption. Data are plotted as means ± standard deviation (SD) and compared with an unpaired two-tailed student's t-test. $p<0.05^*$, $p<0.01^{**}$, $p<0.001^{***}$, $p<0.0001^{****}$.

The online version of this article includes the following source data and figure supplement(s) for figure 2:

**Source data 1.** Proliferation rates, relative metabolite levels by LCMS, and other data in *Figure 2*.

**Source data 2.** Uncropped western blots for *Figure 2C and E*.

**Figure supplement 1.** Characterization of interactions between CI inhibition and SDH status.

**Figure supplement 1—source data 1.** Proliferation rates, metabolite levels by LCMS, and mitochondrial oxygen consumption values in *Figure 2— figure supplement 1*.

**Figure supplement 1—source data 2.** Uncropped western blot for *Figure 2—figure supplement 1D and I*.

mitochondrial membrane potential generation, and the reduction of ubiquinone (*Figure 3—figure supplement 1A*). To ascertain which of these processes is deleterious to SDH-deficient cells, we co-treated cells with AA5 and conditions that target other aspects of the ETC and oxidative phosphorylation. While rotenone directly inhibits CI, the complex III inhibitor Antimycin A similarly impairs NADH oxidation and mitochondrial membrane potential generation but differs from CI inhibition by preventing ubiquinol oxidation (*Figure 3—figure supplement 1A*). Oligomycin blocks the IMM complex ATP synthase, inhibiting mitochondrial membrane potential utilization, inducing hyperpolarization of the IMM, and thereby causing collateral inhibition of ETC activity. Oligomycin treatment

therefore similarly impairs NADH oxidation but differs from CI inhibitors by increasing the mitochondrial membrane potential and slowing electron flux through all three proton pumping complexes (*Figure 3—figure supplement 1A*). Lastly, hypoxia limits the terminal electron acceptor activity of complex IV, slowing upstream ETC function (*Figure 3—figure supplement 1A*). Importantly, we found that treatment with antimycin or oligomycin can substitute for CI inhibitors, albeit less effectively, to restore the proliferation and aspartate levels of AA5 treated cells (*Figure 3—figure supplement 1B and C*). Gratifyingly, since complex III inhibition collaterally impairs both complex I and II, antimycin treatment alone had near identical effects on aspartate levels and proliferation as AA5 and rotenone cotreatment (*Figure 3—figure supplement 1B and C*). Finally, culturing cells in severe hypoxia (1%) was sufficient to improve the proliferation of SDH-impaired cells, albeit to a smaller degree than CI inhibition, while physiological $O_2$ tensions (3–11%) had little effect on the proliferation of AA5 treated cells (*Figure 3—figure supplement 1D*). Collectively, these results suggest that the shared effect of each treatment on mitochondrial NAD+/NADH, rather than other downstream effects on ETC function, is likely a primary mechanism by which ETC inhibition supports aspartate production and proliferation in SDH-deficient cells.

To investigate how CI inhibition changes the redox state of SDH-impaired cells, we measured the effects of co-inhibition on NAD+/NADH by LCMS. While AA5 treatment alone increased NAD+/NADH in the presence of pyruvate, as in *Figure 1D*, the effects of rotenone co-treatment were epistatic to AA5 treatment and reduced cellular NAD+/NADH to that of cells treated with rotenone alone (*Figure 3A*). The increase in NAD+/NADH in cells upon SDH impairment alone is thus likely a consequence of decreased NADH production in the TCA cycle coinciding with persisting NADH oxidation by active mitochondrial CI. We next adapted a selective membrane permeabilization protocol to rapidly isolate and extract metabolites from the cytosol and mitochondria and used an enzymatic assay that enables low sample inputs to measure NAD+/NADH in each compartment (*Figure 3—figure supplement 1E*; *Lee et al., 2019*; *Sullivan et al., 2015*). As expected, in vehicle-treated conditions, mitochondrial NAD+/NADH was lower than cytosolic NAD+/NADH (*Figure 3B*; *Williamson et al., 1967*). In addition, the effects on NAD+/NADH from rotenone and/or AA5 treatment were predominantly observed at their site of action in the mitochondria (*Figure 3B*). Notably, the absence of changes to cytosolic NAD+/NADH upon mitochondrial inhibitor treatments is expected in this context due to exogenous pyruvate pinning cytosolic NAD+/NADH through lactate dehydrogenase activity.

We next asked if mitochondrial NAD+/NADH alterations are required for the proliferation benefits of CI inhibition in SDH-impaired cells. To do so, we expressed cytosolic or mitochondrial-targeted FLAG-tagged *Lactobacillus brevis* NADH oxidase (cyto*LbNOX*/mito*LbNOX*) in 143B cells, an enzyme that directly utilizes molecular oxygen to oxidize NADH to NAD +without requiring ETC function or other carbon substrates (*Figure 3C*; *Titov et al., 2016*). Subcellular fractionation and immunoblotting verified successful subcellular targeting, as each construct was enriched in its expected compartment (*Figure 3D*). Notably, since these cells are cultured in the presence of the cytosolic electron acceptor pyruvate, we thus expected cyto*LbNOX* to be redundant and ineffective to change cell metabolism, whereas mitoLbNOX would be positioned to disrupt the mitochondrial redox changes from rotenone treatment. Indeed, whereas cells expressing cyto*LbNOX* behaved similarly to parental cells expressing eGFP, mito*LbNOX* expression impeded the restorative effect of rotenone treatment on SDH-impaired cell proliferation (*Figure 3E*). Moreover, the increase in aspartate levels upon rotenone co-treatment was similarly diminished by mito*LbNOX* expression, but not by cyto*LbNOX* expression (*Figure 3F*). These data indicate that decreased mitochondrial NAD+/NADH is required for the benefits of CI inhibition when SDH is impaired.

We next asked if decreased mitochondrial NAD+/NADH was sufficient to improve SDH-deficient cell function without CI inhibition. Another determinant of mitochondrial NAD+/NADH is the pyruvate dehydrogenase (PDH) reaction, which converts NAD +to NADH (*Figure 3—figure supplement 2A*). The drug AZD7545 can disinhibit PDH to increase its activity by blocking regulation by pyruvate dehydrogenase kinases (PDKs) (*Morrell et al., 2003*). In the context of electron acceptor limitation, AZD7545 can further drive down NAD+/NADH and impair cell function (*Luengo et al., 2021*) however, our data indicate that lowering mitochondrial NAD+/NADH is beneficial in SDH-deficient cells, so we hypothesized that AZD7545 may instead provide a benefit in this context. To ensure that the AZD7545 mechanism of action functions independently of CI inhibition, we confirmed that AZD7545 treatment does not impair mitochondrial oxygen consumption in AA5 treated cells (*Figure 3—figure*

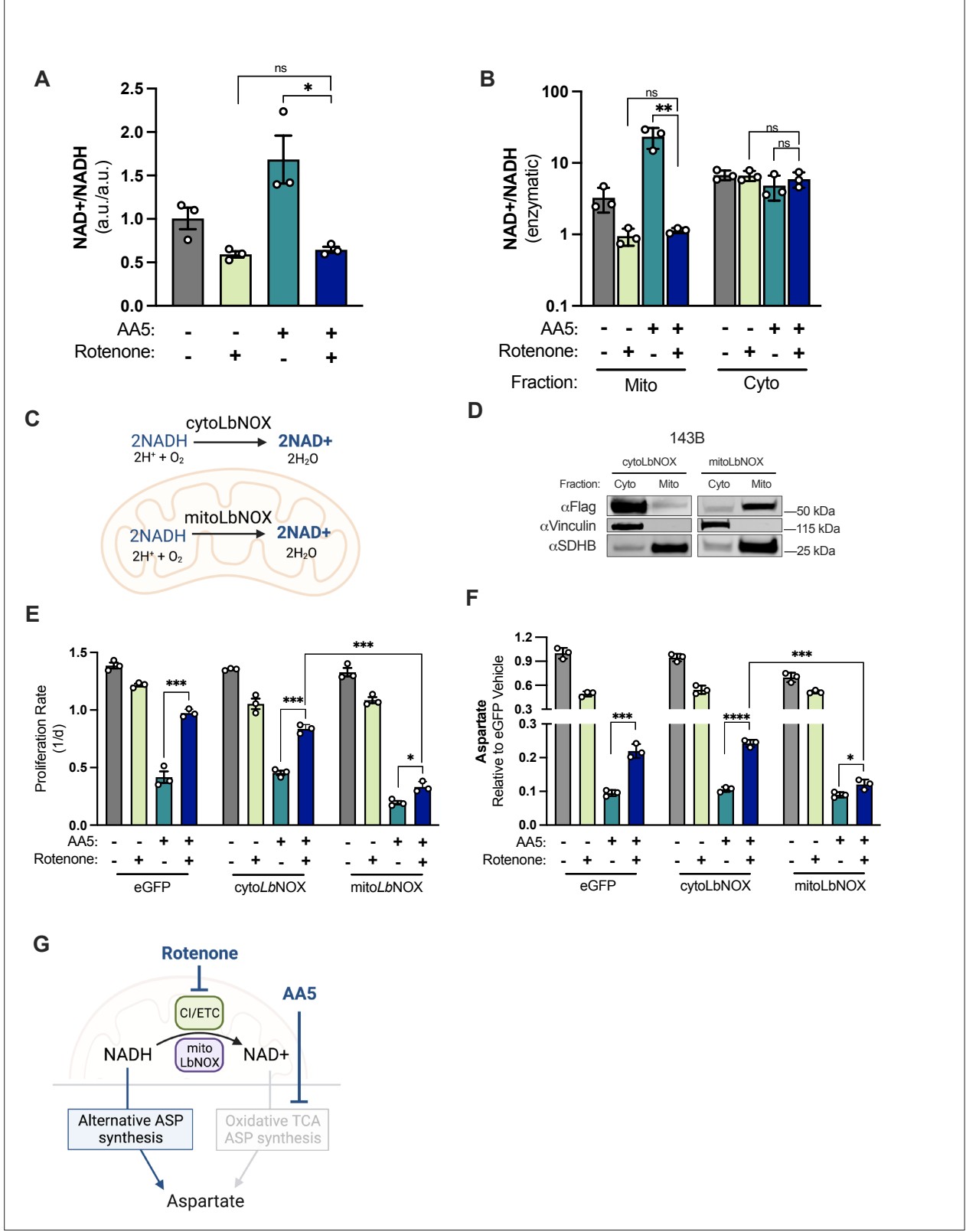

**Figure 3.** CI inhibition decreases mitochondrial NAD+/NADH, which is required for aspartate synthesis and proliferation in SDH-impaired cells. (**A**) Whole cell NAD+/NADH measured by LCMS metabolomics of 143B cells cultured in DMEM with 1 mM PYR and treated with vehicle (DMSO), 50 nM rotenone, 5 μM AA5, or 5 μM AA5 and 50 nM rotenone for 6 hr (n=3). (**B**) Cytosolic and mitochondrial NAD+/NADH measured by enzymatic assay of 143B cells cultured in DMEM with 1 mM PYR and treated with vehicle (DMSO), 50 nM rotenone 5 μM AA5, or 5 μM AA5 and 50 nM rotenone for

*Figure 3 continued on next page*

*Figure 3 continued*

6 hr (n=3). (**C**) Schematic depicting the functions of FLAG-tagged cyto*LbNOX* and mito*LbNOX* in each compartment as indicated. (**D**) Western blot for FLAG, Vinculin, and SDHB from 143B cells expressing FLAG-tagged cyto*LbNOX* or mito*LbNOX*, in cytosolic or mitochondrial fractions isolated by differential centrifugation. Vinculin is a loading control for cytosol and SDHB is a loading control for mitochondria. (**E**) Proliferation rates of eGFP, cyto*LbNOX,* and mito*LbNOX* expressing 143B cells cultured in DMEM with 1 mM PYR treated with vehicle (DMSO), 50 nM rotenone, 5 µM AA5, or 5 µM AA5 and 50 nM rotenone (n=3). (**F**) Aspartate levels measured by LCMS metabolomics of eGFP, cyto*LbNOX,* and mito*LbNOX* expressing 143B cells cultured in DMEM with 1 mM PYR and treated with vehicle (DMSO), 50 nM rotenone, 5 µM AA5, or 5 µM AA5 and 50 nM rotenone for 6 hr (n=3). (**G**) Schematic showing how compartment-specific mitochondrial redox alterations promote alternative aspartate synthesis pathways in SDH-impaired cells. Data are plotted as means ± standard deviation (SD) with the exception of A which is plotted as means ± standard error of the mean (SEM) and compared with an unpaired two-tailed student's t-test. $p < 0.05$*, $p < 0.01$**, $p < 0.001$***, $p < 0.0001$****.

The online version of this article includes the following source data and figure supplement(s) for figure 3:

**Source data 1.** Proliferation rates, relative metabolite levels by LCMS, and enzymatic assay values in *Figure 3*.

**Source data 2.** Uncropped western blot for *Figure 3D*.

**Figure supplement 1.** Effects of ETC inhibition in SDH-deficient cells.

**Figure supplement 1—source data 1.** Proliferation rates and metabolite levels by LCMS in *Figure 3—figure supplement 1*.

**Figure supplement 2.** Effects of PDK inhibition in SDH-deficient cells.

**Figure supplement 2—source data 1.** Proliferation rates, relative metabolite levels by LCMS, and mitochondrial oxygen consumption values in *Figure 3—figure supplement 2*.

---

*supplement 2B*). Additionally, AA5 treated cells co-treated with AZD7545 maintained and marginally increased [13]C glucose-derived cis-aconitate levels, whereas rotenone co-treatment depleted them, highlighting the distinct mechanisms by which these inhibitors drive mitochondrial metabolic changes (*Figure 3—figure supplement 2C*). Importantly, treatment with AZD7545 also slightly improved cell proliferation and increased aspartate levels in AA5-treated cells (*Figure 3—figure supplement 2D and E*). Notably, these effects are likely limited by the fact that they are occurring in the presence of intact CI activity, which would likely buffer major changes in mitochondrial NAD+/NADH. Nonetheless, these data provide additional support that compartment-specific mitochondrial redox alterations are both sufficient and required to reprogram aspartate synthesis in SDH-impaired cells (*Figure 3G*).

## Mitochondrial redox alterations promote alternative aspartate synthesis pathways in SDH-impaired cells

While our results demonstrate that decreased mitochondrial NAD+/NADH is a critical driver of aspartate synthesis in SDH-impaired cells, it remains unclear how these redox changes induce aspartate production. One likely contributor is reductive carboxylation of glutamine-derived AKG (RCQ), whose relative contribution to aspartate production is increased in cells with defects in oxidative mitochondrial metabolism (*Metallo et al., 2011*; *Mullen et al., 2011*, *Mullen et al., 2014*; *Wise et al., 2011*). Another is pyruvate carboxylase (PC), which can have increased expression and activity in SDH-null cancer cells (*Cardaci et al., 2015*; *Lussey-Lepoutre et al., 2015*). To ascertain the source of increased aspartate in SDH-deficient cells upon CI inhibition we conducted stable isotope labeling of WT and low passage SDHB KO cells with or without rotenone from U-[13]C glutamine, which can delineate between aspartate generated from the oxidative TCA cycle or RCQ (*Figure 4A*). SDHB KO decreased total aspartate levels and abolished oxidative M+4 aspartate production, as expected, and RCQ-derived M+3 aspartate was slightly increased in SDHB KO cells upon rotenone treatment (*Figure 4A and B*). While RCQ can occur in both the cytoplasmic and mitochondrial compartments through IDH1 and IDH2, respectively, both RCQ routes rely on ATP-citrate lyase (ACLY) to cleave cytosolic citrate to oxaloacetate (OAA) and acetyl-CoA (*Figure 4—figure supplement 1A*). Therefore, we tested if preventing RCQ with the ACLY inhibitor BMS-303141 (BMS) would alter the proliferation rescue of CI inhibition in SDH-impaired 143B cells. Indeed, BMS had an anti-proliferative effect on AA5 and rotenone co-treated cells, which could be prevented by aspartate supplementation (*Figure 4C*). Correspondingly, aspartate levels were decreased in SDH/CI inhibited cells when treated with BMS, tracking with the small contribution of RCQ to aspartate production in this context (*Figure 4B and D*). These data indicate that RCQ is partially responsible for increased aspartate synthesis and cell proliferation upon CI inhibition in SDH-impaired cells.

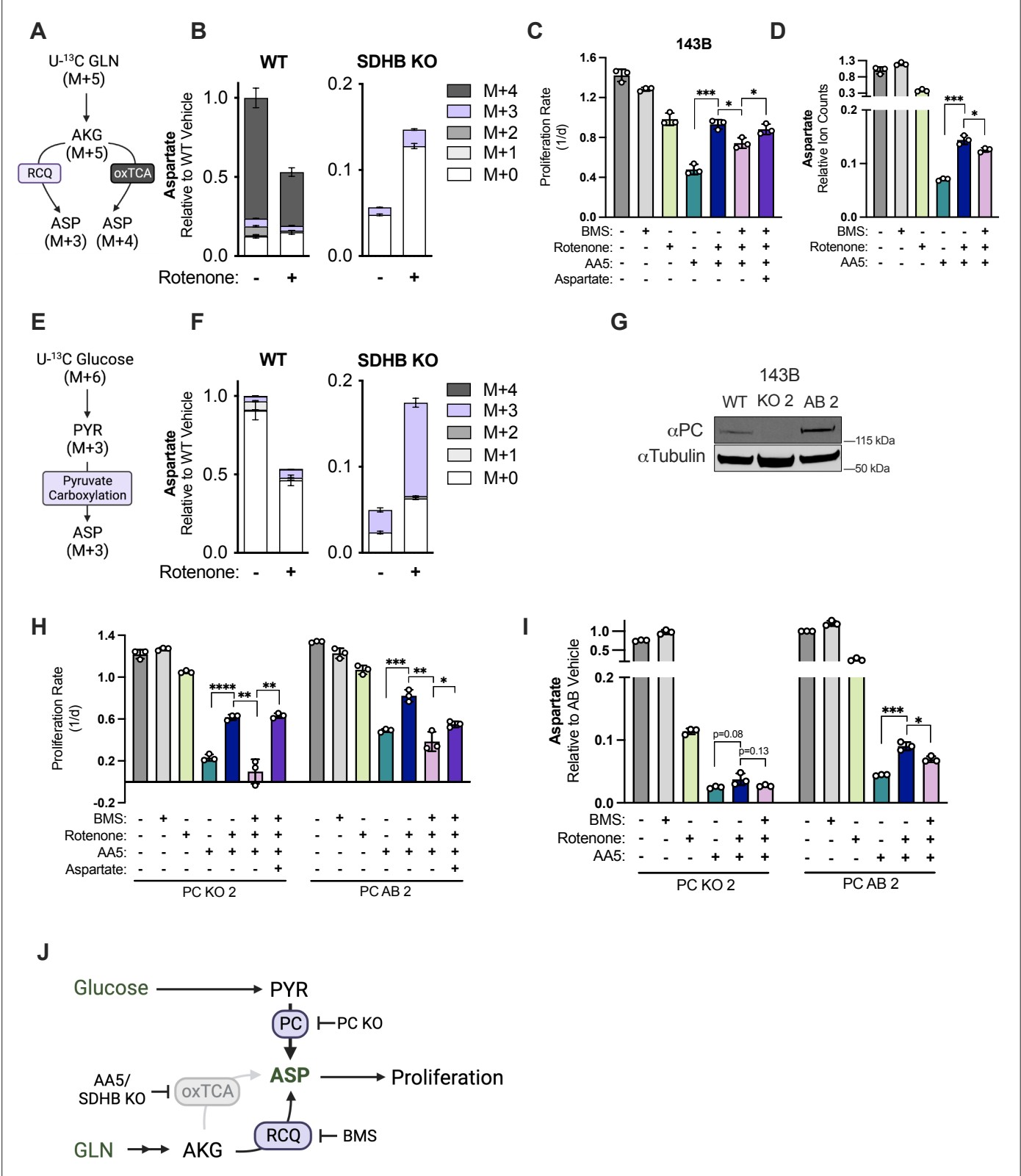

**Figure 4.** Reductive carboxylation and pyruvate carboxylation drive aspartate synthesis in SDH impaired cells upon CI inhibition. (**A**) Schematic depicting metabolic pathway usage for isotopologue patterns of aspartate derived from U-$^{13}$C glutamine. (**B**) Relative ion counts for all aspartate isotopologues derived from U-$^{13}$C glutamine measured by LCMS metabolomics in WT and SDHB KO 143B cells treated with either vehicle (DMSO) or 50 nM rotenone in pyruvate-free DMEM with 1 mM AKB for 6 hr (n=3). (**C**) Proliferation rates of WT 143B cells treated with either vehicle (DMSO);

*Figure 4 continued on next page*

*Figure 4 continued*

25 µM BMS-303141 (BMS); 50 nM rotenone; 5 µM AA5; 5 µM AA5 and 50 nM rotenone; 5 µM AA5, 50 nM rotenone and 25 µM BMS; or 5 µM AA5, 50 nM rotenone, 25 µM BMS, and 20 mM aspartate in DMEM with 1 mM pyruvate (n=3). (**D**) Relative aspartate levels measured by LCMS metabolomics of WT 143B cells treated with either vehicle (DMSO); 25 µM BMS; 50 nM rotenone; 5 µM AA5; 5 µM AA5 and 50 nM rotenone; 5 µM AA5, 50 nM rotenone and 25 µM BMS in pyruvate free DMEM with 1 mM AKB for 6 hr (n=3). (**E**) Schematic depicting metabolic pathway usage for isotopologue patterns of aspartate derived from U-$^{13}$C glucose. (**F**) Relative ion counts for aspartate isotopologues derived from U-$^{13}$C glucose measured by LCMS metabolomics in WT and SDHB KO 143B cells treated with either vehicle (DMSO) or 50 nM rotenone in pyruvate free DMEM with 1 mM AKB for 6 hr (n=3). (**G**) Western blot for PC in WT 143B cells, PC KO clone 2, and PC KO clone 2 with PC cDNA added back. Tubulin is used as a loading control. (**H**) Proliferation rates of PC KO and PC AB 143B cells treated with vehicle (DMSO); 25 µM BMS; 50 nM rotenone; 5 µM AA5; 5 µM AA5 and 50 nM rotenone; 5 µM AA5, 50 nM rotenone and 25 µM BMS; or 5 µM AA5, 50 nM rotenone, 25 µM BMS, and 20 mM aspartate in DMEM with 1 mM PYR (n=3). (**I**) Relative aspartate levels measured by LCMS metabolomics of PC KO and PC AB 143B cells treated with vehicle (DMSO); 25 µM BMS; 50 nM rotenone; 5 µM AA5; 5 µM AA5 and 50 nM rotenone; 5 µM AA5, 50 nM rotenone and 25 µM BMS; or 5 µM AA5, 50 nM rotenone, 25 µM BMS in pyruvate free DMEM with 1 mM AKB for 6 hr (n=3). (**J**) Schematic showing two alternative aspartate synthesis pathways induced by CI inhibition in SDH-impaired cells. Data are plotted as means ± standard deviation (SD) and compared with an unpaired two-tailed student's t-test. p<0.05*, p<0.01**, p<0.001***, p<0.0001****.

The online version of this article includes the following source data and figure supplement(s) for figure 4:

**Source data 1.** Proliferation rates and relative metabolite levels by LCMS in *Figure 4*.

**Source data 2.** Uncropped western blot for *Figure 4G*.

**Figure supplement 1.** Effects of pyruvate versus AKB in SDH and CI impaired cells.

**Figure supplement 1—source data 1.** Proliferation rates and relative metabolite levels by LCMS in *Figure 4—figure supplement 1*.

Next, we measured the contributions of PC activity by culturing WT and SDHB KO cells with U-$^{13}$C glucose, with or without rotenone treatment (*Figure 4E*). To avoid potential caveats that may confound labeling patterns from unlabeled media pyruvate, we conducted U-$^{13}$C glucose isotope tracing experiments in pyruvate-free DMEM with AKB, which serves a similar electron acceptor function as pyruvate but cannot fulfill its carbon fates (*Figure 1A*; *Altea-Manzano et al., 2022*; *Sullivan et al., 2015*). We also confirmed that rotenone is still effective to restore proliferation and aspartate levels of SDH-impaired cells in pyruvate-free DMEM with AKB (*Figure 4—figure supplement 1B and C*). While glucose-derived carbon was a minor contributor to aspartate in WT cells, the majority of aspartate in vehicle-treated SDHB KO cells was labeled M+3 from glucose, as was most of the aspartate induced by rotenone treatment (*Figure 4F*). We next investigated the functional contributions of PC to SDH-impaired cells, with or without CI co-inhibition. First, we generated monoclonal PC KO 143B cells, and to control for variation from single cell cloning, we re-expressed PC cDNA in a KO clone to generate PC 'addback' (AB) cells to serve as a metabolically wild type control (*Figure 4G*). Consistent with previously published results, we find that PC KO cells are sensitive to SDH impairment (*Figure 4H*; *Cardaci et al., 2015*; *Lussey-Lepoutre et al., 2015*). Interestingly, rotenone treatment still provided a proliferative benefit to AA5-treated PC KO cells, although the resultant proliferation rate was still decreased compared to PC AB cells (*Figure 4H*). We hypothesized that this residual proliferation benefit was driven by existing or compensatory RCQ activity, so we added BMS to SDH/CI inhibited PC KO cells, which was sufficient to completely block cell proliferation and was ameliorated by aspartate treatment (*Figure 4H*). Consistent with the proliferation defects, BMS treatment also blocked the increase in aspartate levels upon rotenone treatment in SDH-impaired PC KO cells (*Figure 4I*). Collectively, these data demonstrate that both alternative aspartate synthesis pathways, RCQ and PC, contribute to aspartate levels and cell proliferation during SDH deficiency and are activated by CI impairment (*Figure 4J*).

## Mitochondrial pyruvate entry is essential for alternative aspartate synthesis pathways

Both PC and RCQ have reactions that can occur in the cytosol or mitochondria, obfuscating the mechanism by which mitochondrial redox changes from CI inhibition might drive alternative aspartate synthesis pathways in SDH-impaired cells. To gain insight into these compartmentalized factors, we tested the dependence on mitochondrial pyruvate availability by generating knockout and addback clones for MPC1, a required subunit of the mitochondrial pyruvate carrier (*Figure 5A*). Proliferation of MPC1 KO cells was moderately decreased in vehicle-treated conditions, but completely ablated by SDH impairment and no longer improved by CI co-inhibition, while MPC1 AB control cells mirrored WT cells (*Figure 5B*). Consistent with MPC activity supporting the alternative aspartate

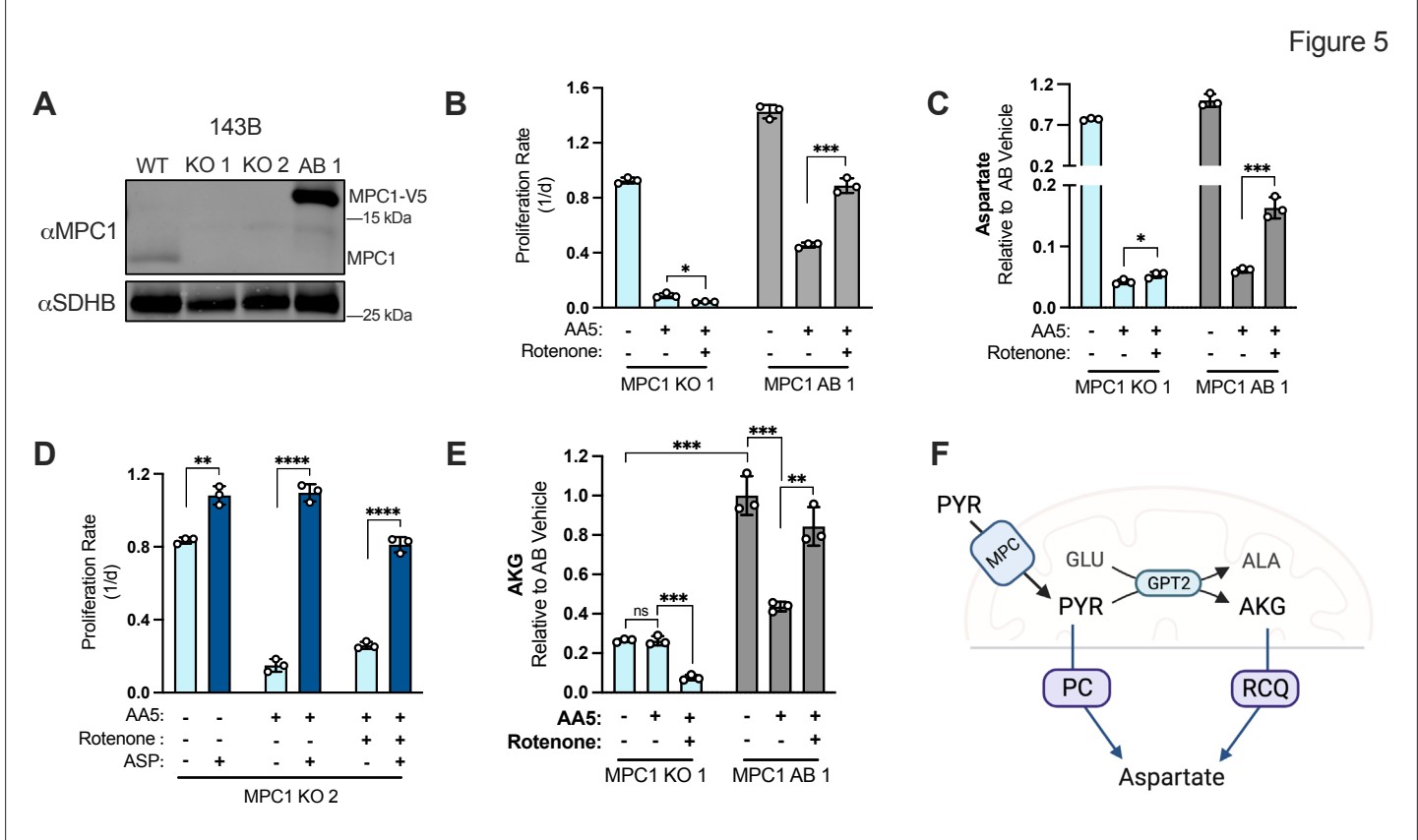

**Figure 5.** Mitochondrial pyruvate import supports alternative aspartate synthesis in SDH-impaired cells. (**A**) Western blot for MPC1 and SDHB in WT 143B cells, two MPC1 KO 143B clones, and MPC1 KO 143B clone 1 with MPC1-V5 cDNA added back (AB). SDHB is used as a loading control. (**B**) Proliferation rates of MPC1 KO 143B cells (clone 1) compared to MPC1 AB 143B cells treated with vehicle (DMSO), 5 µM AA5, or 5 µM AA5 and 50 nM rotenone in DMEM with 1 mM PYR (n=3). (**C**) Relative aspartate levels measured by LCMS metabolomics of MPC1 KO 143B cells (clone 1) compared to MPC1 AB 143B cells treated with vehicle (DMSO), 5 µM AA5, or 5 µM AA5 and 50 nM rotenone in pyruvate free DMEM with 1 mM AKB for 6 hr (n=3). (**D**) Proliferation rates of MPC1 KO 143B cells (clone 2) treated with vehicle (DMSO), 5 µM AA5, 5 µM AA5 and 50 nM rotenone, with and without 20 mM aspartate in DMEM with 1 mM PYR (n=3). (**E**) Relative AKG levels measured by LCMS metabolomics of MPC1 KO 143B cells (clone 1) compared to MPC1 AB 143B cells treated with vehicle (DMSO), 5 µM AA5, or 5 µM AA5 and 50 nM rotenone in pyruvate free DMEM with 1 mM AKB for 6 hours (n=3). (**F**) Schematic depicting the metabolic fates of mitochondrial pyruvate in SDH/CI impaired cells: PC activity to synthesize oxaloacetate and GPT2 activity to generate AKG and alanine as a byproduct. Data are plotted as means ± standard deviation (SD) and compared with an unpaired two-tailed student's t-test. $p<0.05^*$, $p<0.01^{**}$, $p<0.001^{***}$, $p<0.0001^{****}$.

The online version of this article includes the following source data and figure supplement(s) for figure 5:

**Source data 1.** Proliferation rates and relative metabolite levels by LCMS in *Figure 5*.

**Source data 2.** Uncropped western blot for *Figure 5A*.

**Figure supplement 1.** Mitochondrial AKG production is dependent on GPT2.

**Figure supplement 1—source data 1.** Relative metabolite levels by LCMS in *Figure 5—figure supplement 1*.

synthesis pathways induced by CI inhibition in SDH-impaired cells, we found that aspartate levels were decreased by AA5 treatment in MPC1 KO cells and were not restored during rotenone cotreatment (*Figure 5C*). To confirm that the proliferation defects in SDH-impaired MPC1 KO cells were mediated by aspartate depletion, we supplemented MPC1 KO cells with aspartate, and observed robust proliferation rescues during both SDH inhibition and SDH/CI co-inhibition (*Figure 5D*). Thus, mitochondrial pyruvate import is a critical component of alternative aspartate synthesis pathways in SDH-impaired cells.

Our data suggest that MPC1 KO phenocopies the proliferative and aspartate production consequences of impairing PC and RCQ in SDH-deficient cells, raising the question of how mitochondrial pyruvate entry supports these processes. While it is evident that mitochondrial pyruvate supports PC at least in part by serving as an enzymatic substrate, the requirement of mitochondrial pyruvate

for RCQ is less clear. RCQ proceeds by converting glutamine derived glutamate to AKG, which is reductively carboxylated by IDH1 or IDH2, steps toward generating cytosolic citrate that is cleaved to OAA for aspartate synthesis (*Figure 4—figure supplement 1A*). We noted that MPC1 KO was associated with a depletion in AKG, suggesting that mitochondrial pyruvate entry may support RCQ by driving AKG generation (*Figure 5E*). Glutamate conversion to AKG can conventionally occur via three mitochondrial enzymes: GDH, which consumes glutamate and NAD +to generate AKG, NADH, and ammonia; GOT2, which converts glutamate and OAA to aspartate and AKG; and GPT2, which converts glutamate and pyruvate to AKG and alanine. The requirement for MPC1 to maintain AKG levels suggests that pyruvate-dependent GPT2 activity is a dominant route for AKG production in these cells, consistent with recent studies (*Figure 5—figure supplement 1A*; *Rossiter et al., 2021*; *Wei et al., 2022*; *Weinberg et al., 2010*). Indeed, MPC1 KO was also associated with a decrease in alanine levels, corroborating studies that identify GPT2 as a dominant source of alanine production (*Figure 5—figure supplement 1B*; *Rossiter et al., 2021*). While AA5 treatment did not alter AKG levels in MPC1 KO cells, the addition of rotenone co-treatment was associated with further depletion of AKG, but not alanine, suggesting that rotenone blocked the secondary path of AKG generation through redox changes that impair the NAD +consuming GDH reaction (*Figure 5E*, *Figure 5—figure supplement 1B*). Together, these data support the hypothesis that a reduced mitochondrial environment promotes alternative aspartate synthesis in SDH-deficient cells through activating RCQ and PC activity, both of which require mitochondrial pyruvate import (*Figure 5F*).

## Alternative aspartate synthesis upon CI inhibition in SDH-impaired cells is dependent on GOT1

Since alternative aspartate synthesis pathways occur across intracellular compartments, it is unclear whether aspartate transamination occurs in SDH-impaired cells via the cytosolic glutamic-oxaloacetic transaminase (GOT) isoform GOT1 or its mitochondrial counterpart GOT2. On one hand, the mitochondrial enzymes MPC and PC are required for aspartate synthesis in this context, indicating that aspartate production by GOT2 is a plausible path. Indeed, GOT2-mediated aspartate production is observed in wild-type cells using the oxidative TCA cycle for aspartate synthesis and in hypoxic pancreatic cancer cells (*Garcia-Bermudez et al., 2022*). However, RCQ derived OAA is produced in the cytosol, suggesting GOT1 may also drive aspartate synthesis, as has been described in cells with other mitochondrial disruptions (*Birsoy et al., 2015*). Therefore, we next asked whether CI inhibition drives aspartate production in SDH-impaired cells by GOT2 or GOT1. We first generated monoclonal 143B cells with GOT2 KO and re-expressed GOT2 to generate a paired GOT2 AB cell line (*Figure 6A*). Consistent with published phenotypes, GOT2 KO cells were unable to proliferate in pyruvate-free media (*Figure 6—figure supplement 1A*; *Garcia-Bermudez et al., 2022*; *Kerk et al., 2022*). In our standard media containing pyruvate, we found that GOT2 KO cells have increased sensitivity to SDH inhibition alone but maintained the benefits of CI co-inhibition on proliferation rate and aspartate production, similar to GOT2 AB and WT cells (*Figure 6B and C*). The enhanced depletion of aspartate upon AA5 treatment in GOT2 KO cells was also functionally related to its stronger proliferation impairment phenotype, as aspartate supplementation was able to phenocopy the proliferation benefits of rotenone co-treatment (*Figure 6D*). These data indicate that residual aspartate production in SDH-impaired cells is primarily driven by GOT2, but GOT2 is dispensable for the induction of aspartate production in SDH-impaired cells upon CI inhibition.

We next established monoclonal 143B cells with GOT1 KO and re-expressed GOT1 to generate a paired GOT1 AB cell line (*Figure 6E*). GOT1 KO cells had a mild proliferation defect compared to GOT1 AB or WT cells, but unlike GOT2 KO cells they were not obviously sensitized to AA5 treatment (*Figure 6F*). Importantly, SDH-impaired GOT1 KO cells differed from GOT1 AB and WT cells in that CI inhibition no longer provided a proliferative benefit (*Figure 6F*). Correspondingly, aspartate levels were suppressed by SDH impairment in both GOT1 KO and GOT1 AB cell lines, but CI co-inhibition only increased aspartate levels in GOT1 replete cells (*Figure 6G*). Notably, CI inhibition still increased malate levels in SDH-impaired GOT1 KO cells, suggesting that the mitochondrial redox changes from CI inhibition still drive the biochemical pathways required for alternative aspartate synthesis in these cells (i.e. PC and RCQ), but that GOT1 is required for terminal aspartate generation (*Figure 6—figure supplement 1B*). These data also suggest that cytosolic malate dehydrogenase (MDH1) may play a role in alternative aspartate synthesis upstream of GOT1, so we also generated monoclonal MDH1

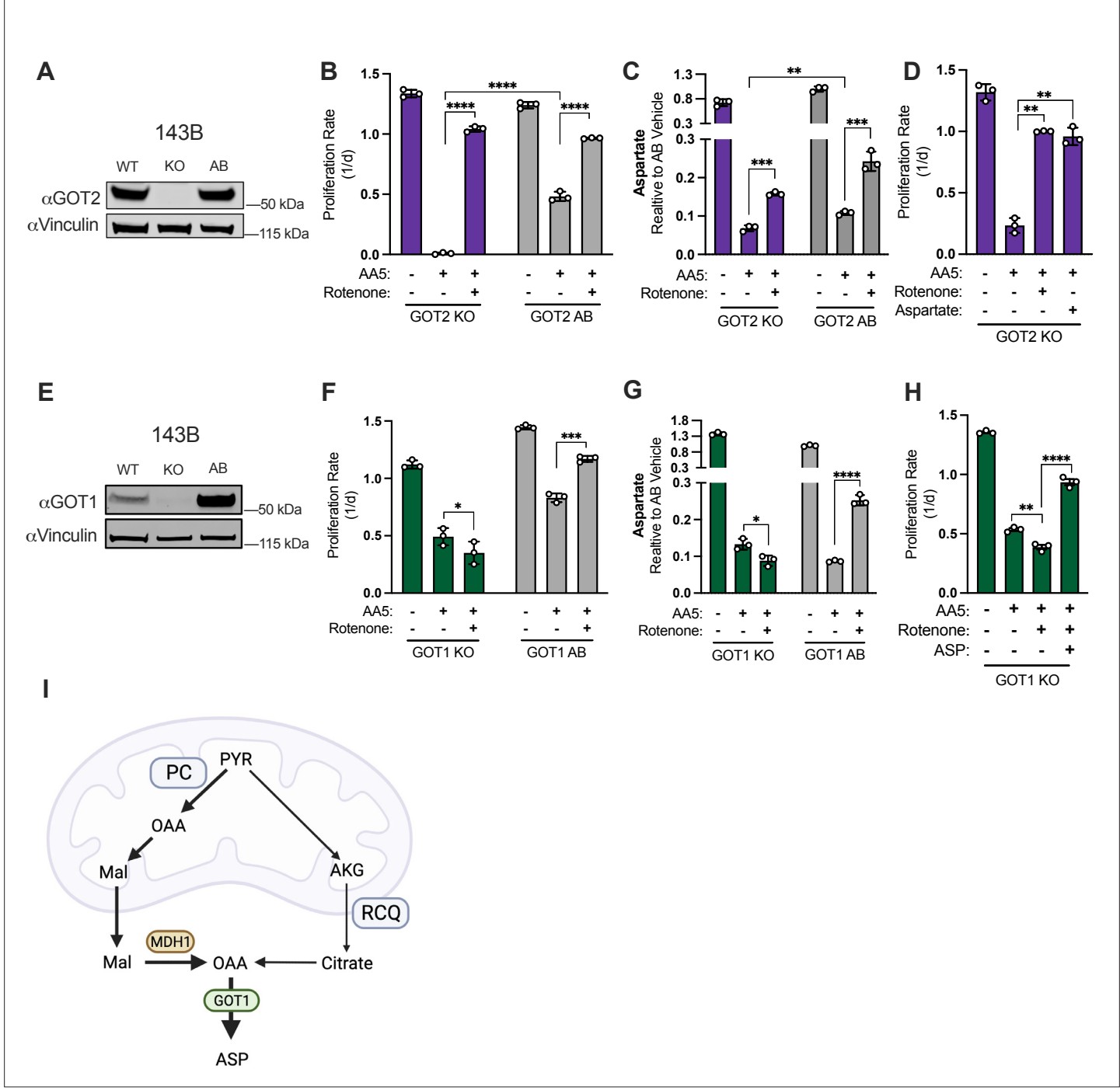

**Figure 6.** GOT1 is required for increased aspartate synthesis in SDH impaired cells upon complex I co-inhibition. (**A**) Western blot for GOT2 and Vinculin from WT 143B cells, GOT2 KO 143B cells, and the same GOT2 KO clone expressing GOT2 cDNA (AB). Vinculin is the loading control. (**B**) Proliferation rates of GOT2 KO 143B cells compared to GOT2 AB 143B cells treated with vehicle (DMSO), 5 μM AA5, or 5 μM AA5 and 50 nM rotenone in DMEM with 1 mM PYR (n=3). (**C**) Relative aspartate levels measured by LCMS metabolomics of GOT2 KO 143B cells compared to GOT2 AB 143B cells treated with vehicle (DMSO), 5 μM AA5, or 5 μM AA5 and 50 nM rotenone in pyruvate-free DMEM with 1 mM AKB for 6 hours (n=3). (**D**) Proliferation rates of GOT2 KO 143B cells treated with vehicle (DMSO), 5 μM AA5, 5 μM AA5 and 50 nM rotenone, or 5 μM AA5 and 20 mM aspartate in DMEM with 1 mM PYR (n=3). (**E**) Western blot for GOT1 and Vinculin from WT 143B cells, GOT1 KO 143B cells, and the same GOT1 KO clone expressing GOT1 cDNA (AB). Vinculin is the loading control. (**F**) Proliferation rates of GOT1 KO 143B cells compared to GOT1 AB 143B cells treated with vehicle (DMSO), 5 μM AA5, or 5 μM AA5 and 50 nM rotenone in DMEM with 1 mM pyruvate (n=3). (**G**) Relative aspartate levels measured by LCMS metabolomics of GOT1 KO 143B cells compared to GOT1 AB 143B cells treated with vehicle (DMSO), 5 μM AA5, or 5 μM AA5 and 50 nM rotenone in pyruvate-free DMEM with 1 mM AKB for 6 hr (n=3). (**H**) Proliferation rates of GOT1 KO 143B cells treated with vehicle (DMSO), 5 μM AA5, 5 μM AA5 and 50 nM

*Figure 6 continued on next page*

*Figure 6 continued*

rotenone, or 5 µM AA5, 50 nM rotenone and 20 mM aspartate in DMEM with 1 mM PYR (n=3). (**I**) Schematic showing how aspartate is synthesized via GOT1 from RCQ and PC in SDH-deficient cells when complex I is also impaired. Data are plotted as means ± standard deviation (SD) and compared with an unpaired two-tailed student's t-test. p<0.05*, p<0.01**, p<0.001***, p<0.0001****.

The online version of this article includes the following source data and figure supplement(s) for figure 6:

**Source data 1.** Proliferation rates and relative metabolite levels by LCMS in *Figure 6*.

**Source data 2.** Uncropped western blots for *Figure 6A and E*.

**Figure supplement 1.** Contributions of components of the malate-aspartate shuttle to alternative aspartate synthesis in SDH-deficient cells.

**Figure supplement 1—source data 1.** Proliferation rates and relative metabolite levels by LCMS in *Figure 6—figure supplement 1*.

**Figure supplement 1—source data 2.** Uncropped western blot for *Figure 6—figure supplement 1D*.

KO and paired MDH1 AB cells (*Figure 6—figure supplement 1C and D*). Similar to GOT1 KO cells, we found that neither proliferation rate nor aspartate levels were restored by CI inhibition in SDH-impaired MDH1 KO cells (*Figure 6—figure supplement 1E and F*). Interestingly, baseline aspartate levels were elevated in both GOT1 KO and MDH1 KO cells, consistent with their conventional aspartate-consuming role in the malate-aspartate shuttle (*Figure 6G*, *Figure 6—figure supplement 1F*). However, the observation that GOT1 KO or MDH1 KO prevents CI inhibition from restoring aspartate levels in SDH-impaired cells indicates a reversal of the malate-aspartate shuttle to synthesize aspartate in SDH/CI impaired cells (*Figure 6—figure supplement 1C*). Finally, we found that the inability of CI inhibition to restore the proliferation of SDH-impaired GOT1 KO cells was due to its lack of aspartate restoration, as aspartate rescues proliferation of SDH/CI impaired GOT1 KO cells (*Figure 6H*). The effects of each metabolic gene disruption on cell proliferation during SDH or SDH/CI inhibition were also replicated across additional KO clones (*Figure 6—figure supplement 1G*). Collectively, these results indicate that in SDH-impaired cells, CI inhibition increases mitochondrial NADH to drive alternative aspartate synthesis pathways via PC and RCQ, both of which yield cytosolic oxaloacetate that is converted to aspartate by GOT1 (*Figure 6I*). Notably, in the case of PC-derived aspartate, the divergent NAD+/NADH ratios in each compartment would support reversal of mitochondrial malate dehydrogenase (MDH2) to reduce OAA to malate, which can be exported to the cytosol and oxidized to OAA by MDH1 prior to transamination to aspartate by GOT1 (*Figure 6I*, *Figure 6—figure supplement 1C*).

## SDHB-null cells adapt through progressive complex I loss

While conducting these studies, we also generated a monoclonal SDHB KO HEK293T cell line, which, as expected, had a proliferation defect that was ameliorated by CI inhibition (*Figure 7—figure supplement 1A and B*). Interestingly, after several months of passaging, we observed that the basal proliferation rate of SDHB KO HEK293T cells progressively increased, and that the proliferation benefit of CI inhibition was lost (*Figure 7—figure supplement 1B*). To understand this phenomenon, we created three categories of cell lines derived from one 143B SDHB KO clone: Early Passage (EP) SDHB KO,<5 passages; Late Passage (LP) SDHB KO,>15 passages; or Addback (AB), where SDHB was re-expressed immediately after SDHB KO clone generation (*Figure 7A*). We measured proliferation rates of parental cells (WT), AB, EP SDHB KO and LP SDHB KO cells and found that SDHB KO decreased their proliferation rate, but that culturing SDHB KO cells to LP allowed them to adapt to an improved proliferation rate (*Figure 7B*). Culturing EP SDHB KO cells in the presence of rotenone or aspartate was able to improve cell proliferation, consistent with the effects of acute SDH inhibition with AA5; however, those benefits were largely lost in LP SDHB KO cells (*Figure 7B*). Consistent with their proliferation rates, aspartate levels were lowest in EP SDHB KO cells and were increased by either rotenone treatment or by culturing to LP (*Figure 7C*). Also similar to the acute effects of AA5 treatment, EP SDHB KO cells had an increased NAD+/NADH ratio that was diminished upon rotenone treatment; however, the NAD+/NADH ratio was already lowered in LP SDHB KO cells and unaffected by rotenone, suggesting redox adaptations occurred during passaging (*Figure 7D*). To determine if these adaptive phenotypes were due to the selective proliferation advantages they provide or simply inherent to passaged SDH-null cells, we separately passaged EP SDHB KO cells in rotenone for 15 passages (*Figure 7—figure supplement 1C*). We found that cells cultured with continuous rotenone

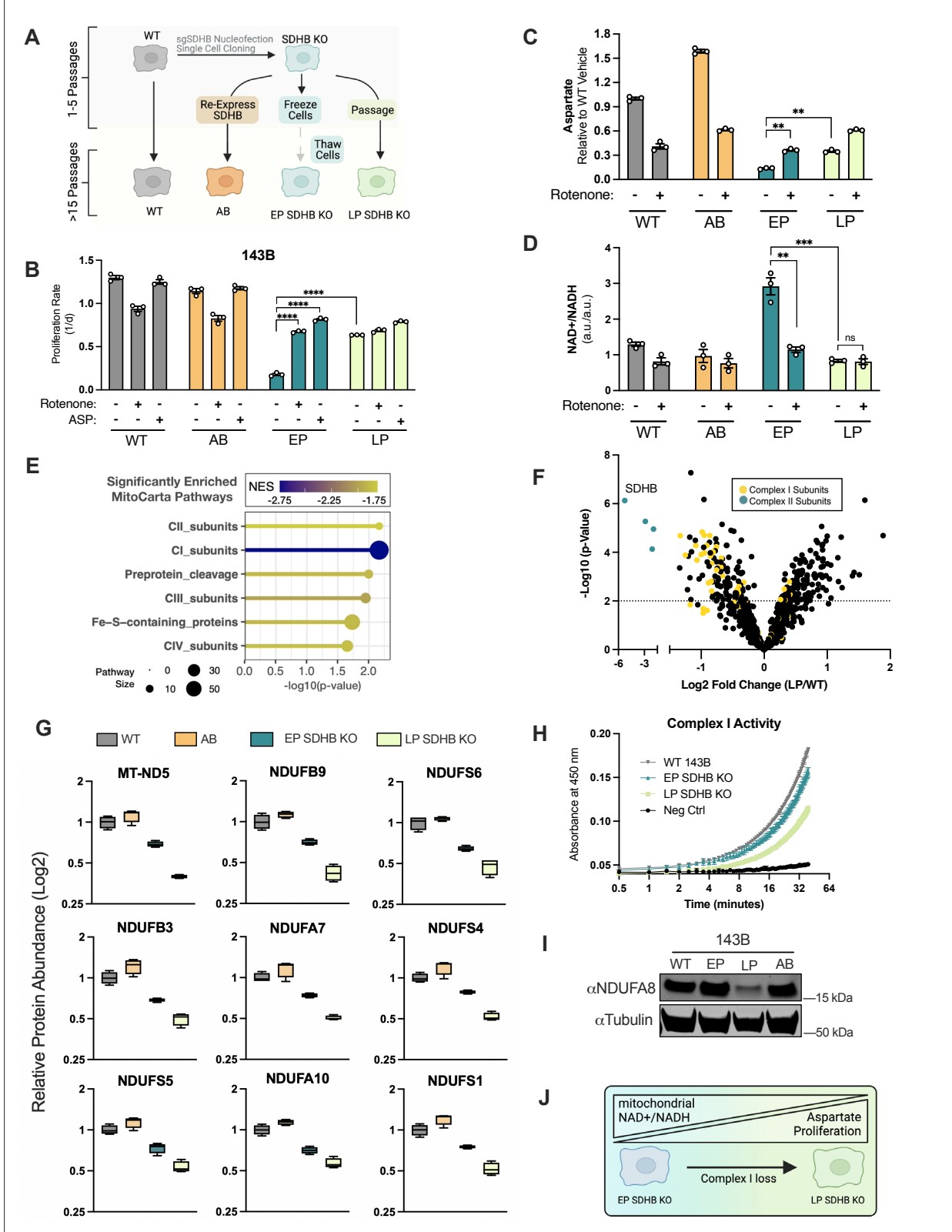

**Figure 7.** Adaptive CI activity loss supports proliferation in SDHB-null cells. (**A**) Schematic showing how SDHB KO 143B clones were used to generate SDHB addback cells (AB), early passage SDHB KO cells (EP), or late passage (LP) SDHB KO cells. (**B**) Proliferation rates of wild-type (WT), SDHB addback (AB), early passage (EP) and late passage (LP) SDHB KO 143B cells cultured in DMEM with 1 mM PYR, treated with vehicle (DMSO), 50 nM rotenone, or 20 mM ASP (n=3). (**C**) Relative aspartate levels measured by LCMS metabolomics of wild-type (WT), SDHB addback (AB), early passage (EP) and late

*Figure 7 continued on next page*

*Figure 7 continued*

passage (LP) SDHB KO 143B cells cultured in DMEM with 1 mM PYR and treated with vehicle (DMSO) or 50 nM rotenone for six hours (n=3). (**D**) Whole cell NAD+/NADH measured by LCMS metabolomics of wild-type (WT), SDHB addback (AB), early passage (EP) and late passage (LP) SDHB KO 143B cells cultured in DMEM with 1 mM PYR and treated with vehicle (DMSO) or 50 nM rotenone for 6 hr (n=3). (**E**) Gene set enrichment analysis (GSEA) of mitochondrial protein expression in LP SDHB KO 143B cells compared to WT 143B cells using MitoCarta 3.0 pathways. (**F**) Volcano plot of compiled LP SDHB KO 143B cells compared to WT 143B cells showing all mitochondrial proteins (black), complex II subunits (blue), and complex I subunits (yellow) (n=4). (**G**) Box plots of a subset of CI subunits showing relative expression in WT 143B, SDHB AB 143B, EP SDHB KO 143B, and LP SDHB KO 143B cells (n=4). (**H**) Complex I activity assay of WT 143B, EP SDHB KO, and LP SDHB KO cells (n=2). (**I**) Western blot for NDUFA8 and tubulin from WT 143B, EP SDHB KO, LP SDHB KO, and SDHB AB cells. Tubulin is used as a loading control. (**J**) Schematic detailing how CI activity is suppressed to generate an ideal mitochondrial redox state in SDH-deficient cells to enable aspartate synthesis. Data are plotted as means ± standard deviation (SD) and compared with an unpaired two-tailed student's t-test. $p<0.05^*$, $p<0.01^{**}$, $p<0.001^{***}$, $p<0.0001^{****}$.

The online version of this article includes the following source data and figure supplement(s) for figure 7:

**Source data 1.** Proliferation rates, relative metabolite levels by LCMS, volcano plot data, and complex I activity values in *Figure 7*.

**Source data 2.** Uncropped western blot for *Figure 7I*.

**Figure supplement 1.** Characterization of adaptations in SDHB KO cells.

**Figure supplement 1—source data 1.** Proliferation rates in *Figure 7—figure supplement 1*.

**Figure supplement 1—source data 2.** Uncropped western blot for *Figure 7—figure supplement 1A*.

treatment remained phenotypically similar to EP cells when CI inhibitors were removed, suggesting the selective pressure is stalled by exogenous CI inhibition (*Figure 7—figure supplement 1D*).

To learn more about the mitochondrial alterations occurring during the passaging of SDHB KO cells, we conducted tandem mass tag (TMT) quantitative proteomics on isolated mitochondria from WT cells, AB cells, EP SDHB KO cells, and LP SDHB KO cells (*Figure 7—figure supplement 1E*). Unsupervised clustering of protein levels found that each group clustered together and that pairs of SDHB KO and replete cells clustered together (*Figure 7—figure supplement 1F*). Comparing WT and LP samples, we conducted MitoPathways 3.0 pathway enrichment analysis (*Pagliarini et al., 2008*; *Rath et al., 2021*) of mitochondrial protein abundance changes and found that, following complex II proteins, CI proteins were the most statistically significantly different (*Figure 7E*). Indeed, the abundances of most CI subunits were decreased in LP SDHB KO cells compared to WT cells (*Figure 7F*). In addition, CI subunit levels progressively decreased from WT/AB to EP SDHB KO to LP SDHB KO and resulted in decreased CI activity, suggesting that SDHB KO cells adapt over time by selecting against CI activity (*Figure 7G and H*). We confirmed these findings by western blot, where LP SDHB KO 143B cells were associated with an ~80% reduction in the CI subunit NDUFA8 compared to WT cells (*Figure 7I*). Together, these data suggest that SDH-deficient cells suppress CI activity to generate an ideal mitochondrial redox state that enables aspartate synthesis (*Figure 7J*).

## Complex I activity is deleterious to SDH-mutant cancer cell proliferation and tumor growth

A potential caveat of generalizing these effects to SDH-mutant cancers is that SDH wild type parental cell lines may not recapitulate the intrinsic metabolic features of naturally arising SDH-mutant cancer cells. We thus measured the CI activity of the SDH-null UOK269 cells and found that similar to LP SDHB KO 143B cells, UOK269 cells had substantially decreased CI activity compared to WT 143B cells (*Figure 8—figure supplement 1A*). We next investigated the metabolic state of late passage (LP) SDHB restored UOK269 cells. We first conducted stable isotope labeling metabolomics from U-$^{13}$C glucose and U-$^{13}$C glutamine to measure carbon contributions to aspartate in WT and LP SDHB UOK269 cells. Consistent with the metabolic phenotypes observed in 143B cells, aspartate was predominantly labeled M+3 from U-$^{13}$C glucose and U-$^{13}$C glutamine in WT UOK269 cells. SDHB restoration was associated with a loss of glucose-derived aspartate and a restoration of oxidative TCA cycle derived M+4 aspartate from glutamine (Figures *Figure 8—figure supplement 1B and C*). Consistent with their low CI activity, WT UOK269 cells had minimal NDUFA8 protein expression compared to 143B cells, but interestingly LP SDHB UOK269 cells had restored their NDUFA8 expression (*Figure 8A*). Correspondingly, LP SDHB UOK269 cells also had substantially increased mitochondrial oxygen consumption and CI activity compared to their SDH-deficient WT counterparts

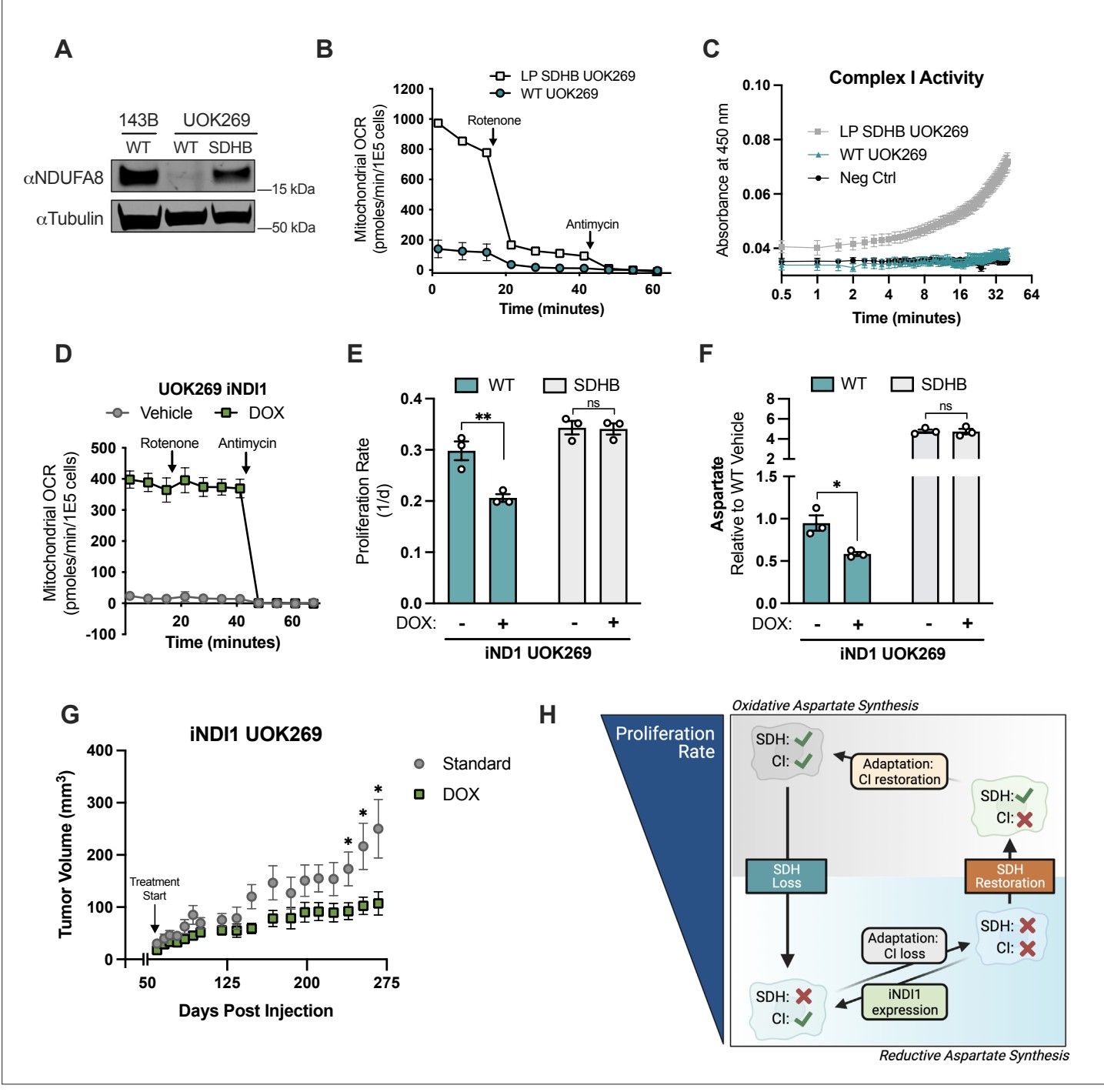

**Figure 8.** Complex I activity is deleterious in SDHB-mutant renal cell carcinoma cells. (**A**) Western blot for NDUFA8 and tubulin from WT 143B, WT UOK269, and late-passage (LP) SDHB UOK269 cells. Tubulin is used as a loading control. (**B**) Mitochondrial oxygen consumption rate of WT UOK269 cells compared to LP SDHB UOK269 cells with indicated injections of 100 nM rotenone and 10 μM antimycin (n=3). (**C**) Complex I activity assay of WT UOK269 and LP SDHB UOK269 cells (n=3). (**D**) Mitochondrial oxygen consumption rates of pInducer20-NDI1 expressing UOK269 cells (UOK269 iNDI1) that were pretreated with vehicle ($H_2O$) or 1 μg/mL doxycycline for 40 hr as indicated, followed by injections of 100 nM rotenone and 10 μM antimycin (n=3). (**E**) Proliferation rates of WT and SDHB UOK269 cells with expression of pInducer20-NDI1 (iNDI1) cultured in DMEM with 1 mM PYR and treated with vehicle ($H_2O$) or 1 μg/mL doxycycline (n=3). (**F**) Aspartate levels measured by LCMS metabolomics of WT and SDHB UOK269 cells with expression of pInducer20-NDI1 (iNDI1) cultured in DMEM with 1 mM PYR and treated with vehicle ($H_2O$) or 1 μg/mL doxycycline for 24 hr (n=3). (**G**) Tumor volumes of iNDI1 UOK269 cells implanted into the flanks of immunocompromised SCID mice. After tumors became palpable, mice were fed either standard or doxycycline-containing chow (n=10 tumors per condition). (**H**) Model demonstrating adaptations to SDH loss and SDH restoration, correlating to

*Figure 8 continued on next page*

*Figure 8 continued*

aspartate levels and proliferation rate. Data are plotted as means ± standard deviation (SD) except in B and D which are means ± standard error of the mean (SEM) and compared with an unpaired two-tailed student's t-test. p<0.05*, p<0.01**, p<0.001***, p<0.0001****.

The online version of this article includes the following source data and figure supplement(s) for figure 8:

**Source data 1.** Proliferation rates, mitochondrial oxygen consumption values, complex I activity values, relative metabolite levels by LCMS, and tumor volumes in *Figure 8*.

**Source data 2.** Uncropped western blot for *Figure 8A*.

**Figure supplement 1.** Characterization of WT and SDHB UOK269 cells.

**Figure supplement 1—source data 1.** Proliferation rates, relative metabolite levels by LCMS, fractional isotopologue distribution values, and mitochondrial oxygen consumption values in *Figure 8—figure supplement 1*.

(*Figure 8B and C*). These results therefore indicate that the adaptations in CI status are reversible to achieve concordant activity with that of SDH.

We next hypothesized that CI loss was critical for UOK269 cells by enabling aspartate production and cell proliferation, and that restoring CI activity would therefore be deleterious. Mammalian CI activity is dependent on the expression of >50 genes encoding core subunits and assembly factors, so increasing CI activity by traditional gene overexpression is not feasible. Instead, we expressed a doxy-cycline (DOX) inducible construct of the *Saccharomyces cerevisiae* NDI1, a rotenone-insensitive CI analog that can restore NADH oxidation and ETC activity in mammalian cells when CI is impaired (*Seo et al., 1998*; *Wheaton et al., 2014*). DOX treatment of WT UOK269 cells expressing the inducible (iNDI1) construct resulted in a substantial induction of mitochondrial oxygen consumption that was resistant to rotenone treatment, verifying NDI1 expression and confirming that decreased CI activity is a bottleneck for ETC function in UOK269 cells (*Figure 8D*). NDI1 expression decreased the proliferation of WT UOK269 cells but had no effect on the proliferation of their SDHB-restored counterparts (*Figure 8E*). Metabolomics measurements also found that NDI1 expression lowered aspartate levels in WT UOK269 cells, but not in the SDHB-restored UOK269 cells (*Figure 8F*). Since DOX can act as a mitochondrial inhibitor at some doses, we also confirmed that DOX had no effect on proliferation rates or aspartate levels in the absence of the iNDI1 construct (*Figure 8—figure supplement 1D and E*). Finally, we tested whether the deleterious effects of CI activity restoration on UOK269 cell proliferation were maintained in a physiological metabolic context by generating mouse xenograft tumors. Notably, DOX-mediated NDI expression also impaired the growth of tumors arising from iNDI1 UOK269 cells (*Figure 8G*). Together, these findings highlight that cells adapt to SDH loss by selecting for concordant CI loss, which is required to maximize aspartate production, cell proliferation rate, and tumorigenesis in SDH-null cancer cells (*Figure 8H*).

## Discussion

Mitochondria are established contributors to catabolic cell metabolism through their canonical oxidative TCA cycle and ETC activities. In proliferating cells, the anabolic contributions of these processes in NAD+/NADH homeostasis and TCA cycle-derived aspartate production have recently gained appreciation, defining important amphibolic roles for oxidative mitochondrial metabolism (*Birsoy et al., 2015*; *Sullivan et al., 2015*; *Titov et al., 2016*). However, the fact that loss-of-function mutations in genes encoding the TCA cycle enzymes SDH and Fumarate Hydratase (FH) can promote tumorigenesis in some contexts challenges the notion that mitochondrial oxidative anabolism is strictly essential for aspartate production. Here, we investigate the metabolic roles of SDH and find that, in cells that employ oxidative TCA cycle function, SDH activity is primarily required for cell proliferation by supporting aspartate production. Interestingly, we found that cells can tolerate SDH loss if they also have concomitant CI impairment, which supports cell proliferation by decreasing mitochondrial NAD+/NADH to enable alternative aspartate synthesis in the cytosol, driven by PC and RCQ. The linked nature of SDH and CI was also revealed by reciprocal experiments where either SDHB was knocked out in wild type cells or restored in cancer cells arising from SDHB mutations. In both cases, functional SDH selects for cells with high CI activity and impaired SDH selects for cells with low CI activity, suggesting two modalities for mitochondrial anabolic metabolism: (1) CI regenerates NAD +to drive oxidative anabolism via the TCA cycle, or (2) CI loss prevents NADH consumption, supporting

NADH-driven reductive anabolism in the mitochondria of cells without canonical TCA cycle function. Overall, these findings support a growing understanding that mitochondrial function has critical roles in supporting cell function beyond its classical role in oxidative ATP synthesis, which can be revealed by divergent phenotypic effects of impairments to discrete components of mitochondrial metabolism (*Arnold et al., 2022*; *Martínez-Reyes et al., 2020*; *Mullen et al., 2011*; *Ryan et al., 2021*; *To et al., 2019*).

Our results highlight the interconnectedness of cellular metabolism by demonstrating how inter-compartmental metabolic variables can collaborate to promote metabolic adaptations and support cell function. The observations that CI inhibition primarily decreases mitochondrial NAD+/NADH and that only mito*LbNOX* blocks the benefits of CI inhibition in SDH-impaired cells localize the site of redox effect to the mitochondria. Interestingly, we find that CI inhibition promotes aspartate synthesis in SDH-deficient cells by driving PC and RCQ activity; however, neither path directly consumes mitochondrial NADH for its function. Therefore, understanding the mechanistic connection between mitochondrial NAD+/NADH and the activation of those pathways is an important goal for future study. Additionally, we note that SDH/CI impaired cells also require a supply of cytosolic electron acceptors to proliferate, indicating that this metabolic state depends on intercompartmental metabolic differences including a reduced mitochondrial compartment and an oxidized cytosolic environment. These results therefore reveal that compartmentally localized redox state changes can functionally alter cell phenotypes, highlighting the need for tools to measure subcellular metabolic states to gain a comprehensive understanding of cell metabolism.

Metabolic changes are linked with tumorigenesis, which is epitomized by cancers deriving from mutations in TCA cycle genes. We found that cancer cells devoid of SDH activity enact further metabolic changes by suppressing CI, which is critical for optimal tumor growth and drives alternative aspartate synthesis pathways through MPC, PC, RCQ, and GOT1, suggesting that these pathways could be potential targets for SDH-mutant cancers. Interestingly, the phenomenon of CI loss has also been observed in several other kidney neoplasias, including FH-mutant type 2 papillary renal cell carcinomas (pRCC), renal oncocytomas driven by mitochondrial DNA mutations, and VHL-mutant clear cell renal cell carcinomas (ccRCC), which are characterized by suppression of oxidative mitochondrial metabolism (*Courtney et al., 2018*; *Crooks et al., 2021*; *Mayr et al., 2008*; *Simonnet et al., 2003*; *Tomlinson et al., 2002*). Decreases in CI expression and/or heteroplasmic mutations in mitochondrial DNA CI genes have also been reported across diverse cancers, suggesting that activation of reductive mitochondrial anabolism may be a metabolic feature of those cancers and correspond to similar metabolic vulnerabilities (*Gorelick et al., 2021*; *Kumari et al., 2020*; *Reznik et al., 2017*). On the contrary, CI inhibitors have also been identified as a therapeutic avenue for cancer treatment, which can decrease tumor growth by affecting redox homeostasis and suppressing intratumoral aspartate (*Baccelli et al., 2019*; *Gui et al., 2016*; *Martínez-Reyes et al., 2020*; *Molina et al., 2018*; *Sullivan et al., 2018*; *Wheaton et al., 2014*). Notably, the responses to these mitochondrial inhibitors have been variable across clinical trials and cancer models suggesting that heterogeneity in mitochondrial metabolism may govern the responses to CI inhibitors, and potentially to future inhibitors of reductive mitochondrial anabolism (*Janku et al., 2021*; *Momcilovic et al., 2019*; *Yap et al., 2019*; *Yap et al., 2023*). Thus, understanding mitochondrial phenotypes will be critical for leveraging cancer cell metabolic changes to improve cancer therapy.

## Materials and methods
### Cell culture
Cell lines were acquired from ATCC (143B, HEK293T, A549, HCT116), as a gift from Dr. W. Marston Linehan, NCI (UOK269), or as a gift from Dr. Stanley Lee, Fred Hutch (TF-1). 143B cells are deficient in thymidine kinase (TK). Cell identities were confirmed by satellite tandem repeat profiling and cells were tested to be free from mycoplasma (MycoProbe, R&D Systems). Cells were maintained in Dulbecco's Modified Eagle's Medium (DMEM) (Gibco, 50–003-PB) supplemented with 3.7 g/L sodium bicarbonate (Sigma-Aldrich, S6297), 10% fetal bovine serum (FBS) (Gibco, 26140079) and 1% penicillin-streptomycin solution (Sigma-Aldrich, P4333). TF-1 cells were also supplemented with the 2 ng/ml Human Granulocyte Macrophage-Colony Stimulating Factor (GM-CSF) (Shenandoah Biotechnology, 100–08). Cells were incubated in a humidified incubator at 37 °C with 5% $CO_2$.

## Proliferation assays

Cells were trypsinized (Corning, 25,051 CI), resuspended in media, counted (Beckman Coulter Counter Multisizer 4 or Nexcelom Auto T4 Cellometer), and seeded overnight onto six-well dishes (Corning, 3516) with an initial seeding density of 20,000 cells/well (143B, HEK293T, HCT116, 786-O, A549) or 30,000 cells/well (UOK269, SDHB KO 143B). After overnight incubation, 3–6 wells were counted for a starting cell count at the time of treatment. Cells were washed twice in phosphate-buffered saline (PBS) and 4 mL of treatment media was added. Experiments were conducted in DMEM without pyruvate (Corning 50–013-PB) supplemented with 3.7 g/L sodium bicarbonate 10% dialyzed fetal bovine serum (FBS) (Sigma-Aldrich, F0392) and 1% penicillin-streptomycin solution, with or without 1 mM sodium pyruvate (PYR) (Sigma-Aldrich, P8574), 1 mM 2-ketobutyric acid (AKB) (Sigma-Aldrich, K401), 1–20 mM Aspartic Acid (ASP) (Sigma-Aldrich, A7219), or 1 mM Asparagine (Sigma-Aldrich, A7094). Proliferation assays contain 1 mM PYR unless otherwise noted. Proliferation assays using Human Plasma-Like Medium (HPLM) (ThermoFisher, A4899101) were supplemented with 10% dialyzed FBS, 1% penicillin-streptomycin solution, with or without 20 mM ASP. Drug treatments included rotenone (Sigma-Aldrich, R8875), metformin (Sigma-Aldrich, D150959), piericidin A (Cayman Chemical, 15379), atpenin A5 (Cayman Chemical, 11898; AdipoGen, AG-CN2-0110; Abcam, ab144194; or Enzo Life Sciences, ALX-380–313), doxycycline hydrochloride (Sigma-Aldrich, D3447), AZD7545 (Cayman Chemical, 19282), antimycin A (Sigma-Aldrich, A8674), oligomycin (Sigma-Aldrich, 495455), BMS-303141 (Cayman, 16239; MedChemExpress, HY-16107) and DMSO vehicle (D2650). Cells were incubated in a humidified incubator at 37 °C with 5% $CO_2$ then counted after 4–6 days. Cells for the hypoxia experiments were incubated in a 37 °C humidified hypoxia chamber (Coy Lab Products, 8375065) at various $O_2$ concentrations after drug treatment. Proliferation rate was determined by the following equation: Proliferation rate (doublings per day, 1/d) = ($\log_2$(final cell count / initial cell count))/total days.

## Lentiviral production and infection

The following plasmids were obtained from DNASU Plasmid Repository: pLenti6.3-V5 DEST_SLC1A3, pLenti6.3-V5-DEST_SDHB, pLX304_PC, pLX304_GOT1, pLenti6.3-V5-DEST_MPC1, pLX304_MDH1, and pDONR201-NDI1. pLX304-gpASNase1 and pDONR-cyto*LbNOX* were previously described. Cyto*LbNOX* was cloned into pLX304 (Addgene, 25890) and NDI1 was cloned into pInducer20 (Addgene, 44012, gift from Stephen Elledge) using LR Clonase II (Fisher, 11791100). pLentiCMV-GOT2 was constructed by purchasing GOT2 cDNA (Integrated DNA Technologies), which was amplified by PCR and assembled with the pENTR1A vector using NEBuilder HiFI DNA Assembly Cloning Kit (New England BioLabs, E2621). pENTR1A-GOT2 was then used to transfer GOT2 to the pLenti-CMV-Hygro Destination vector (w117-1) (Addgene, 17,454 a gift from Eric Campeau & Paul Kaufman) using Gateway LR Clonase II (Fisher, 11791020). Lentivirus was generated by transfection of HEK293T cells with expression construct plasmid DNA with pMDLg/pRRE (Addgene, 12251), pRSV-Rev, (Addgene, 12253), and pMD2.G (Addgene, 12259) packaging plasmids. and FuGENE transfection reagent (Fisher, PRE2693) in DMEM (Fisher, MT10017CV) without FBS or penicillin-streptomycin. The supernatant containing lentiviral particles was filtered through 0.45 µM membrane (Fisher, 9720514) and was supplemented with 8 µg/µL polybrene (Sigma, TR-1003-G) prior to infection. 143B and UOK269 cells were seeded at 50% confluency in six-well dishes and centrifuged with lentivirus (900 x*g*, 90 min, 30 °C). After 24 hr media was replaced with fresh media and 48 hr after infection cells were treated with either 1 µg/mL blasticidin (Fisher, R21001), 150 µg/mL hygromycin (Sigma, H7772), or 1 mg/mL G418 (Sigma, A1720) and maintained in selection media until all uninfected control cells died.

## Generation of mitoLbNOX expressing cells

*LbNOX*-FLAG DNA was amplified from pDONR-cyto*LbNOX* by PCR, an oligonucleotide containing the zmLOC100282174 mitochondrial translocation sequence (MTS), which causes potent mitochondrial localization, (*Chin et al., 2018*) was purchased (Integrated DNA Technologies) and the pENTR1A vector (Fisher, A10462) was amplified by PCR to create a linear fragment. The three fragments were assembled using the NEBuilder HiFi DNA Assembly Cloning Kit (New England BioLabs, E2621) to generate pENTR1A-mito*LbNOX*, which was then used as a donor to transfer mitoLbNOX into pLenti-CMV-Hygro-DEST (w117-1) (Addgene, 17,454 a gift from Eric Campeau & Paul Kaufman) using Gateway LR Clonase II (Fisher, 11791020). Lentivirus was generated with pLentiCMV-mito*LbNOX* as

described above and 143B cells were infected and then selected in 150 µg/mL Hygromycin B (Sigma, H7772) for 4 days.

*LbNOX*-FLAG primers:

> 5'-GCCGCGAGACCGTATGCTCATAAGGTCACCGTG-3' (fw primer).
> 5'-CAAGAAAGCTGGGTCTAGTTACTTGTCATCGTCATCC-3' (rv primer).

pENTR primers:

> 5'-CTAGACCCAGCTTTCTTGTAC-3' (fw)
> 5'-CTGGCTTTTAGTAAGCGAATTC-3' (rv).

## MTS sequence:

5'-CGCTTACTAAAAGCCAGGCCACCATGGCACTGCTTCGCGCCGCCGTTTCAGAACTCAGAC
GGAGAGGACGGGGTGCGCTTACTCCCCTCCCGGCGCTGTCTAGCTTGCTTTCCTCACTTAGCCC
CCGAAGTCCCGCCTCAACGCGCCCAGAGCCAAACAATCCACACGCAGATCGACGCCATGTCATC
GCTTTGAGGCGATGCCCCCCACTTCCTGCCTCTGCCGTTCTGGCACCTGAACTCCTGCATGCAC
GAGGATTGCTCCCGAGACATTGGTCTCATGCCTCTCCCTTGTCCACGTCCTCTTCATCCAGTAG
ACCAGCAGATAAGGCGCAGTTGACCTGGGTCGATAAATGGATCCCAGAAGCCGCGAGACCGTAT
-3'.

## Generation of knockout cell lines

Protocol was adapted from *Hoellerbauer et al., 2020*. Three chemically synthesized 2'-O-methyl 3'phosphorothioate-modified single guide RNA (sgRNA) sequences targeting the gene of interest were purchased (Synthego) and are listed in the table below. Each sgRNA was resuspended in nuclease-free water, combined with SF buffer (Lonza, V4XC-2032), and sNLS-spCas9 (Aldevron, 9212). $2x10^5$ 143 B or HEK293T cells were resuspended in the resulting solution containing ribonucleoprotein complexes (RNPs) and electroporated using a 4D-Nucleofector (Amaxa, Lonza) programs FP-133 (143B) and DS-150 (HEK293T). Nucleofected cells were then moved to a 12-well plate (Corning, 3513) and, after achieving confluence, were single-cell cloned by limiting dilution by plating 0.5 cells/well in a 96-well plate. Gene knockout was confirmed using western blots on the nucleofected pool and each single-cell clone used in this study.

| Gene | sgRNA sequence (5'–3') |
|------|------------------------|
| SDHB | UCGCCCUCUCCUUGAGGCG<br>AGAAAUUUGCCAUCUAUCGA<br>CUUUGUUAGAUGUGGCCCCA |
| PC | GCAGGCCCGGAACACACGGA<br>GCUGGAGGAGAAUUACACCC<br>ACACCGGCCGCAUUGAGGU |
| MPC1 | UUGCCUACAAGGUACAGCCU<br>GGGCUACUUCAUUUGUUGCG<br>AUGUCAAAGAAUAGCAACAG |
| GOT1 | CAGUCAUCCGUGCGAUAUGC<br>GCACGGAUGACUGCCAUCCC<br>CGAUCUUCUCCAUCUGGGAA |
| GOT2 | UUUCUCAUUUCAGCUCCUGG<br>CGGACGCUAGGCAGAACGUA<br>UCCUUCCACUGUUCCGGACG |
| MDH1 | CCAAUCAGAGUCCUUGUGAC<br>CCACAAGAAUGGCCACAUCC<br>GUGAAAAUCUUCAAAUCCCA |

## Oxygen consumption

Oxygen consumption measurements were conducted using an Agilent Seahorse Xfp Analyzer. 143B or UOK269 cell lines were trypsinized and seeded at $2.5x10^5$ cells/well of a Seahorse XFp cell culture

miniplate (Agilent, 103025–1000) overnight in 80 µL media. The following day, 100 µL cell culture media was added with or without pretreatment, as indicated. Before the assay Seahorse XFp sensor cartridges (Agilent, 103022–100) were incubated with calibrant solution, following manufacturer's instructions and loaded with injection solutions yielding the following final concentrations: AA5; 5 µM, rotenone; 100 nM. AZD7545; 5 µM, antimycin A; 10 µM. Following each assay, cells of each well were counted by Coulter Counter and mitochondrial oxygen consumption rate was determined as the measured oxygen consumption rate minus the average of post-antimycin oxygen consumption rates, per 100,000 cells.

## Mitochondrial fractionation

A total of $2x10^7$ cells were trypsinized, washed with PBS, and centrifuged (300 x$g$, 5 min, 4 °C). Pellets were washed with ice-cold PBS and resuspended in Homogenization buffer (10 mM Tris-HCl pH 6.7, 10 mM KCl, 0.15 mM MgCl$_2$, 1 mM PMSF Sigma, 10837091001). Cells were then vortexed, incubated on ice for 2 min, then transferred to a Dounce Homogenizer (Sigma, D9063) and homogenized with 40–60 strokes. Cells were transferred to mitochondrial suspension buffer (10 mM Tris-HCl pH 6.7, 0.15 mM MgCl$_2$, 0.25 mM sucrose, 1 mM PSMF) and centrifuged (700 x$g$, 10 min, 4 °C). The supernatant was transferred and centrifuged again (10,500 x$g$, 15 min, 4 °C). The resulting supernatant (cytosolic fraction) was transferred and centrifuged once more, (17,000 x$g$, 15 min, 4 °C) while the pellet containing the mitochondrial fraction was washed in suspension buffer and centrifuged again (12,000 x$g$, 15 min, 4 °C). Mitochondrial proteins were then harvested with RIPA buffer (see below).

## Western blotting

Protein lysates were harvested in RIPA buffer (Sigma, R0278) supplemented with protease inhibitors (Fisher, A32953). Protein concentration was determined using a Bicinchoninic Acid Assay (Fisher, 23225) using bovine serum albumin (BSA) as a protein standard. Equal amounts of protein were denatured with Bolt 4 x Loading Dye (ThermoFisher, B0007) and Bolt 10 x reducing agent (ThermoFisher, B0004), heated at 95 °C for 5 min, and loaded onto 4–12% by SDS-polyacrylamide gels (Invitrogen, NW04127), apart from MPC1, for which we used 16% Novex Tricine gels for separation of low molecular weight proteins (Invitrogen, EC6695C). After electrophoretic separation, proteins were transferred onto a 0.22 mm nitrocellulose using iBlot2 transfer stacks (Fisher, IB23001) and transferred with the P0 system setting. Membranes were blocked with 5% milk in Tris-buffered saline with 0.1% Tween-20 (TBS-T) and incubated at 4 °C overnight with the following antibodies: anti-FLAG (Sigma, F1804; 1:1000), anti-SLC1A3 (Genetex, GTX20262; 1:500), anti-SDHB (Atlas, HPA002868; 1:1000), anti-PC (Proteintech, 66615–1-IG, 1:1000), anti-MPC1, (Cell Signaling, 14462 S, 1:1000), anti-GOT2 (Proteintech, 14800–1-AP, 1:750), anti-GOT1 (Cell Signaling, 34423 S, 1:1000), anti-MDH1 (Proteintech, 15904–1-AP, 1:5000), anti-NDUFA8 (Atlas, HPA041510; 1:1000), anti-Vinculin (Sigma, SAB4200729; 1:10,000), and anti-Tubulin (Sigma, T6199; 1:10,000). The next morning, membranes were washed three times with TBS-T and the following secondary antibodies were added: 800CW Goat anti-Mouse IgG (LiCOR, 926–32210; 1:15,000), 680RD Goat anti-Rabbit IgG (LiCOR, 926–68071; 1:15,000). Membranes were washed three more times with TBS-T and imaged on a LiCOR Odyssey Near-Infrared imaging system.

## TMT-quantitative mitochondrial proteomics

### Sample preparation

Mitochondrial fractions of WT 143B, SDHB AB 143B, EP SDHB KO 143B, and LP SDHB KO 143B cells in quadruplicate were fractionated using the protocol described above. Mitochondrial pellets were kept at –80 °C until analysis.

### Disulfide bond reduction/alkylation

Protein solutions (100 µg) were diluted to 2 µg/µL in 100 mM ammonium bicarbonate. Protein disulfide bonds were reduced by adding tris (2-carboxyethyl) phosphine to a final concentration of 5 mM and mixing at room temperature for 15 min. The reduced proteins were alkylated by adding 2-chloroacetamide to a final concentration of 10 mM and mixing in the dark at room temperature for 30 min. Excess 2-chloroacetamide was quenched by the addition of dithiothreitol to 10 mM and mixing at room temperature for 15 min.

## Methanol-chloroform precipitation and protease digestion

Samples (100 µg) were diluted to 1 µg/µL with 100 mM ammonium bicarbonate in a 1.5 mL Eppendorf low bind tube. Protein precipitation was done as follows: 400 µL of methanol was added to each sample and vortexed for 5 s. A total of 100 µL of chloroform was added to each sample and vortexed for 5 s. A total of 300 µL of water was added to each sample and vortexed for 5 s. The samples were centrifuged for 1 min at 14,000 x*g*. The aqueous and organic phases were removed, leaving a protein wafer in the tube. The protein wafers were washed with 400 µL of methanol and centrifuged at 21,000 *g* at room temperature for 2 min. The supernatants were removed, and the pellets were allowed to air dry, but not to complete dryness. The samples were resuspended in 70 µL 100 mM HEPES (pH 8.5) and digested with rLys-C protease (100:1, protein to protease) with mixing at 37 °C for 4 hr. Trypsin protease (100:1, protein to protease) was added and the reaction was mixed overnight at 37 °C.

## TMT-labeling

TMTpro16plex labeling reagent (Pierce 500 µg) was brought up in 30 µL acetonitrile and added to the digested peptide solution (100 µg) yielding a final organic concentration of 30% (v/v) and mixed at room temperature for 1 hr. A 2 µg aliquot from each sample was combined, dried to remove the acetonitrile, concentrated on a C18 ZipTip (Millipore) and analyzed via LC/MS as a 'label check'. The label check was used for two purposes. First, it ensured the TMT labeling efficiency was greater than 95%. Second, sample equalization volumes were determined after summing the total intensity from each sample TMT 'channel' and calculating the volume of each sample that should be mixed to provide equal amounts from each sample in the final mixture. After labeling efficiency was determined, the reactions were quenched with hydroxylamine to a final concentration of 0.3% (v/v) for 15 min with mixing. The TMTpro16plex-labeled samples were pooled at 1:1 ratio based on calculated equalization volumes and concentrated by vacuum centrifugation to remove acetonitrile. Half of the material was desalted over an Oasis HLB 3 cc cartridge (Waters) and taken to dryness.

## bRP fractionation

Pierce's High pH Reversed-Phased Peptide Fractionation kit (part# 84868) was used to fractionate the sample (100 µg) into eight fractions with cuts at 5, 7.5, 10.0, 12.5, 15, 17.5, 20.0, 22.5, 25.0, and 50.0% acetonitrile in 0.1% triethylamine following the manufacturer's protocol. The fractions were dried in a speedvac.

## Mass spectrometry analysis

The generated basic reverse phase fractions were brought up in 2% acetonitrile in 0.1% formic acid (20 µL) and analyzed (2 µL) by LC/ESI MS/MS with a Thermo Scientific Easy1200 nLC (Thermo Scientific, Waltham, MA) coupled to a tribrid Orbitrap Eclipse with FAIMS pro (Thermo Scientific, Waltham, MA) mass spectrometer. In-line de-salting was accomplished using a reversed-phase trap column (100 µm×20 mm) packed with Magic $C_{18}$AQ (5 µm, 200 Å resin; Michrom Bioresources, Auburn, CA) followed by peptide separations on a reversed-phase column (75 µm×270 mm) packed with Repro-Sil-Pur $C_{18}$AQ (3 µm, 120 Å resin; Dr. Maisch, Baden-Würtemburg, Germany) directly mounted on the electrospray ion source. A 180-min gradient from 4% to 44% B (80% acetonitrile in 0.1% formic acid/water) at a flow rate of 300 nL/min was used for chromatographic separations. A spray voltage of 2300 V was applied to the electrospray tip in-line with a FAIMS pro source using varied compensation voltage −40,−60, −80 while the Orbitrap Eclipse instrument was operated in the data-dependent mode. MS survey scans were in the Orbitrap (Normalized AGC target value 300%, resolution 120,000, and max injection time 50ms) with a 3 s cycle time and MS/MS spectra acquisition were detected in the linear ion trap (Normalized AGC target value of 100% and injection time 50ms) using CID activation with a normalized collision energy (NCE) of 32% using turbo speed scan. Selected ions were dynamically excluded for 60 s after a repeat count of 1. Following MS2 acquisition, real time searching (RTS) was employed, and spectra were searched against a Human database (UP00005640 Human 120119) using COMET. Searches were performed with settings for the proteolytic enzyme trypsin. Maximum missed cleavages were set to 1 and maximum variable modifications on peptides was set to 3. Variable modifications included oxidation (+15.995 Da on M) with static modifications TMTpro

(+304.207 DA on K) and carbamidomethyl (+57.021 on C). Maximum search time was 35ms. Scoring thresholds were set to the following: Xcorr 1.4, dCn 0.1, precursor PPM 10 and charge state 2. TMT Synchronous precursor selection (SPS) MS3 was collected on the top 10 most intense ions detected in the MS2 spectrum. SPS-MS3 precursors were subjected to higher energy collision-induced dissociation (HCD) for fragmentation with an NCE of 45% and analyzed using the Orbitrap (Normalized AGC target value of 400%, resolution 60,000 and max injection time 118ms).

## Data analysis
Data analysis was performed using Proteome Discoverer 2.5 (Thermo Scientific, San Jose, CA). The data were searched against a Human database (UP00005640 Human 120119) that included common contaminants (cRAPome). Searches were performed with settings for the proteolytic enzyme trypsin. Maximum missed cleavages were set to 2. The precursor ion tolerance was set to 10 ppm and the fragment ion tolerance was set to 0.6 Da. Dynamic peptide modifications included oxidation (+15.995 Da on M). Dynamic modifications on the protein terminus included acetyl (+42.–11 Da on N-terminus), Met-loss (–131.040 Da on M) and Met-loss +Acetyl (–89.030 Da on M) and static modifications TMTpro (+304.207 Da on any N-termius), TMTpro (+304.207 DA on K), and carbamidomethyl (+57.021 on C). Sequest HT was used for database searching. All search results were run through Percolator for scoring. Unsupervised clustering of the 16 proteomic samples was performed using the z-score based R (version 4.0.3) package GMD (*Zhao and Sandelin, 2012*) and visualized with the heatmap.3 function.

## Polar metabolite extractions
### Adherent cells
For standard metabolic analysis, cells were seeded overnight at $2x10^5$ cells per well of a six-well dish. The next morning, cells were washed twice with PBS and changed to the indicated media supplemented with 10% dialyzed FBS, 1% penicillin-streptomycin, and treatments as indicated, and returned to the tissue culture incubator. After 6 hr, polar metabolites were extracted from cells by three rapid washes with ice-cold blood bank saline, (Fisher, 23293184) then 250 µL of 80% HPLC grade methanol in HPLC grade water containing 2.5 µM D8 Valine (Cambridge Isotope Laboratories, DLM-488) or $^{13}$C amino acid mix (Cambridgre Isotope Laboratories, MSK-CAA-1) was added to each well and cells were scraped with the back of a P1000 pipet tip and transferred to Eppendorf tubes. Tubes were centrifuged (17,000 x*g*, 15 min, 4 °C) and the supernatant containing polar metabolites was transferred to a new centrifuge tube and placed in a centrivap until lyophilized.

### Suspension cells
TF-1 cells were seeded at $2.5x10^5$ cells per well of a ix-well dish the night before treatment and extraction in standard media containing 2 ng/mL human GM-CSF. The next day, cells were washed with PBS and treated with media containing 2 ng/mL human GM-CSF, 10% dialyzed FBS, and the treatments as indicated. After 6 hr, cells were collected and washed in 1 mL of ice-cold saline containing 2% dialyzed FBS. Metabolism was quenched and metabolites were extracted with 600 µL of 80% HPLC grade methanol in HPLC grade water. Samples were centrifuged, (17,000 x*g*, 15 min, 4 °C) moved to new Eppendorf tubes, lyophilized, and stored at –20 °C until analysis.

## Isotope Tracing
### Aspartate uptake
SLC1A3 143B cells were plated at $2x10^5$ cells per well of a six-well dish. The next morning, cells were washed twice with PBS and changed to DMEM without pyruvate supplemented with 10% dialyzed FBS, 1% penicillin-streptomycin, and treated with the indicated concentrations of 1,4-$^{13}$C2 aspartate (Cambridge Isotopes, CLM-4455) for 1 hr.

### Glutamine tracing
WT and SDHB KO 143B cells were seeded at $2x10^5$ cells and UOK269 cell lines were plated at $3x10^5$ cells per well of a six-well dish. The next morning, cells were washed twice with PBS and changed to DMEM without glucose, glutamine, pyruvate, or phenol red (Sigma, D5030) supplemented with 10% dialyzed

FBS, 1% penicillin-streptomycin, 1 mM pyruvate, 25 mM $^{12}$C glucose (Sigma, G7528), and 4 mM U-$^{13}$C glutamine (Cambridge Isotopes, CLM-1822). 143B cells were treated as indicated for 6 hr.

## Glucose tracing

WT and SDHB KO 143B cells were seeded at $2 \times 10^5$ cells per well of a six-well dish. The next morning, cells were washed twice with PBS and swapped to DMEM without glucose, glutamine, pyruvate, or phenol red supplemented with 10% dialyzed FBS, 1% penicillin-streptomycin, 1 mM AKB, 25 mM U-$^{13}$C glucose (Cambridge Isotopes, CLM-1396), and 4 mM $^{12}$C glutamine (Sigma, G5792). Polar metabolites were extracted with the above technique.

## Rapid subcellular NAD+/NADH detection

### Rapid subcellular fractionation

143B cells were plated at $1 \times 10^5$ cells per well of a six-well dish the night before rapid subcellular fractionation, a protocol adapted from *Lee et al., 2019*. Cells were washed twice with PBS and treated with DMEM containing 10% dialyzed FBS, 1% penicillin-streptomycin, 1 mM pyruvate, and treatments as indicated. After 6 hr of treatment, cells were washed with ice cold PBS and scraped into a 1.5 mL microcentrifuge tube. After a brief centrifuge (13,500 x*g*, 10 s, 4 °C), the supernatant was discarded, and the cells were resuspended in 250 μL of digitonin (Sigma, D141) dissolved at 1 mg/mL in ice cold PBS and triturated five times with a P1000 micropipette tip. The sample was then centrifuged (13,500x*g*, 10 s, 4 °C) and the supernatant was collected for the cytosolic fraction, while the remaining pellet contained the mitochondria. The cytosolic fraction was mixed 1:1 with 250 μL of 0.2 N NaOH containing 1% dodecyl trimethylammonium bromide (DTAB; Sigma, D5047). Metabolites for the mitochondrial fraction were extracted immediately using 500 μL of cold lysis buffer (1% DTAB in 0.2 N NaOH at 1:1 (v/v) with PBS). Whole cell lysates were obtained by the same manner as the mitochondrial fraction.

### Compartmentalized NAD+/NADH measurements

NAD +and NADH were then detected using the NAD/NADH-Glo Assay (Promega, G9072) according to the manufacturer's directions and previous reports (*Gui et al., 2016*; *Sullivan et al., 2015*). To detect NAD+, 50 μL of sample was transferred to a PCR tube and treated with 25 μL of 0.4 N HCl and heated to 60 °C for 15 min where acidic conditions selectively degrade NADH. For NADH detection, 50 μL of sample was transferred to a PCR tube and heated to 75 °C for 30 min for selective basic degradation of NAD+. Following incubation, samples were equilibrated at room temperature, then quenched by neutralizing the NAD +acid-treated samples with 25 μL of 0.5 M Trizma base (Sigma, T6066) and the NADH base-treated samples with 50 μL of 1:1 (v/v) 0.4 N HCl:0.5 M Trizma base. Once neutralized, 50 μL of each sample was then mixed with 50 μL of NAD/NADH-Glo Detection Reagent in an opaque white 96-well flat bottom plate. Selective degradation protocols and detection in the linear range were confirmed using chemical standards for NAD+ (Sigma, N1511) and NADH (Sigma N8129). Enzyme-linked luminescence measurements were then recorded on a Tecan Infinite M Plex microplate reader according to the manufacturer instructions.

## Liquid chromatography-mass spectrometry

Lyophilized samples were resuspended in 80% HPLC-grade methanol in HPLC-grade water and transferred to liquid chromatography-mass spectrometry (LCMS) vials for measurement by LCMS. Metabolite quantitation was performed using a Q Exactive HF-X Hybrid Quadrupole-Orbitrap Mass Spectrometer equipped with an Ion Max API source and H-ESI II probe, coupled to a Vanquish Flex Binary UHPLC system (Thermo Scientific). Mass calibrations were completed at a minimum of every 5 days in both the positive and negative polarity modes using LTQ Velos ESI Calibration Solution (Pierce). Polar Samples were chromatographically separated by injecting a sample volume of 1 μL into a SeQuant ZIC-pHILIC Polymeric column (2.1x150 mm 5 mM, EMD Millipore). The flow rate was set to 150 mL/min, autosampler temperature set to 10 °C, and column temperature set to 30 °C. Mobile Phase A consisted of 20 mM ammonium carbonate and 0.1% (v/v) ammonium hydroxide, and Mobile Phase B consisted of 100% acetonitrile. The sample was gradient eluted (%B) from the column as follows: 0–20 min.: linear gradient from 85 % to 20 % B; 20–24 min.: hold at 20% B; 24–24.5 min.: linear gradient from 20 % to 85 % B; 24.5 min.-end: hold at 85% B until equilibrated with ten column

volumes. Mobile Phase was directed into the ion source with the following parameters: sheath gas = 45, auxiliary gas = 15, sweep gas = 2, spray voltage = 2.9 kV in the negative mode or 3.5 kV in the positive mode, capillary temperature = 300 °C, RF level = 40 %, auxiliary gas heater temperature = 325 °C. Mass detection was conducted with a resolution of 240,000 in full scan mode, with an AGC target of 3,000,000 and maximum injection time of 250 msec. Metabolites were detected over a mass range of 70–1050 *m/z*. Quantitation of all metabolites was performed using Tracefinder 4.1 (Thermo Scientific) referencing an in-house metabolite standards library using ≤5 ppm mass error. Metabolite abundances were quantified by comparing ion counts relative to cells reflecting the most unmodified metabolic state for each experiment (WT cells, vehicle treatment, AB cells, as indicated) from cells cultured in DMEM containing an electron acceptor (PYR or AKB, as indicated). In situations where the LCMS dataset contained one or more samples had high sample loading variance, all samples in that dataset were normalized to isotopically labeled loading controls (either valine D8 or U-$^{13}$C aspartate from $^{13}$C amino acid mix) as annotated in *Supplementary file 1*. Isotopically labeled standards for NAD +and NADH are not widely available, so LCMS experiments measuring NAD+/NADH ratios do not incorporate differences in ionization efficiencies or matrix effects for each analyte. Thus, while changes in the ratio of measured ion counts for NAD +and NADH reflect biological changes, they should not be interpreted as an absolute intracellular ratio, which is more accurately quantified by enzymatic methods. Individual ion counts for NAD +and NADH are available in *Supplementary file 1*. Data from stable isotope labeling experiments includes correction for natural isotope abundance using IsoCor software v.2.2.

## Complex I activity assay

Complex I activity was measured using an assay kit (Abcam, ab109721) according to the manufacturer's instructions. Absorbances at 450 nm were read every 30 s for 40 min on a Tecan Infinite M Plex microplate reader.

## Gene set enrichment analysis

Gene Set Enrichment Analysis (GSEA) (*Subramanian et al., 2005*) was conducted on the proteomics dataset utilizing 149 hierarchical, mitochondrial specific pathways curated by MitoCarta3.0 (*Rath et al., 2021*) as the a priori defined gene/protein sets. Proteins were ranked by the signed (negative or positive by the direction of the fold-change), -log of the adjusted p-value, both of which were calculated during differential expression analysis between LP and WT cells and used as input for GSEA. Individual MitoCarta3.0 pathways were evaluated for enrichment of upregulated proteins (positive normalized enrichment score) and downregulated proteins (negative normalized enrichment score). GSEA analysis was implemented in R (*R Core Team, 2020*; version 4.0.3) using the Bioconductor package fgsea (*Korotkevich et al., 2021*). Adjusted p-values and normalized enrichment scores have been reported.

## Mouse xenografts

All mouse work was performed in accordance with FHCC-approved IACUC protocol 51069 and AAALAS guidelines and ethical regulations. iNDI1 UOK269 cells were suspended in 100 µl PBS and injected subcutaneously into both flanks of ten 6–8 week-old, NSG (NOD.*Cg-Prkdc$^{scid}$Il2rg$^{tm1Wjl}$*/SzJ) male mice at $5 \times 10^6$ cells per injection. When tumors became palpable, one cage (n=5 mice per cage) was randomly assigned DOX chow with 625 mg/kg doxycycline hyclate (Envigo, TD.01306), while the other continued with standard chow. Tumor volumes were measured by calipers in two dimensions and estimated volumes were calculated using the equation $V = (\pi/6)(L \times W^2)$.

## Statistical analysis

All graphs and statistical analyses were made in GraphPad Prism 9.0. Technical replicates, defined as parallel biological samples independently treated, collected, and analyzed during the same experiment, are shown. Experiments were verified with independent repetitions showing qualitatively similar results. Details pertaining to all statistical tests can be found in the figure legends.

## Acknowledgements

We thank Brian Milless, Dr. Philip Gafken and the staff of the Fred Hutch Proteomics and Metabolomics Shared Resource for LCMS metabolomics assay development, technical support, and for their assistance with conducting TMT-mitochondrial proteomics. We also thank members of the Sullivan lab, Dr. Lydia Finley, and members of the Finley lab for manuscript discussion and feedback.

## Additional information

### Funding

| Funder | Grant reference number | Author |
|---|---|---|
| National Cancer Institute | P30CA015704 | Lucas B Sullivan |
| National Institute of General Medical Sciences | T32GM095421 | Madeleine L Hart |
| National Cancer Institute | R00CA218679-03S1 | Madeleine L Hart |
| National Cancer Institute | R00CA218679 | Lucas B Sullivan |
| National Institute of General Medical Sciences | R35GM147118 | Lucas B Sullivan |
| Andy Hill Cancer Research Endowment | CARE Award | Lucas B Sullivan |
| National Cancer Institute | U54CA132381 | Samantha M Carlisle |

The funders had no role in study design, data collection and interpretation, or the decision to submit the work for publication.

### Author contributions

Madeleine L Hart, Conceptualization, Data curation, Formal analysis, Validation, Investigation, Visualization, Methodology, Writing – original draft, Writing – review and editing; Evan Quon, Anna-Lena BG Vigil, Ian A Engstrom, Oliver J Newsom, Validation, Investigation, Writing – review and editing; Kristian Davidsen, Methodology, Writing – review and editing; Pia Hoellerbauer, Methodology; Samantha M Carlisle, Data curation, Formal analysis; Lucas B Sullivan, Conceptualization, Resources, Data curation, Formal analysis, Supervision, Funding acquisition, Validation, Investigation, Visualization, Methodology, Writing – original draft, Project administration, Writing – review and editing

### Author ORCIDs

Madeleine L Hart ⓘ http://orcid.org/0000-0001-9125-0627
Kristian Davidsen ⓘ http://orcid.org/0000-0002-3821-6902
Lucas B Sullivan ⓘ http://orcid.org/0000-0002-6745-8222

### Ethics

All mouse work was performed in accordance with FHCC-approved IACUC protocol 51069 and AAALAS guidelines and ethical regulations.

### Decision letter and Author response

Decision letter https://doi.org/10.7554/eLife.78654.sa1
Author response https://doi.org/10.7554/eLife.78654.sa2

## Additional files

### Supplementary files

• MDAR checklist

• Supplementary file 1. Ion counts from each metabolomics experiment conducted in this study. Organized by tab for each figure; tables include cell line, treatment, and metabolite measured.

• Supplementary file 2. Mitochondrial proteomics dataset used to generate *Figure 7E-G*, *Figure 7—figure supplement 1F*.

### Data availability

All data generated or analyzed during this study are included in the manuscript and supporting files. Source data files have been provided for Figures 1–8. Ion counts from metabolomics experiments are available in *Supplementary file 1*, separated by tabs for each graph. The mitochondrial proteomics dataset is provided in *Supplementary file 2*.

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
