## [Editor Report]

Mutations in subunits of succinate dehydrogenase (SDH) are implicated in tumorigenesis, but it is not fully understood how cell proliferation occurs in cells that lack SDH. Hart et al. elegantly demonstrate that lack of mitochondrial complex I increases proliferation in cells lacking SDH. The authors have expertly dissected the mechanism behind the rescue and show that inhibition of complex I restores aspartate production by modulating NAD+/NADH. This study thus demonstrates that compartmentalized redox changes control distinct aspartate biosynthetic pathways that promote the proliferation of tumor cells.

---

## [Decision Letter]

**Decision letter after peer review:**

Thank you for submitting your article "Mitochondrial Redox Adaptations Enable Aspartate Synthesis in SDH-deficient Cells" for consideration by *eLife*. Your article has been reviewed by 3 peer reviewers, one of whom is a member of our Board of Reviewing Editors, and the evaluation has been overseen by Utpal Banerjee as the Senior Editor. The following individual involved in review of your submission has agreed to reveal their identity: Heather Christofk (Reviewer #2).

The manuscript has been reviewed by three reviewers who were all in agreement that the work has potential. However, all of the reviewers identified a number of issues with the manuscript that will need to be addressed for it to be accepted by *eLife*. The authors will see from the essential revisions and from the individual recommendations that a lot of work is expected and it will not be straightforward for the authors to sufficiently improve the manuscript within a reasonable period of time. However, the reviewers all agree that the authors should be given the opportunity to revise the manuscript if they wish to do so. Yet, given the extensive nature of the requested revisions, the reviewers ask that the authors carefully consider whether trying to revise the manuscript for publication in *eLife* is the best option for them and it might be that submission to another journal is preferable.

Essential revisions:

1) The reviewers feel that the data linking Asp to rescue of proliferation are mostly correlative. Much more evidence is needed to conclude that the small changes in aspartate abundance are sufficient to restore proliferation and the presented correlations between aspartate levels and proliferation are not sufficient evidence that aspartate biosynthesis is the primary mode of supporting cell proliferation in SDH-deficient cells.

Similarly, comparison of Asp rescue between different conditions is also challenging and the authors are inconsistent in their interpretation of meaningful changes in aspartate. For example, the recovery of Asp caused by "AA5 + pyruvate" (Figure 1D) appears to be greater than that caused by "AA5 and rotenone" (Figure 2B), yet the recovery of proliferation by "AA5 + pyruvate" is less. As another example, Asp levels in "AA5+mitoLbNOX" (Figure 3E) appear greater than Asp levels in "AA5+ rotenone" (Figure 2B). However, only "AA5 + rotenone" rescues proliferation. In other words, the relationship between Asp recovery and proliferation is not clear.

As one further example, in Figure 1E, the authors investigated the M4 aspartate isotopomer derived from U-13C-glutamine and conclude that pyruvate is insufficient to restore progression through the TCA cycle in AA5 treated cells because it resulted in only a 2.8X increase; yet in Figure 2B the authors observed a 2.3X increase in aspartate levels in cells treated with AA5 + rotenone, which leads the authors to conclude that CI and SDH co-inhibition partially restored aspartate levels.

Can the authors more clearly describe how they are determining their thresholds and when (and how much) an increase in Asp is considered noteworthy?

As a result of this uncertainty in the extent of Asp recovery, the proposed rescue of Asp as a factor in facilitating proliferation in SDH and CI inhibition is not convincing. Right now, a key question can't be assessed by the reader: how much Asp is needed for proliferation? The quantitative relationship between Asp levels and proliferation should be made clearer.

2) A full mechanistic model should be clearly presented by the authors and the manuscript would be strengthened if the authors would take the next steps and provide experimental support for potential mechanisms. For example, currently, it is not clear which pathways of Asp synthesis are controlled by mitochondrial NADH (e.g. is reductive carboxylation or PC activity at play here). Moreover, it is not clear in this system, whether Asp regulates proliferation via "reverse" operation of GOT to support biosynthesis or forward "operation" to generate NAD+ for aerobic glycolysis.

More generally, there is a lack of a coherent and consistent explanation for the phenotype and the authors would help the reader if they would present a clear model throughout the manuscript.

3) The data relating NAD+/NADH to the rescue of Asp and proliferation are hard to interpret. On a practical level, the data presentation is not clear and would be easier to interpret if the ion counts for NAD+ and NADH were presented rather than relative ratios.

There are also several conceptual issues with the NAD+/NADH mechanism that need to be cleared up. For example, the data shows that the NAD+/NADH pools are compartmentalized, but their connection to proliferation and aspartate synthesis is not clear from the presented data. Providing additional data would help interpret the role of NAD/NADH in the LbNOX experiments. Are cyto- and mito-LbNox causing the same level of reversal of the NAD+/NADH balance? Is the LbNOX construct just less functional when in the mitochondrial versus the cytosolic form?

Furthermore, the authors should address the variable correlation between the NAD+/NADH ratio and proliferation. For example, in Figure 3G, H, the change in proliferation seems to be relatively minor compared to other proliferation changes presented in this work.

To reiterate the comment above, clarifying and quantifying the mechanistic links between mitochondrial NAD+/NADH and Asp and proliferation is critical if this manuscript is to be published by *eLife*.

4) The data presentation needs to be clearer and to be more consistent. For example, the Asp recovery experiments are sometimes presented using log scales which make very small changes in Asp levels appear much more substantial than they are; linear scales may be more appropriate and easier to interpret. The metabolite measurement data would also be clearer if presented as normalized ion counts rather than relative amounts; the ion counts should at the very least be provided somewhere in the manuscript.

5) In the current version of the manuscript, key data are not always consistent between panels. For example, the recovery of proliferation in SDH inhibited cells by rotenone appears somewhat inconsistent between experiments (e.g. Figure 2A and Figure S3B). As another example, in Figure 1, SDHB expression in UOK269 cells led to a proliferation rate that was nearly 5X higher than the control cells. Yet in Figure 2l, the proliferation rate of UOK269 cells expressing SDHB is the same as those not expressing SDHB. Further still, in 1F, addition of AA5 without PYR leads to minimal change in the NAD+/NADH ratio, while in 3A, sole addition of AA5 roughly doubles the NAD+/NADH ratio. The lack of consistency between panels is a recurrent problem throughout the manuscript and it is critical that the authors address these inconsistencies.

6) As detailed in the individual responses, several panels lack important controls and the appropriate statistical comparisons are not always included.

7) Some of the data included in the manuscript does not add support to the model. For example, data regarding the combinatorial treatment of cells with AA5+AZD7545 (Figures 3G-H) shows only slight changes in proliferation and aspartate levels. This adds little to the argument that mitochondria-specific alterations in NAD+/NADH and aspartate biosynthesis are responsible for proliferation in the context of SDH inhibition.

8) Could the authors address/discuss the physiological relevance of their findings. Do CI subunits exhibit decreased expression in SDH-deficient tumors in patients? Is this a phenomenon of cell culture or is this also observed in vivo? Could more physiological oxygen tension (or even hypoxia) affect SDH-deficient cell proliferation?

*Reviewer #1 (Recommendations for the authors):*

1. The Asp recovery experiments are hard to interpret.

The panels showing Asp levels (e.g. in Figure 1D, 2B, 1G, 3E, 1H) would be easier for the reader to interpret if they were shown with a linear scale on the y-axis. On a linear scale the recovery of Asp by rotenone treatment in SDH inhibited cells (Figure 2B) would look quite minor compared to the loss of Asp that is caused by SDH inhibition. The authors also use different non-linear scales to show Asp levels in different panels of the same figure (e.g. Figure 2G and J), which is especially confusing.

Comparison of Asp rescue between different conditions is also challenging. For example, the recovery of Asp caused by "AA5 + pyruvate" (Figure 1D) appears to be greater than that caused by "AA5 and rotenone" (Figure 2B), yet the recovery of proliferation by "AA5 + pyruvate" is less. As another example, Asp levels in "AA5+mitoLbNOX" (Figure 3E) appear greater than Asp levels in "AA5+ rotenone" (Figure 2B). However, only "AA5 + rotenone" rescues proliferation. In other words, the relationship between Asp recovery and proliferation is not clear.

Part of the problem is that the authors use relative ion counts as well as non-linear scales. They also show the data for closely related conditions in different figures and panels. The best way to resolve the problem would be to present the data as absolute values rather than relative ion counts (e.g. fmol of Asp per cell). If this is not available, the authors should present the (normalized) ion counts (i.e. not the relative ion counts) on a linear scale, with error bars representing the variance of the ion counts. The authors do this in Figure 4B and it is much clearer.

Do the authors have the data from a control sample (143B wild-type cells) that could be included in Figure 4B, so that the reader could see how the rescue of Asp by SDHB KO compares to wild-type levels? Incidentally, Figure 4A would also be easier to interpret if the proliferation rate of wild-type 143B cells were also included.

More generally, if the experiments were performed at the same time, it would be helpful if the data for all the conditions were to be collated in a Sup. Table so that the recovery of Asp between different conditions can easily be compared.

As a result of this uncertainty in the extent of Asp recovery, the proposed rescue of Asp as a factor in facilitating proliferation in SDH and CI inhibition is not convincing. Right now, a key question can't be assessed by the reader: how much Asp is needed for proliferation? The quantitative relationship between Asp levels and proliferation should be made clearer.

2. The data relating NAD+/NADH to the rescue of Asp and proliferation is not entirely convincing and is quite hard to interpret (Figure 1F, G).

i) It is not quite clear what is being measured in Figure 1F. The NAD+/NADH ratio reported in the Figure (i.e. ratio is approx. "1" for controls) is far lower than values that are reported in the literature for this cell line using a different approach (e.g. in the corresponding authors previous work with this cell line) or using carefully validated methods to extract NAD from other cell lines (e.g. by the Rabinowitz group, 2018, Antioxid Redox Signal.). It may be too much to ask the authors to redo the experiments using standards (to provide absolute amounts or concentrations), but it would easier to interpret claims about the role of redox state in these processes easier if the ratio reported here more closely reflects the intracellular NAD+/NADH ratio.

Perhaps the authors are reporting a relative ratio (a ratio of ratios?) that is normalized to the control group? If so, the figure legend isn't clear. Furthermore, having a ratio of ratios is very hard to interpret and can't be compared to previous studies or between experiments in this study. Could the authors please report either the NAD+/NADH ratio that they have measured for each group (on a linear scale), or (better) report the measured values for NAD+ and NADH (e.g. fmol per cell).

ii) Could the authors comment on the lack of an increase in the NAD+/NAD ratio in control cells when they are treated with pyruvate. Isn't this unexpected and inconsistent with previous studies? Several previous studies in very similar systems have all shown an increase in the NAD+/NADH following pyruvate treatment (e.g. Gui et al. 2016 Cell Metabolism), confirming many older studies.

iii) The authors use global NAD+/NADH measurements to infer compartment specific changes. The follow-up experiments using LbNOX are quite an indirect way to check whether this is the case. FLIM would provide some indication if the cytoplasmic and mitochondria pools of NAD(P)H are acting as proposed by the authors.

3. A full mechanistic model is not clearly presented by the authors.

The scheme in Figure 1G presumably doesn't reflect the author's full model, and it would be informative for the authors to present their proposed mechanism somewhere in the manuscript. Right now, it is not clear (for example) how the increase of mitochondrial NADH by dual inhibition of SDH and CI leads to an increase in Asp, or how the rescued Asp levels restore proliferation in this system.

The authors address how mitochondrial NADH increase may increase Asp in the Discussion. However, the manuscript would be strengthened if these questions were addressed experimentally. For example, it is straightforward to test whether reductive carboxylation of glutamine or anaplerosis by pyruvate carboxylase is responsible for the rescue of Asp when both SDH and CI are inhibited.

Furthermore, the link between Asp levels and proliferation has been extensively described in other conditions, but not in this specific condition. Given that the role and "direction" of Asp metabolism varies markedly between different conditions (e.g., Birsoy et al. 2015 vs Gaude et al. 2018) the authors should test which model is operating here. E.g. are MDH1 and GOT1 operating in reverse to generate Asp that is used for biosynthetic reactions, or are GOT1 and MDH1 is operating in the forward direction to consume Asp in order to generate NAD+ for aerobic glycolysis (perhaps more plausible here?).

As a whole the manuscript covers a lot of previously covered ground (e.g. loss of SDH causes loss of CI in tumors (e.g. Fendt group)), loss of SDH and CI creates a requirement for Asp synthesis (Gottlieb group). A clear mechanism of how mitochondrial NADH rescues Asp levels and proliferation would contribute much to its novelty.

4. Key data are not always consistent between panels

For example, the recovery of proliferation in SDH inhibited cells by rotenone appears somewhat inconsistent between experiments (e.g. Figure 2A and Figure S3B). Some variability is to be expected, but, in this version of the manuscript, the experimental variability isn't reflected in the error bars.

5. Are the 143B cells used in these experiments capable of pyrimidine salvage?

It would be useful to know whether the 143B line that is used for most of the experiments is thymidine kinase negative (e.g, the 143B cells from ATCC are TK-). If so, this should be clearly stated in the methods, as lack of thymidine salvage will impact how these results are interpreted.

6. The rescue by AZD7545 does not add support to the model

The rescue of proliferation and especially Asp levels by AZD7545 is so small that if is hard to assess its importance. Is a difference between ~27% and ~31% of wild-type Asp levels really meaningful? When placed on a linear scale it would be hard to discern a difference between Asp levels in these groups.

*Reviewer #2 (Recommendations for the authors):*

1. While quite interesting, the physiological relevance of SDH and ETC complex I inhibition is unclear. Do CI subunits exhibit decreased expression in SDH-deficient tumors in patients? Is this a phenomenon of cell culture or is this also observed in vivo? Could more physiological oxygen tension (or even hypoxia) affect SDH-deficient cell proliferation?

2. The data regarding the importance of aspartate biosynthesis in the context of SDH inhibition are mostly correlative and lack important controls.

a. Figure 1C – How was the dose selected for Asp treatment in the SLC1A3-expressing cells? While the tracing data confirms the increased uptake of Asp relative to non-SLC1A3 expressing cells (S1B-C), the authors do not report Asp levels or indicate how the dosage was selected.

b. Figures 1I and S1K – SDHB expression in UOK269 cells permitted glutamine-dependent aspartate biosynthesis and increased cell proliferation, yet the authors did not report how SDHB expression affected aspartate levels.

c. The authors are inconsistent in their interpretation of meaningful changes in aspartate. For example, regarding Figure 1E, the authors investigated the M4 aspartate isotopomer derived from U-13C-glutamine and conclude that pyruvate is insufficient to restore progression through the TCA cycle in AA5 treated cells because it resulted in only a 2.8X increase; yet in Figure 2B the authors observed a 2.3X increase in aspartate levels in cells treated with AA5 + rotenone, which leads the authors to conclude that CI and SDH co-inhibition partially restored aspartate levels.

d. Is the increase in proliferation upon SDH and CI inhibition dependent on GOT?

3. Overstatements of results reduce clarity of the findings. For example:

a. The authors argument that "SDH inhibition blocks proliferation, which is restored by aspartate but not electron acceptors" (Figure 1 title) seems to be an overstatement. Though aspartate seems to increase cell proliferation to a greater degree than electron acceptors, data presented in 1A and 1C indicate that pyruvate, AKB, and LbNOX are all sufficient to increase cell proliferation.

b. The presented correlations between aspartate levels and proliferation are not sufficient evidence that aspartate biosynthesis is the primary mode of supporting cell proliferation in SDH-deficient cells. For example, in Supp. Figure 2B-C aspartate levels are not rescued to the same extent as proliferation across the panel of cell lines.

c. Data regarding the combinatorial treatment of cells with AA5+AZD7545 (Figures 3G-H) shows only slight changes to proliferation and aspartate levels. This adds little to the argument that mitochondria-specific alterations in NAD+/NADH and aspartate biosynthesis are responsible for proliferation in the context of SDH inhibition.

4. Statistical analysis was frequently missing or improperly conducted. For example, regarding panel 1A the authors noted that AA5 and rotenone treatment decreased cell proliferation and that co-treatment with pyruvate, AKB, LbNox, or Asp restored proliferation. Statistical comparisons were made between AA5 and rotenone for all groups; however, based on the text, the statistical analysis should be restricted to comparisons between groups for each AA5 and rotenone.

5. The data presentation throughout the manuscript is confusing. For example, in some panels the y-axis is log-scale while in complimentary panels the y-axis is linear, and for some graphs the y-axis does not start at 0.

6. In Figure 1, SDHB expression in UOK269 cells led to a proliferation rate that was nearly 5X higher than the control cells. Yet in Figure 2l, the proliferation rate of UOK269 cells expressing SDHB is the same as those not expressing SDHB. Why are the proliferation rates different?

*Reviewer #3 (Recommendations for the authors):*

During our reading of the manuscript, we felt the model the authors proposed within this work was not consistent or clear throughout the text. We feel that some attention could be made to try to emphasize and clarify this model throughout and would greatly assist other readers of the work. Currently, we feel like we as readers had to do a lot of work to try to make sense of the model and how the data are supporting that model.

The authors state that "this work provides the first mechanistic evidence that CI loss can directly support core metabolic functions":

1) We do not feel the mechanistic data provided in the current form of the manuscript are strong enough to make this claim.

2) Perhaps this is the first instance of loss of CI & II together are beneficial, but there seem to be several other works that seem to make a case that loss of CI is beneficial to core metabolism processes, such as:

a. PMID: 29915083

b. PMID: 18784283

While we appreciate that many of the scripts used to generate data within this work are using pre-existing toolkits, it is still important to present the code used (how the toolkits were used exactly) to aid in the reproducibility of the work. These scripts could be included in the supplement, or hosted on a public repository, such as GitHub. Raw and processed metabolomics data should also be included as supplements, or hosted on a public repository, such as Metabolomics Workbench.

In Figure S4F, the authors state what "unsupervised clustering" was used. This is very generic – what was the exact algorithm used in the hierarchal clustering of the proteomics data and why? For example, scipy.cluster.hierarchy.linkage provide 7 different clustering methods (https://docs.scipy.org/doc/scipy/reference/generated/scipy.cluster.hierarchy.linkage.html)

There are also a couple of changes to the presentation of the figures that would improve the manuscript:

– Figure 3B/F: There is a line showing up at the top of each of these subpanels in our version of the manuscript.

–

This is a very minor point, but perhaps bar plot colors could be more standardized between sub-panels for the different conditions?

In summary, while we do not feel like this manuscript in its current form meets the criteria for publication in *eLife*, we are certainly positive about its potential and suggest filling in some of the remaining gaps outlined above and some mechanisms to explain the observations would make the work more complete and suitable for publication.

---

## [Author Response]

Essential revisions:1) The reviewers feel that the data linking Asp to rescue of proliferation are mostly correlative. Much more evidence is needed to conclude that the small changes in aspartate abundance are sufficient to restore proliferation and the presented correlations between aspartate levels and proliferation are not sufficient evidence that aspartate biosynthesis is the primary mode of supporting cell proliferation in SDH-deficient cells.Similarly, comparison of Asp rescue between different conditions is also challenging and the authors are inconsistent in their interpretation of meaningful changes in aspartate. For example, the recovery of Asp caused by "AA5 + pyruvate" (Figure 1D) appears to be greater than that caused by "AA5 and rotenone" (Figure 2B), yet the recovery of proliferation by "AA5 + pyruvate" is less. As another example, Asp levels in "AA5+mitoLbNOX" (Figure 3E) appear greater than Asp levels in "AA5+ rotenone" (Figure 2B). However, only "AA5 + rotenone" rescues proliferation. In other words, the relationship between Asp recovery and proliferation is not clear.As one further example, in Figure 1E, the authors investigated the M4 aspartate isotopomer derived from U-13C-glutamine and conclude that pyruvate is insufficient to restore progression through the TCA cycle in AA5 treated cells because it resulted in only a 2.8X increase; yet in Figure 2B the authors observed a 2.3X increase in aspartate levels in cells treated with AA5 + rotenone, which leads the authors to conclude that CI and SDH co-inhibition partially restored aspartate levels.Can the authors more clearly describe how they are determining their thresholds and when (and how much) an increase in Asp is considered noteworthy?As a result of this uncertainty in the extent of Asp recovery, the proposed rescue of Asp as a factor in facilitating proliferation in SDH and CI inhibition is not convincing. Right now, a key question can't be assessed by the reader: how much Asp is needed for proliferation? The quantitative relationship between Asp levels and proliferation should be made clearer.

We thank the reviewers for bringing up this potential source of misunderstanding related to the relationship between aspartate levels and cell proliferation. We agree that, as previously written, the relationship between the relative levels of aspartate and the corresponding expectation for proliferation effects was not clearly described, leading to confusion about what fold change thresholds are considered meaningful. Compounding this issue was some variability in the LCMS methods used in the early stages of this project that occasionally distorted relative levels, leading to some inconsistencies in the actual numbers generated across experiments (e.g. when samples were too concentrated, aspartate detection could fall outside the linear range for our mass spectrometer, affecting relative levels). While the qualitative conclusions of the manuscript were not affected by those issues, we agree that describing a consistent quantitative relationship between aspartate levels and proliferation would improve the manuscript. To address these important issues, we have (1) improved our LCMS protocols and repeated many experiments and (2) conducted experiments to directly describe the quantitative relationship between ASP levels and proliferation during mitochondrial inhibition.

1) Standardizing aspartate quantification:

To ensure that we are accurately measuring changes in aspartate levels, we first standardized our protocol to ensure uniform cell material inputs by normalizing metabolite extract loading to total cell volume. Cell volumes were measured from parallel wells at the time of extraction using a Coulter Counter, which we have found to be a preferred method for calculating the cell volume from adherent cells (DOI: 10.21769/BioProtoc.4216). For each condition, the volume of metabolite extract containing 1 uL of total cell volume was calculated, aliquoted into new tubes, dried, and resuspended in 150 uL of solvent prior to running. Next, we incorporated an isotopically labeled amino acid mix containing aspartate at 1 μm into the resuspension solvent to serve as an internal standard. Finally, we also measured the linearity of aspartate detection across a standard curve of isotopically labeled amino acid mix. Importantly, we found that aspartate ion counts had a linear relationship with the loaded concentration of standard between 1 x 10^6^ and 5 x 10^9^ ions, and possibly higher (Author response image 1). Using the standardized protocol described here, we also found that the ion counts of aspartate measured from cells cultured in representative conditions from this study was between 1 x 10^7^ and 1 x10^9^ ions, well within the linear range for linear aspartate quantification. Collectively these data support the conclusion that this method allows for accurate measurements of relative aspartate levels across conditions (Author response image 1). The methods section was edited to describe these methodological differences (see: “liquid chromatography-mass spectrometry” section).

**Author response image 1. sa2fig1:** Quantitative analysis of aspartate detection by LCMS. (A) Detection of isotopically labeled aspartate by LCMS across a standard curve of aspartate levels. (B) Measurement of aspartate from 143B cells treated with vehicle, 50 nM rotenone, or 5 µM AA5, with or without 1 mM pyruvate in DMEM for 6 hours. Aspartate ion counts are within the linear range for aspartate detection.

The presentation of aspartate quantification figures was also edited to improve clarity. Including changing to a linear Y axis (with break) instead of a log axis. The linear Y axis also “fold change” values unnecessary and so were removed, since they were an additional source of confusion.

2) Quantifying the relationship between aspartate levels and proliferation:

A critical conclusion from this study is that aspartate levels correlate with, and determine, the proliferation rate across conditions of SDH inhibition. This conclusion is reached by two fundamental observations: (1) The proliferation defects after SDH inhibition are ameliorated by aspartate restoration (Revised Manuscript Figures 1A, 1G, 2F, 4C, 4H, 5D, 6D, 6H, 7B, and Figure 1—figure supplement 1D, 1G, Figure 2—figure supplement 2H, 2K, and Figure 7—figure supplement 1D). (2) The proliferation rate upon SDH inhibition (and related cotreatments) directly correlates with aspartate levels. However, as noted by the reviewers, the relative levels of aspartate do not necessarily correlate linearly with proliferation levels, since a small increase in aspartate can have a large effect on proliferation when cells are proliferating slowly, leading to confusion about what is a meaningful change. To address the fundamental relationship between aspartate levels and proliferation, we generated a standard curve comparing the levels of aspartate and the corresponding proliferation rate in conditions well characterized to cause aspartate limitation – treatment with an ETC complex I inhibitor (PMIDs: 27746050, 26232225, 29941931, 26232224, 29941933, 36411320, 29892070). Indeed, when cultured in DMEM without pyruvate across eight doses of rotenone (0, 0.1, 1.25, 3.125, 6.25, 12.5, 25, 50 nM) cells showed a dose dependent decrease in cell proliferation that correlated with a dose dependent decrease in aspartate levels (Revised Manuscript Figures 1C, Figure 1—figure supplement 1K-L see below). Interestingly, the relationship between aspartate levels and proliferation rate followed a hyperbolic curve, where relatively small increases in aspartate levels cause a large change in proliferation rate when aspartate is very low, and only a minor change in proliferation rate when aspartate is near untreated levels. This curve is consistent with a model where cells have sufficient aspartate when untreated and decreasing aspartate does not immediately cause robust cell proliferation defects until it becomes limiting for a downstream metabolic process (or processes). Our data indicate that, relative to the aspartate levels present in untreated 143B cells (grown in DMEM with 1 mM pyruvate), aspartate only becomes a significant limitation for cell proliferation once depleted to around 0.4, and is unable to support proliferation around 0.05, with a strong correlation between proliferation and aspartate levels between 0.05-0.4. Culturing cells treated with high dose rotenone (50 nM) in media containing the electron acceptor pyruvate restored both aspartate levels and proliferation rate to values consistent with the measured proliferation rate to aspartate levels curve. Finally, we also included measurements of proliferation rate and aspartate levels upon AA5 treatment, with or without pyruvate in the media, and found that they too matched the values observed in rotenone treated cells. Together, these data provide detailed quantitative information about the relationship between aspartate levels and cell proliferation in the context of complex I and SDH inhibition, providing considerable support for our claim that small changes in aspartate levels can impact cell proliferation rate in SDH impaired cells and contextualizing aspartate level changes throughout the study

2) A full mechanistic model should be clearly presented by the authors and the manuscript would be strengthened if the authors would take the next steps and provide experimental support for potential mechanisms. For example, currently, it is not clear which pathways of Asp synthesis are controlled by mitochondrial NADH (e.g. is reductive carboxylation or PC activity at play here). Moreover, it is not clear in this system, whether Asp regulates proliferation via "reverse" operation of GOT to support biosynthesis or forward "operation" to generate NAD+ for aerobic glycolysis.More generally, there is a lack of a coherent and consistent explanation for the phenotype and the authors would help the reader if they would present a clear model throughout the manuscript.

We concur with the reviewers’ suggestion that identifying the metabolic enzymes and pathways required for aspartate synthesis is a reasonable and important next step for understanding of this metabolic phenotype and for generating a clear mechanistic model. To address these goals we (1) Conducted stable isotope resolved metabolomics from U-^13^C glucose and U-^13^C glutamine to ascertain the source of aspartate in each condition; (2) Used genetic knockouts and metabolic inhibitors to identify key pathway components of aspartate synthesis in SDH impaired cells; (3) Generated several model figures to clarify our metabolic model(s) of our findings. These experiments generated a significant amount of new data, necessitating the addition of three new figures, each of which has a linked supplemental figure (Revised Manuscript Figures 4, 5, and 6 and Revised Manuscript Supplemental Figures 4, 5, and 6), as well as seven additional model panels.

1) Stable isotope resolved metabolomics.

Three primary paths of aspartate synthesis have been described in mammalian cells – the oxidative TCA cycle (primarily deriving from glutamine), reductive carboxylation of α-ketoglutarate (primarily deriving from glutamine, “RCQ”), and pyruvate carboxylation (primarily deriving from glucose, “PC”). Since the oxidative TCA cycle is blocked in SDH impaired cells, it raises the question of which metabolic source contributes to increased aspartate levels when complex I is co-inhibited. We first conducted isotope tracing from wild type (WT) or SDHB KO cells cultured in U-^13^C glutamine for 6 hours, with or without rotenone treatment (Revised Manuscript Figure 4A and B). As expected, glutamine was the predominant carbon source for aspartate in wild type cells, and rotenone predictably decreased aspartate levels and processive TCA cycling, based on M+2 label. Additionally, rotenone treatment increased aspartate levels in SDHB KO cells, as expected, with the only major aspartate isotopotologue species from glutamine being M+3, consistent with being derived from RCQ. While the abundance of M+3 aspartate was increased upon rotenone co-treatment in SDHB KO cells, the total amount of labeled carbon from glutamine remained a relatively small contributor to total aspartate levels in both cases. These data confirm that oxidative TCA cycling is impaired in SDHB KO cells and suggest that RCQ is a modest contributor to increased aspartate upon complex I inhibition.

We also conducted isotope tracing in identical conditions using U-^13^C glucose. Since unlabeled pyruvate in the media could contaminate the pyruvate labeling from U-^13^C glucose. we cultured cells in media containing 1 mM AKB, which had similar effects on proliferation as 1 mM PYR (Figure 1A). We also confirmed that rotenone co-treatment still restored cell proliferation and aspartate levels to cells cultured in AKB media instead of PYR media (Revised Manuscript Figure 4—figure supplement 1A and B, below). Consistent with the aforementioned glutamine labeling results, WT cells had minimal labeling from glucose and the small contribution oxidative acetyl-CoA production and processive TCA cycling (M+2) that was diminished by rotenone cotreatment (Revised Manuscript Figure 4E and F, above). However, also consistent with glutamine labeling results, glucose was a dominant contributor to aspartate carbon in SDHB KO cells, with the increase in aspartate levels upon rotenone treatment primarily having M+3 label from U-^13^C glucose. These data suggest that PC is a major contributor to aspartate levels in 143B SDHB KO cells, consistent with results from prior work (Cardaci et al., 2015), and that complex I inhibition can enhance this aspartate synthesis route in SDH impaired cells.

To corroborate these findings in the patient-derived SDHB mutant UOK269 cells without (WT UOK269) or with SDHB re-expression (SDHB UOK269), we also added U-^13^C glucose tracing to the U-^13^C glutamine isotope tracing conducted in the prior version of the manuscript. As in 143B cells, we found that SDH loss was associated with an increased contribution of glucose to aspartate by PC (M+3) and a switch of glutamine contribution to aspartate from the oxidative TCA cycle (M+4) to RCQ (M+3) (Revised Manuscript Figure 8B and C).

1) Genetic knockouts and metabolic inhibitors

a. Dependence on RCQ and PC

As suggested by the reviewers, we investigated which components of alternative aspartate synthesis pathways were required for CI inhibition to drive aspartate synthesis in SDH impaired cells – evaluating contributions from RCQ, PC, GOT enzymes, and related metabolic enzymes. We first evaluated the relative contributions of RCQ and PC to cell proliferation and aspartate levels in SDH impaired cells, with and without CI co-inhibition. To test the dependence of RCQ we leveraged BMS-303141 (BMS) an inhibitor of ACLY, a shared reaction for the RCQ pathway to aspartate regardless of whether RCQ occurs in the mitochondria by IDH2 or the cytosol by IDH1 (See Revised Manuscript Figure 4—figure supplement 1A, below, for metabolic model). We found that BMS treatment caused minor cell proliferation defects and lowered aspartate levels in SDH/CI dual-inhibited cells, and that this minor proliferation defect could be remedied by aspartate co-treatment (Revised Manuscript Figure 4C and D, above). Notably, these effects also track with the minor contributions of RCQ to aspartate synthesis, measured by isotope tracing. We next evaluated the dependence on PC, using CRISPR/cas9 to generate isogenic knockout clones for PC, comparing to paired “Addback” (AB) clones Revised Manuscript Figure 4G, above. We prefer to compare KO to AB cell lines for each gene knock out rather than to parental WT cells, as it better controls for any unrelated phenotypic changes that can result from the process of single cell cloning. As expected, PC KO sensitized cells to the proliferative and aspartate depleting effects of AA5 treatment and somewhat decreased the corresponding proliferative benefits of rotenone co-treatment (Revised Manuscript Figure 4H and I, above). To test whether residual or compensatory RCQ activity was contributing to the enduring aspartate and proliferation rescue in PC KO cells, we also co-treated with BMS and ASP. Importantly, the combination of PC KO and BMS treatment abolished the proliferation of SDH/CI impaired cells, which could be restored by aspartate co-treatment, indicating that the loss of those pathways prevents cell proliferation in SDH/CI impaired cells by decreasing aspartate availability. Correspondingly, while not quite statistically significant, BMS treatment mitigated the increase of aspartate upon rotenone co-treatment in PC KO cells, with the partial effect in PC AB cells mirroring that of wild type cells in Figure 4D. Collectively, these experiments reveal that RCQ and PC are essential contributors to the induction of aspartate synthesis in SDH impaired cells upon CI co-inhibition (Revised Manuscript Figure 4J, above).

b. Dependence on mitochondrial pyruvate import

While not explicitly requested by the reviewers, we sought to dive deeper to mechanistically understand how compartmentalized redox changes could drive alternative aspartate synthesis via RCQ and PC. While PC is conventionally reported to be active in the mitochondria, recent reports have concluded that it may have some cytoplasmic activity (PMID: 34547241) and so we investigated whether mitochondrial pyruvate entry was required for in the induction of aspartate and proliferation upon CI inhibition in SDH impaired cells. To do so, we generated knockout and AB cells for the essential mitochondrial pyruvate carrier subunit, MPC1. MPC1 KO and AB were verified by western blot (Revised Manuscript Figure 5A). We then tested the effects of AA5 with or without rotenone and found that MPC1 KO cells had severe proliferation defects upon AA5 treatment that were not rescued by rotenone co-treatment (Revised Manuscript Figure 5B). Importantly aspartate levels corroborated this result since rotenone co-treatment in AA5 treated MPC1 KO cells had minimal ability to increase aspartate levels (Revised Manuscript Figure 5C). Once again, this inability to increase aspartate upon rotenone treatment was functionally limiting in AA5 and AA5/rotenone treated MPC1 KO cells since aspartate treatment could still restore their cell proliferation (Revised Manuscript Figure 5D). Notably, the loss of mitochondrial pyruvate uptake not only decreases pyruvate for potential PC reactions, but also pyruvate for transamination by GPT2, which generates alanine from pyruvate while converting glutamate to AKG – a necessary substrate for RCQ driven aspartate synthesis. In support of this hypothesis, MPC1 KO cells were associated with depletion of both alanine and AKG in both vehicle and AA5 treated conditions (Revised Manuscript Figures 5E and Figure 5—figure supplement 1B). Notably, AKG levels (but not alanine levels) were further depleted in MPC1 KO cells when co-treated with rotenone, suggesting that residual GDH activity provides some AKG production in MPC1KO cells, which is blocked upon NAD+ depletion by rotenone treatment. Together, these data identify a novel metabolic dependency of SDH deficient cancer cells on MPC that supports both pathways of alternative aspartate synthesis.

c. Dependence on compartmentalized GOT enzymes

Aspartate is generated from oxaloacetate, using glutamate as a nitrogen donor, through either the cytosolic transaminase GOT1 or the mitochondrial transaminase GOT2. Recent studies have found that GOT2 generates aspartate during canonical oxidative TCA cycle metabolism and in PDAC cells cultured in hypoxia (PMID: 35726024), whereas GOT1 can generate aspartate during other forms of mitochondrial dysfunction (PMID: 26232224). We again turned to CRISPR/Cas9 to generate knockout clones for GOT1 or GOT2 and to compare the effects of AA5 with or without rotenone co-treatment, comparing KO cell lines to their restored counterparts (AB). We first tested the effects of GOT2 KO our phenotypes. GOT2 KO and AB were confirmed by western blot and functionally, since GOT2 KO cells cannot grow without pyruvate, as has been recently published (Revised Manuscript Figures 6A and Figure 6—figure supplement 1A) (PMID: 35815941). While GOT2 loss caused an increase in sensitivity to AA5, it had essentially no effect on the proliferation restoration from rotenone co-treatment when compared AB cell lines (Revised Manuscript Figure 6B). Measurement of aspartate levels corroborated these results, where aspartate levels were restored by rotenone in both GOT2 KO and AB cell lines (Revised Manuscript Figure 6C). Finally, the proliferative defects from AA5 treatment in GOT2 KO cells was found to result from aspartate deficiency, as treatment with aspartate could restore proliferation to AA5 treated cells to a similar degree as rotenone co-treatment (Revised Manuscript Figure 6D). We then generated KO and AB cell lines for GOT1, which were confirmed by western blot, and then conducted similar experiments (Revised Manuscript Figure 6E). Notably, GOT1 KO cells did not have notable sensitivity to AA5 treatment, consistent with their dependency on GOT2 in this condition, but the proliferation benefit from rotenone co-treatment was abolished in the absence of GOT1 (Revised Manuscript Figure 6F). Measuring aspartate levels similarly showed that the increase in aspartate levels upon rotenone co-treatment was absent in AA5 treated cells without GOT1, commensurate with their proliferation phenotype (Revised Manuscript Figure 6G). Interestingly, we noted that this defect in aspartate production was downstream of the increased anaplerosis following CI inhibition in SDH impaired cells, as malate levels still increase upon CI/SDH co-inhibition in GOT1 KO cells (Revised Manuscript Figure 6—figure supplement 1B). Nonetheless, the impaired ability to synthesize aspartate was functionally limiting for proliferation since aspartate treatment could still restore the proliferation to AA5/rotenone treated GOT1 KO cells (Revised Manuscript Figure 6H). Finally, we corroborated the finding that cells dependent on GOT1 for aspartate synthesis also depend on MDH1 as well, presumably to generate cytosolic OAA for GOT1 dependent aspartate synthesis (Revised Manuscript Figure 6—figure supplement 1D-F and Figure 6I).

Collectively, the above experiments indicate that the increased aspartate levels and proliferation upon complex I inhibition in SDH impaired cells arise primarily from mitochondrial redox changes driving pyruvate carboxylation and, to a lesser degree, reductive carboxylation of glutamine, both of which depend on mitochondrial pyruvate, to generate cytosolic OAA to ultimately being converted to aspartate in the cytosol via GOT1.

1) Metabolic models

As recommended by the reviewers, we have incorporated our data into seven additional model figure panels to help communicate our understanding of the metabolic pathways and phenotypes relevant to SDH deficient cells, see Revised Manuscript Figures 2H, 3G, Figure 4—figure supplement 1A, Figure 5—figure supplement 1A, Figure 6—figure supplement 1C, 7J, and 8H, (new models for Main Figures 4, 5, and 6 are shown above).

3) The data relating NAD+/NADH to the rescue of Asp and proliferation are hard to interpret. On a practical level, the data presentation is not clear and would be easier to interpret if the ion counts for NAD+ and NADH were presented rather than relative ratios.There are also several conceptual issues with the NAD+/NADH mechanism that need to be cleared up. For example, the data shows that the NAD+/NADH pools are compartmentalized, but their connection to proliferation and aspartate synthesis is not clear from the presented data. Providing additional data would help interpret the role of NAD/NADH in the LbNOX experiments. Are cyto- and mito-LbNox causing the same level of reversal of the NAD+/NADH balance? Is the LbNOX construct just less functional when in the mitochondrial versus the cytosolic form?Furthermore, the authors should address the variable correlation between the NAD+/NADH ratio and proliferation. For example, in Figure 3G, H, the change in proliferation seems to be relatively minor compared to other proliferation changes presented in this work.To reiterate the comment above, clarifying and quantifying the mechanistic links between mitochondrial NAD+/NADH and Asp and proliferation is critical if this manuscript is to be published by eLife.

We are glad to have the opportunity to further clarify the connection between (compartmentalized) NAD+/NADH, aspartate levels, and cell proliferation. The comments raise two general issues, each of which will be addressed in turn: (1) Ion counts vs relative ratios, (2) Evidence for compartmentalized redox changes in the comments about Original Submission Figures3G and H will be incorporated into the discussion about those panels below.

1) NAD+/NADH ion counts, ratios, and relative ratios.

NAD+/NADH is a compound measurement of synthesis pathways of NAD(H) and the redox reactions that govern the NAD+/NADH ratio, which in turn affects other reactions that use those coenzymes. We have previously found that, in the context of mitochondrial inhibition, treatments that alter NAD+ and NADH pool sizes can alter NAD+/NADH ratios and dependent metabolic reactions, but that the ratio change from this approach is temporary as the ratio adjusts to the larger total pool size and that the resulting ratio ultimately governs the phenotypic effects on proliferation (PMID: 27746050). Thus, we believe that displaying the NAD+/NADH ratio is still the most relevant output for understanding changes in redox status and metabolism, rather than NAD+ or NADH ion counts individually. Nonetheless, in the interest of full transparency we have added all ion counts for NAD+ and NADH to Supplementary File 1 for easy reference.

We do however sympathize with the issue reviewers raise regarding to “relative ratios,” where we normalized the ratio of ion counts for NAD+ and NADH ratio from each sample to that of the vehicle treated cells. The reason we used this approach in the prior version is that, because standards for isotopically labeled NAD+ and NADH are not widely available, LCMS ion counts for NAD+ and NADH do not incorporate differences in matrix effects or ionization efficiencies for each analyte (and so are not comparable from an absolute quantitation perspective) and thus do not constitute a “true molar” NAD+/NADH ratio. We thought that, by normalizing samples it might decrease the likelihood that these data would be misperceived to a quantitative representation of the true cellular NAD+/NADH (which we have found to be a common misunderstanding). Nonetheless, we agree that is confusing to present a “relative ratio” and so we have changed all LCMS derived NAD+/NADH measurements to a simple ratio of ion counts, changing the Y axis legend to include “a.u./a.u.” to hopefully make this caveat apparent, as below:

In addition, we added the following text to the “Liquid Chromatography-Mass Spectrometry (LCMS)” section of the Materials and methods, to describe these caveats:

“Isotopically labeled standards for NAD+ and NADH are not widely available, so LCMS experiments measuring NAD+/NADH ratios do not incorporate differences in ionization efficiencies or matrix effects for each analyte. Thus, while changes in the ratio of measured ion counts for NAD+ and NADH reflect biological changes, they should not be interpreted as an absolute intracellular ratio, which is more accurately quantified by enzymatic methods. Individual ion counts for NAD+ and NADH are available in Supplementary File 1.”

2) Compartmentalized measurements of redox changes

Our data indicate that decreases in mitochondrial NAD+/NADH are required for the aspartate synthesis and proliferative benefits of complex I inhibition in SDH impaired cells, as evidenced by the result that these benefits can be blunted by expression of mitoLbNOX, but not cytoLbNOX. The lack of benefit for cytoLbNOX was expected since these cells are cultured in the presence of media pyruvate, which makes cytoLbNOX redundant for maintaining cytosolic NAD+/NADH. To underscore this experimental detail, we edited the text to make this point more clearly:

“Notably, since these cells are cultured in the presence of the cytosolic electron acceptor pyruvate, we thus expected cyto*LbNOX* to be redundant and ineffective to change cell metabolism, whereas mitoLbNOX would be positioned to disrupt the mitochondrial redox changes from rotenone treatment. Indeed, whereas cells expressing cyto*LbNOX* behaved similarly to parental cells expressing eGFP, mito*LbNOX* expression impeded the restorative effect of rotenone treatment on SDH-impaired cell proliferation (Figure 3E).”

Nonetheless the reviewers raise a good point that since changes in mitochondrial compartmentalized NAD+/NADH are concluded to be the driver of our phenotype, it would be reasonable to measure them directly. To do so, we combined a recently published rapid subcellular enrichment protocol for metabolomics (Lee et al., 2019) with an enzymatic NAD+/NADH assay that allows measurements on NAD+/NADH from low sample input (protocol optimized in Sullivan et al., 2015), allowing us to separately measure NAD+/NADH from the cytosolic and mitochondrial fractions. We found that in basal conditions, as expected, cytosolic NAD+/NADH was higher than mitochondrial NAD+/NADH, indicating that the assay was likely reflecting endogenous redox differences between compartments (Revised Manuscript Figure 3B). Notably, mitochondrial inhibitors only had small effects on cytosolic NAD+/NADH, which was expected in this experimental context since exogenous pyruvate in the media can fix the cytosolic NAD+/NADH ratio through lactate dehydrogenase. We did however observe large changes in mitochondrial NAD+/NADH, where rotenone lowered the ratio, AA5 increased the ratio, and rotenone was epistatic to AA5 with regards to the effects on mitochondrial NAD+/NADH in co-treated cells. The disproportionate amount of NADH in the mitochondrial compartment thus likely drives the ”whole cell” NAD+/NADH changes observed by LCMS in the Revised Manuscript Figures 1D, 3A, and 7D, We believe these data provide important corroborative evidence for the conclusion of this figure that mitochondrial redox changes are required for the benefits of CI inhibition in SDH impaired cells.

4) The data presentation needs to be clearer and to be more consistent. For example, the Asp recovery experiments are sometimes presented using log scales which make very small changes in Asp levels appear much more substantial than they are; linear scales may be more appropriate and easier to interpret. The metabolite measurement data would also be clearer if presented as normalized ion counts rather than relative amounts; the ion counts should at the very least be provided somewhere in the manuscript.

We thank the reviewers for this suggestion and opportunity to improve the interpretability of the manuscript. Since SDH loss causes a severe depletion in aspartate by ~90% to a regime where small changes in aspartate levels can have large changes in proliferation rate (Revised Manuscript Figure 1C), we initially thought that a continuous log axis would be a fairer depiction of the data. It was certainly not our intention to mislead the readers and so we are happy to change the presentation to avoid that impression. We also recognize that log scales can be unintuitive and so, in accordance with the reviewers’ suggestion, we have changed all the metabolite graphs to linear scales, with appropriate line breaks to allow the reader to easily see relevant changes in aspartate levels when it is decreased. We still believe that relative ion counts are easier for reader interpretation compared to raw ion counts, particularly in light of the addition of Revised Manuscript Figure 1C that should help contextualize the relationship between relative aspartate levels and proliferation rates throughout the manuscript. Nonetheless, we have further clarified the quantification details by making the Y axis labels more descriptive (to make it easier to understand the condition for which all other metabolite levels are made relative) and by describing the metabolite quantification approach in detail in the methods section. Finally, to assuage any further concerns, we have included all ion counts used to generate the relative metabolite level graphs in Revised Manuscript Supplementary File 1.

5) In the current version of the manuscript, key data are not always consistent between panels. For example, the recovery of proliferation in SDH inhibited cells by rotenone appears somewhat inconsistent between experiments (e.g. Figure 2A and Figure S3B). As another example, in Figure 1, SDHB expression in UOK269 cells led to a proliferation rate that was nearly 5X higher than the control cells. Yet in Figure 2l, the proliferation rate of UOK269 cells expressing SDHB is the same as those not expressing SDHB. Further still, in 1F, addition of AA5 without PYR leads to minimal change in the NAD+/NADH ratio, while in 3A, sole addition of AA5 roughly doubles the NAD+/NADH ratio. The lack of consistency between panels is a recurrent problem throughout the manuscript and it is critical that the authors address these inconsistencies.

We apologize for this misunderstanding. While some variability in proliferation rates is a natural feature of conducting these experiments, the reviewers astutely noted several situations where the variability could be misinterpreted as potentially suggesting non-reproducible results, which is not the case. We have repeated each of the indicated experiments (and many others) and replaced them as necessary to ensure that the most representative experimental results are displayed.

However, we believe the discrepancies noted by the reviewers for UOK269 cells and NAD+/NADH relate to confusion about the media conditions used for each experiment, rather than inconsistent data. For example, the “5X higher” proliferation rate is true if comparing the proliferation rates of WT UOK269 cells in pyruvate-free DMEM (Original Submission Figure 1I/Revised Manuscript Figure 1G) to those cultured in DMEM with 1 mM pyruvate (Original Submission Figure 2I/Revised Manuscript Figure 8E). However, if the reviewers compare the proliferation rates of each cell line in the +PYR condition, they will see they are similar across figures. Likewise, the NAD+/NADH measurements of Original Submission 1F/Revised Manuscript Figure 1D are actually well in line with the data in Original Submission 3A/Revised Submission 3A, if comparing the +PYR conditions in the Figure 1 panel to those in Figure 3, which are from cells cultured in pyruvate containing media. Nonetheless, this perception of inconsistent results was our fault because we did not delineate experimental details clearly enough to guide the reader to the correct comparisons across figure panels. To address this issue, we have made several edits throughout the manuscript to remind the reader that all experiments after Figure 1 are conducted in media containing an electron acceptor (Pyruvate or AKB, as indicated).

Examples:

1) “We tested if, in the presence of electron acceptors, rotenone could alter the proliferation of cells treated with AA5 and, surprisingly, found it caused a robust restoration of cell proliferation (Figure 2A).”

2) Manuscript text edits highlighted in regards to “Compartmentalized measurements of redox changes,” (above):

“To avoid potential caveats that may confound labeling patterns from unlabeled media pyruvate, we conducted U-13C glucose isotope tracing experiments in pyruvate-free DMEM with AKB, which serves a similar electron acceptor function as pyruvate but cannot fulfill its carbon fates (Figure 1A) (Altea-Manzano et al., 2022; Sullivan et al., 2015). We also confirmed that rotenone is still effective to restore proliferation and aspartate levels of SDH-impaired cells in pyruvate-free DMEM with AKB (Figure 4—figure supplement 1B, 1C).”

3) In the discussion:

“Additionally, we note that SDH/CI impaired cells also require a supply of cytosolic electron acceptors to proliferate, indicating that this metabolic state depends on intercompartmental metabolic differences including a reduced mitochondrial compartment and an oxidized cytosolic environment.”

In addition, we have revised every figure legend for which the presence or absence of an electron acceptor (i.e. pyruvate or AKB) is a relevant variable for the experimental output to include the media condition explicitly. We hope these edits will help dispel any potential interpretation of variable results and aid in interpretation of our findings.

6) As detailed in the individual responses, several panels lack important controls and the appropriate statistical comparisons are not always included.

As suggested, we have repeated experiments to include important controls and have added the appropriate statistics corresponding to the individual reviewer comments.

7) Some of the data included in the manuscript does not add support to the model. For example, data regarding the combinatorial treatment of cells with AA5+AZD7545 (Figures 3G-H) shows only slight changes in proliferation and aspartate levels. This adds little to the argument that mitochondria-specific alterations in NAD+/NADH and aspartate biosynthesis are responsible for proliferation in the context of SDH inhibition.

We appreciate the reviewers concern for unnecessary data, but we respectfully disagree that these data do not add support to the model. While our mitoLbNOX data indicate that decreases in mitochondrial NAD+/NADH are required for the aspartate production and proliferative benefits of CI inhibition in SDH impaired cells, they do not necessarily confirm that decreased mitochondrial NAD+/NADH is sufficient to increase aspartate and proliferation in the absence of a CI inhibitor. We believe that the fact that AZD7545 increases aspartate and cell proliferation without blocking oxygen consumption provides evidence that decreased mitochondrial NAD+/NADH is both required and sufficient to induce aspartate and proliferation, without inherently requiring an ETC inhibitor. Nonetheless we agree that the very small effect sizes may confuse readers. So, we have de-emphasized these data in the revised manuscript by moving them to the supplemental figures (Revised Manuscript Figure 3—figure supplement A-E) and edited the manuscript to more accurately describe the effect sizes and clarify our interpretation:

“Importantly, treatment with AZD7545 also slightly improved cell proliferation and increased aspartate levels in AA5 treated cells (Figure 3—figure supplement 2D, 2E). Notably, these effects are likely limited by the fact that they are occurring in the presence of intact CI activity, which would likely buffer major changes in mitochondrial NAD+/NADH. Nonetheless, these data provide additional support that compartment-specific mitochondrial redox alterations are both sufficient and required to reprogram aspartate synthesis in SDH-impaired cells (Figure 3G).”

8) Could the authors address/discuss the physiological relevance of their findings. Do CI subunits exhibit decreased expression in SDH-deficient tumors in patients? Is this a phenomenon of cell culture or is this also observed in vivo? Could more physiological oxygen tension (or even hypoxia) affect SDH-deficient cell proliferation?

We thank the reviewers for bringing up this important point. Due to the rarity of these tumors, acquiring sufficient patient samples to analyze expression patterns of CI subunits on SDH-deficient tumors is difficult at the current moment. We did however seek to increase the physiological relevance of our findings, as suggested by the reviewers, by measuring these phenotypes in physiological oxygen tensions and in a tumor model. First, we asked whether decreased oxygen was sufficient to promote the proliferation of SDH impaired cells in the absence of a CI inhibitor. We conducted paired experiments culturing 143B cells in either 21% O_2_ (in a standard tissue culture incubator) or at 11%, 5%, 3%, or 1% O_2_ in a hypoxia chamber (See “Proliferation Assays” in the Materials and methods section for details). Interestingly, we found that physiological oxygen tensions were unable to significantly increase cell proliferation of AA5 treated cells from 3% or above, but that severe hypoxia at 1% O2 was able to partially improve the proliferation of AA5 treated cells, albeit to a lesser degree than rotenone (Revised Manuscript Figure 3—figure supplement 1D). These data suggest that severe hypoxia may benefit the proliferation of SDH deficient tumor cells and also provide plausibility that CI loss would be beneficial to SDH deficient cells at physiological oxygen concentrations in vivo.

Reviewer #1 (Recommendations for the authors):1. The Asp recovery experiments are hard to interpret.The panels showing Asp levels (e.g. in Figure 1D, 2B, 1G, 3E, 1H) would be easier for the reader to interpret if they were shown with a linear scale on the y-axis. On a linear scale the recovery of Asp by rotenone treatment in SDH inhibited cells (Figure 2B) would look quite minor compared to the loss of Asp that is caused by SDH inhibition. The authors also use different non-linear scales to show Asp levels in different panels of the same figure (e.g. Figure 2G and J), which is especially confusing.Comparison of Asp rescue between different conditions is also challenging. For example, the recovery of Asp caused by "AA5 + pyruvate" (Figure 1D) appears to be greater than that caused by "AA5 and rotenone" (Figure 2B), yet the recovery of proliferation by "AA5 + pyruvate" is less. As another example, Asp levels in "AA5+mitoLbNOX" (Figure 3E) appear greater than Asp levels in "AA5+ rotenone" (Figure 2B). However, only "AA5 + rotenone" rescues proliferation. In other words, the relationship between Asp recovery and proliferation is not clear.Part of the problem is that the authors use relative ion counts as well as non-linear scales. They also show the data for closely related conditions in different figures and panels. The best way to resolve the problem would be to present the data as absolute values rather than relative ion counts (e.g. fmol of Asp per cell). If this is not available, the authors should present the (normalized) ion counts (i.e. not the relative ion counts) on a linear scale, with error bars representing the variance of the ion counts. The authors do this in Figure 4B and it is much clearer.Do the authors have the data from a control sample (143B wild-type cells) that could be included in Figure 4B, so that the reader could see how the rescue of Asp by SDHB KO compares to wild-type levels? Incidentally, Figure 4A would also be easier to interpret if the proliferation rate of wild-type 143B cells were also included.More generally, if the experiments were performed at the same time, it would be helpful if the data for all the conditions were to be collated in a Sup. Table so that the recovery of Asp between different conditions can easily be compared.As a result of this uncertainty in the extent of Asp recovery, the proposed rescue of Asp as a factor in facilitating proliferation in SDH and CI inhibition is not convincing. Right now, a key question can't be assessed by the reader: how much Asp is needed for proliferation? The quantitative relationship between Asp levels and proliferation should be made clearer.

These concerns are addressed in detail in response to Essential Revisions Comment 1. In brief, we have changed the presentation of aspartate levels to linear scales, standardized the presentation of aspartate levels amongst conditions, included ion counts for all LCMS data in Supplementary File 1, repeated experiments including additional control samples, and generated dose response curves that compare the effects of mitochondrial inhibitors on aspartate and proliferation to contextualize changes in aspartate throughout the manuscript (see revised submission Figure 1C; Figure 1—figure supplement 1K-L). We believe this latter point is a substantial improvement to the manuscript and so thank the reviewer again for the suggestion.

2. The data relating NAD+/NADH to the rescue of Asp and proliferation is not entirely convincing and is quite hard to interpret (Figure 1F, G).i) It is not quite clear what is being measured in Figure 1F. The NAD+/NADH ratio reported in the Figure (i.e. ratio is approx. "1" for controls) is far lower than values that are reported in the literature for this cell line using a different approach (e.g. in the corresponding authors previous work with this cell line) or using carefully validated methods to extract NAD from other cell lines (e.g. by the Rabinowitz group, 2018, Antioxid Redox Signal.). It may be too much to ask the authors to redo the experiments using standards (to provide absolute amounts or concentrations), but it would easier to interpret claims about the role of redox state in these processes easier if the ratio reported here more closely reflects the intracellular NAD+/NADH ratio.Perhaps the authors are reporting a relative ratio (a ratio of ratios?) that is normalized to the control group? If so, the figure legend isn't clear. Furthermore, having a ratio of ratios is very hard to interpret and can't be compared to previous studies or between experiments in this study. Could the authors please report either the NAD+/NADH ratio that they have measured for each group (on a linear scale), or (better) report the measured values for NAD+ and NADH (e.g. fmol per cell).ii) Could the authors comment on the lack of an increase in the NAD+/NAD ratio in control cells when they are treated with pyruvate. Isn't this unexpected and inconsistent with previous studies? Several previous studies in very similar systems have all shown an increase in the NAD+/NADH following pyruvate treatment (e.g. Gui et al. 2016 Cell Metabolism), confirming many older studies.iii) The authors use global NAD+/NADH measurements to infer compartment specific changes. The follow-up experiments using LbNOX are quite an indirect way to check whether this is the case. FLIM would provide some indication if the cytoplasmic and mitochondria pools of NAD(P)H are acting as proposed by the authors.

These concerns are addressed in detail in response to Essential Revisions Comment 3. Overall, we agree with the reviewer’s concerns – normalizing the ratios was intended to avoid the misperception that ion count ratios of NAD+ and NADH represent the true molar ratio, which cannot be done without isotopically labeled standards to account for matrix effects and differences in ionization efficiencies between analytes. Nonetheless, as detailed above, we changed all LCMS derived NAD+/NADH ratios to untransformed ratios of ion counts (with corresponding Y axis label changes to make this more obvious), explicitly describe these caveats in the Materials and methods section, and included all ion counts in Supplementary File 1. As suggested by the reviewer we also measured compartment specific NAD+/NADH changes. While we did not have access to FLIM for this purpose, we were able to leverage a rapid subcellular enrichment protocol combined with an enzymatic assay to measure compartmental changes to NAD+/NADH (see revised submission Figure 3B).

Related to comment ii, above – We agree that culturing cells in the presence of pyruvate is expected to increase NAD+/NADH, as was found in the paper mentioned, and many others. However, we respectfully disagree that our data are inconsistent with those studies. As the last author on this study was also a lead author on the referenced Gui et al. 2016 Cell Metabolism manuscript, we are able to directly compare the data from each. One issue with comparing across studies is that basal NAD+/NADH ratios were higher in the Gui paper, likely a result of using an enzyme-based assay specific for NAD+/NADH rather than LCMS (which has the caveats mentioned above). So, for the sake of comparing changes in NAD+/NADH in relevant treatment conditions across studies, we normalized each to the NAD+/NADH ratio measured in vehicle treated cells in DMEM-PYR. Importantly, regardless of differences in the cell line or measurement method, we see qualitatively similar changes upon the addition of pyruvate, rotenone, and both (Author response image 2). We believe these data highlight that, while the ratio of ion counts detected by LCMS may not reflect a true molar ratio in NAD+/NADH, it is still a valid method to measure changes in NAD+/NADH across treatment conditions.

**Author response image 2. sa2fig2:** Comparing relative changes in NAD+/NADH when adding pyruvate and/or rotenone, across studies. NAD+/NADH measurements from the indicated figure panels in Gui et al. 2016 or this revised manuscript, with each made relative to vehicle treated cells cultured in pyruvate-free DMEM.

3. A full mechanistic model is not clearly presented by the authors.The scheme in Figure 1G presumably doesn't reflect the author's full model, and it would be informative for the authors to present their proposed mechanism somewhere in the manuscript. Right now, it is not clear (for example) how the increase of mitochondrial NADH by dual inhibition of SDH and CI leads to an increase in Asp, or how the rescued Asp levels restore proliferation in this system.The authors address how mitochondrial NADH increase may increase Asp in the Discussion. However, the manuscript would be strengthened if these questions were addressed experimentally. For example, it is straightforward to test whether reductive carboxylation of glutamine or anaplerosis by pyruvate carboxylase is responsible for the rescue of Asp when both SDH and CI are inhibited.Furthermore, the link between Asp levels and proliferation has been extensively described in other conditions, but not in this specific condition. Given that the role and "direction" of Asp metabolism varies markedly between different conditions (e.g., Birsoy et al. 2015 vs Gaude et al. 2018) the authors should test which model is operating here. E.g. are MDH1 and GOT1 operating in reverse to generate Asp that is used for biosynthetic reactions, or are GOT1 and MDH1 is operating in the forward direction to consume Asp in order to generate NAD+ for aerobic glycolysis (perhaps more plausible here?).As a whole the manuscript covers a lot of previously covered ground (e.g. loss of SDH causes loss of CI in tumors (e.g. Fendt group)), loss of SDH and CI creates a requirement for Asp synthesis (Gottlieb group). A clear mechanism of how mitochondrial NADH rescues Asp levels and proliferation would contribute much to its novelty.

We thank the reviewer for raising these issues, which are addressed in detail in Essential Revisions Comment 2. Briefly, we investigated these variables by conducting stable isotope tracing, using inhibitors of reductive carboxylation of glutamine, and making genetic knockouts for PC, MPC1, GOT2, GOT1, and MDH1 to reveal mechanisms for how CI inhibition promotes aspartate synthesis and cell proliferation in SDH deficient cells. These experiments generated a substantial amount of new data, with three new Figures added to the manuscript (Revised Manuscript Figures 3-5).

4. Key data are not always consistent between panelsFor example, the recovery of proliferation in SDH inhibited cells by rotenone appears somewhat inconsistent between experiments (e.g. Figure 2A and Figure S3B). Some variability is to be expected, but, in this version of the manuscript, the experimental variability isn't reflected in the error bars.

Addressed in detail in Essential Revisions Comment 2.

5. Are the 143B cells used in these experiments capable of pyrimidine salvage?It would be useful to know whether the 143B line that is used for most of the experiments is thymidine kinase negative (e.g, the 143B cells from ATCC are TK-). If so, this should be clearly stated in the methods, as lack of thymidine salvage will impact how these results are interpreted.

The 143B cells used in this study are from ATCC and are indeed TK-. However, this variable is unlikely to be a critical effector of this phenotype since DMEM does not contain thymidine (or any other nucleosides) in its formulation and experiments were done in media containing dialyzed FBS. In addition, our results were recapitulated in several other cell lines that are capable of thymidine salvage, without any discernable differences. Nonetheless, we aim to be fully transparent in our methods and so this information has been added to the Materials and methods (Cell Culture).

6. The rescue by AZD7545 does not add support to the modelThe rescue of proliferation and especially Asp levels by AZD7545 is so small that if is hard to assess its importance. Is a difference between ~27% and ~31% of wild-type Asp levels really meaningful? When placed on a linear scale it would be hard to discern a difference between Asp levels in these groups.

Addressed in detail in Essential Revisions Comment 7.

Reviewer #2 (Recommendations for the authors):1. While quite interesting, the physiological relevance of SDH and ETC complex I inhibition is unclear. Do CI subunits exhibit decreased expression in SDH-deficient tumors in patients? Is this a phenomenon of cell culture or is this also observed in vivo? Could more physiological oxygen tension (or even hypoxia) affect SDH-deficient cell proliferation?

Addressed in detail in Essential Revisions Comment 8. We believe the addition of the added hypoxia proliferation assays and inducible NDI1 xenograft tumor experiment add important physiological context to the study.

2. The data regarding the importance of aspartate biosynthesis in the context of SDH inhibition are mostly correlative and lack important controls.a. Figure 1C – How was the dose selected for Asp treatment in the SLC1A3-expressing cells? While the tracing data confirms the increased uptake of Asp relative to non-SLC1A3 expressing cells (S1B-C), the authors do not report Asp levels or indicate how the dosage was selected.b. Figures 1I and S1K – SDHB expression in UOK269 cells permitted glutamine-dependent aspartate biosynthesis and increased cell proliferation, yet the authors did not report how SDHB expression affected aspartate levels.c. The authors are inconsistent in their interpretation of meaningful changes in aspartate. For example, regarding Figure 1E, the authors investigated the M4 aspartate isotopomer derived from U-13C-glutamine and conclude that pyruvate is insufficient to restore progression through the TCA cycle in AA5 treated cells because it resulted in only a 2.8X increase; yet in Figure 2B the authors observed a 2.3X increase in aspartate levels in cells treated with AA5 + rotenone, which leads the authors to conclude that CI and SDH co-inhibition partially restored aspartate levels.d. Is the increase in proliferation upon SDH and CI inhibition dependent on GOT?

We thank the reviewers for the opportunity to clarify these issues:

a. We selected 1 mM aspartate for the indicated experiment (now Revised Manuscript Figure 1—figure supplement 1D) by identifying an aspartate dose that was sufficiently high to allow robust incorporation of media aspartate into SLC1A3 expressing cells (as measured in Revised Manuscript Figure 1—figure supplement 1C) and would be unlikely to be depleted over the course of a four-day proliferation assay. The goal of this experiment was to determine the magnitude cell proliferation changes upon SDH inhibition when ASP was not limiting, so an excess was desired. Nonetheless we estimated that 1 mM media aspartate would be reasonable based on net glutamine consumption (glutamine consumption – glutamate excretion) that we previously measured (Sullivan et al., 2018 *Nature Cell Biology*) to be around 170 fmol/cell*hour in 143B cells – resulting in a consumption of around 2.5 mM glutamine from the media during a standard proliferation assay (based on ~6E7 total cell-hours, 4 mL media with 4 mM glutamine starting concentration). Since glutamine is used for intracellular demands for glutamine, glutamate, aspartate, and aspartate fates (asparagine, pyrimidines, purines) we estimate that roughly 50% (+/- 25%) of glutamine consumption would be used towards aspartate and its fates, and so 1 mM aspartate would approximately meet those demands in the absence of functional aspartate synthesis.

b. Our apologies for this oversight – this data is now available in Revised Manuscript Figure 1—figure supplement 1N and Figure 8—figure supplement 1E.

c. We agree that changes in aspartate levels could be better presented and have overhauled our presentation of them (see reply to Essential Revisions Comment 1). We hope the reviewer will agree that the interpretation of meaningful aspartate changes is now consistent throughout the manuscript, especially when contextualized by Revised Manuscript Figure 1C.

d. This question is addressed in detail in response to Essential Revisions Comment 2 and by Revised Manuscript Figure 6.

3. Overstatements of results reduce clarity of the findings. For example:a. The authors argument that "SDH inhibition blocks proliferation, which is restored by aspartate but not electron acceptors" (Figure 1 title) seems to be an overstatement. Though aspartate seems to increase cell proliferation to a greater degree than electron acceptors, data presented in 1A and 1C indicate that pyruvate, AKB, and LbNOX are all sufficient to increase cell proliferation.b. The presented correlations between aspartate levels and proliferation are not sufficient evidence that aspartate biosynthesis is the primary mode of supporting cell proliferation in SDH-deficient cells. For example, in Supp. Figure 2B-C aspartate levels are not rescued to the same extent as proliferation across the panel of cell lines.c. Data regarding the combinatorial treatment of cells with AA5+AZD7545 (Figures 3G-H) shows only slight changes to proliferation and aspartate levels. This adds little to the argument that mitochondria-specific alterations in NAD+/NADH and aspartate biosynthesis are responsible for proliferation in the context of SDH inhibition.

We are happy to have the ability to correct the text to avoid overstating our results:

a. This is a good point, we have changed the Figure 1 title to: “SDH inhibition blocks proliferation, which is incompletely rescued by electron acceptors but robustly restored by aspartate”

b. We have provided a detailed response with corresponding new data in the response to Essential Revisions Comment 2, which we hope the reviewer will agree provides data to support our conclusion that aspartate synthesis is a metabolic limitation for cell proliferation in SDH impaired cells. While we agree that different cell lines have different relationships between the degree of aspartate and proliferation rescue upon CI co-inhibition, these findings are in line with the non-linear relationship we found between aspartate levels and proliferation rate during mitochondrial inhibition (Revised Manuscript Figure 1C). The variability in aspartate levels and proliferation rate is likely a consequence of cell intrinsic differences in starting aspartate concentration, efficacy of the inhibitors to impair their target, and sensitivity differences to changes in aspartate levels. Nonetheless, we believe that the similar qualitative changes in aspartate and proliferation upon SDH and CI inhibition across cell lines supports the generalizability of our findings.

c. This comment is addressed in detail in response to Essential Revisions Comment 7.

4. Statistical analysis was frequently missing or improperly conducted. For example, regarding panel 1A the authors noted that AA5 and rotenone treatment decreased cell proliferation and that co-treatment with pyruvate, AKB, LbNox, or Asp restored proliferation. Statistical comparisons were made between AA5 and rotenone for all groups; however, based on the text, the statistical analysis should be restricted to comparisons between groups for each AA5 and rotenone.

This point is well taken, and we have reorganized Figure 1A to make the appropriate comparisons clear and added appropriate statistical tests throughout the manuscript.

5. The data presentation throughout the manuscript is confusing. For example, in some panels the y-axis is log-scale while in complimentary panels the y-axis is linear, and for some graphs the y-axis does not start at 0.

We have overhauled our data presentation in accordance with the reviewer critiques, see detailed response to Essential Revisions Comment 1.

6. In Figure 1, SDHB expression in UOK269 cells led to a proliferation rate that was nearly 5X higher than the control cells. Yet in Figure 2l, the proliferation rate of UOK269 cells expressing SDHB is the same as those not expressing SDHB. Why are the proliferation rates different?

We believe that apparent inconsistency is a misunderstanding based on comparing cells grown in different media conditions, a result of insufficient labeling clarity on our previous draft. Please see the detailed discussion of this point in response to Essential Revisions Comment 5.

Reviewer #3 (Recommendations for the authors):During our reading of the manuscript, we felt the model the authors proposed within this work was not consistent or clear throughout the text. We feel that some attention could be made to try to emphasize and clarify this model throughout and would greatly assist other readers of the work. Currently, we feel like we as readers had to do a lot of work to try to make sense of the model and how the data are supporting that model.

We thank the reviewer bringing this to our attention and providing us the opportunity to clarify our model(s) and improve the manuscript. As detailed in response to Essential Revisions Comment 2, we have added a substantial amount of new data to better describe the metabolic pathways governing aspartate synthesis in this system. In addition, we have included models in each main figure to help underscore the conclusions of each, and the manuscript as a whole. We believe, and hope the reviewer will agree, that the inclusion of these models will help readers understand the conclusions of our work.

The authors state that "this work provides the first mechanistic evidence that CI loss can directly support core metabolic functions":1) We do not feel the mechanistic data provided in the current form of the manuscript are strong enough to make this claim.2) Perhaps this is the first instance of loss of CI & II together are beneficial, but there seem to be several other works that seem to make a case that loss of CI is beneficial to core metabolism processes, such as:a. PMID: 29915083b. PMID: 18784283

We are glad to have the opportunity to avoid overstating our data – That statement has been removed from the discussion of the revised submission and the section has been edited accordingly.

We do however feel that the novelty of our phenotype is distinct from the listed papers – The first (Gopal et al.) is an observational study comparing the exome, transcriptome, and metabolome of kidney cancers with and without mutations in mitochondrial CI. We agree that the association of mtDNA CI mutations with the disease raises the possibility that a loss of CI activity is beneficial, however no functional experiments are done in this study to test that hypothesis. It remains to be seen if these cancers (or others associated with mtDNA CI mutations, as mentioned in our Discussion) are truly driven to grow by the loss of CI activity per se, or as a result of another consequence of those mutations. We speculate that the finding that these oncocytomas are exclusively driven by mutations in the mtDNA, and not by mutations in CI subunits encoded by nuclear DNA, would suggest a more complicated mechanism than simply CI loss driving transformation and proliferation. The second paper (Garmier et al.) does investigate the functional effects of CI inhibition by rotenone treatment in Arabidopsis cells, measuring effects on cell growth and metabolic state. Importantly though, the authors note defects in cell growth upon rotenone treatment, in contrast to the effects observed in SDH impaired cells in our study. We believe a major novelty distinction between the listed studies and ours is the finding of experimental conditions where CI inhibition benefits cell proliferation rate and the characterization of the metabolic variables that govern that effect.

While we appreciate that many of the scripts used to generate data within this work are using pre-existing toolkits, it is still important to present the code used (how the toolkits were used exactly) to aid in the reproducibility of the work. These scripts could be included in the supplement, or hosted on a public repository, such as GitHub. Raw and processed metabolomics data should also be included as supplements, or hosted on a public repository, such as Metabolomics Workbench.In Figure S4F, the authors state what "unsupervised clustering" was used. This is very generic – what was the exact algorithm used in the hierarchal clustering of the proteomics data and why? For example, scipy.cluster.hierarchy.linkage provide 7 different clustering methods (https://docs.scipy.org/doc/scipy/reference/generated/scipy.cluster.hierarchy.linkage.html)

We would also like to ensure we are aiding other groups in their ability to reproduce this work. As the reviewer mentioned, much of the data analysis was done using standard approaches from pre-existing software (e.g., LCMS data analysis by Tracefinder). Most other data (proliferation assays, cloning, western blots, Seahorse assays, Western blots, CI activity assays, etc.) did not require the generation of code beyond simple analyses in Excel or Prism and so does not seem applicable, although we are happy to add any missing information that the reviewer deems pertinent. We will however include all information that is available so we can to support the goal of transparency and reproducibility. Processed ion counts for all LCMS experiments are now included in Supplementary File 1. In addition, raw metabolomics files for key figures were uploaded onto Metabolomics Workbench with appropriate documentation. For the mentioned unsupervised clustering we have added the following statement to the methods section under TMT-Quantitative Mitochondrial Proteomics.

“Unsupervised clustering of the 16 proteomic samples was performed using the z-score based R (version 4.0.3) package GMD (Zhao & Sandelin, 2012) and visualized with the heatmap.3 function.”

There are also a couple of changes to the presentation of the figures that would improve the manuscript:– Figure 3B/F: There is a line showing up at the top of each of these subpanels in our version of the manuscript.Perhaps bar plot colors could be more standardized between sub-panels for the different conditions?

We thank the reviewer for identifying these issues and have worked to improve the aesthetics of our presentation, which we hope will aid in easier reader comprehension.

In summary, while we do not feel like this manuscript in its current form meets the criteria for publication in eLife, we are certainly positive about its potential and suggest filling in some of the remaining gaps outlined above and some mechanisms to explain the observations would make the work more complete and suitable for publication.

We again thank this reviewer, and the others, for their positive comments about the potential of our work. We believe the incorporating the data from the suggested experiments into the revised manuscript has addressed many of the mechanistic gaps and as a result increased the scope and impact of the work.